# ADAR1 averts fatal type I interferon induction by ZBP1

Huipeng Jiao[1,2,13], Laurens Wachsmuth[1,2,13], Simone Wolf[1,2], Juliane Lohmann[1,2], Masahiro Nagata[1,2], Göksu Gökberk Kaya[1,2], Nikos Oikonomou[1,2], Vangelis Kondylis[3,4], Manuel Rogg[5], Martin Diebold[6], Simon E. Tröder[2,7], Branko Zevnik[2,7], Marco Prinz[6,8,9], Christoph Schell[5], George R. Young[10], George Kassiotis[11,12] & Manolis Pasparakis[1,2,4 ✉]

Mutations of the *ADAR1* gene encoding an RNA deaminase cause severe diseases associated with chronic activation of type I interferon (IFN) responses, including Aicardi–Goutières syndrome and bilateral striatal necrosis[1–3]. The IFN-inducible p150 isoform of ADAR1 contains a Zα domain that recognizes RNA with an alternative left-handed double-helix structure, termed Z-RNA[4,5]. Hemizygous *ADAR1* mutations in the Zα domain cause type I IFN-mediated pathologies in humans[2,3] and mice[6–8]; however, it remains unclear how the interaction of ADAR1 with Z-RNA prevents IFN activation. Here we show that Z-DNA-binding protein 1 (ZBP1), the only other protein in mammals known to harbour Zα domains[9], promotes type I IFN activation and fatal pathology in mice with impaired ADAR1 function. ZBP1 deficiency or mutation of its Zα domains reduced the expression of IFN-stimulated genes and largely prevented early postnatal lethality in mice with hemizygous expression of ADAR1 with mutated Zα domain (*Adar1^{mZα/−}* mice). *Adar1^{mZα/−}* mice showed upregulation and impaired editing of endogenous retroelement-derived complementary RNA reads, which represent a likely source of Z-RNAs activating ZBP1. Notably, ZBP1 promoted IFN activation and severe pathology in *Adar1^{mZα/−}* mice in a manner independent of RIPK1, RIPK3, MLKL-mediated necroptosis and caspase-8-dependent apoptosis, suggesting a novel mechanism of action. Thus, ADAR1 prevents endogenous Z-RNA-dependent activation of pathogenic type I IFN responses by ZBP1, suggesting that ZBP1 could contribute to type I interferonopathies caused by *ADAR1* mutations.

Z-DNA and Z-RNA are nucleic acids with an alternative left-handed double-helix structure and have poorly understood biological function[10–13]. These Z-form nucleic acids are recognized by specific protein domains, termed Zα domains, which bind Z-DNA and Z-RNA in a conformation-specific manner[5,9,14,15]. Two proteins are known to harbour Zα domains in mammals, namely adenosine deaminase acting on RNA 1 (ADAR1) and Z-DNA-binding protein 1 (ZBP1, also known as DAI or DLM-1)[5,9,14]. ADAR1 is produced in two isoforms, the constitutively expressed nuclear p110 and the interferon (IFN)-inducible cytosolic p150 that contains a Zα domain[1]. ADAR1 p150 edits self-RNA derived predominantly from endogenous retroelements (EREs) to prevent its recognition by the cytosolic RNA sensor melanoma differentiation-associated gene 5 (MDA5) and the activation of mitochondrial antiviral signalling (MAVS)-dependent pathogenic type I IFN responses[16–19]. *ADAR1* mutations mapping to the Zα domain combined with alleles resulting in loss of ADAR1 or specifically its

p150 isoform were shown to cause Aicardi–Goutières syndrome (AGS) and bilateral striatal necrosis (BSN) in human patients[2,3] and severe MDA5–MAVS-mediated type I IFN-dependent pathology in mice[6–8], indicating that the interaction of ADAR1 with Z-RNA is required to prevent activation of pathogenic IFN responses. ZBP1 is an IFN-inducible protein that senses viral and endogenous Z-form nucleic acids via its Zα domains and triggers cell death to induce antiviral immunity, but also causes tissue damage and inflammation[20–27]. Previous studies have shown that ZBP1 causes cell death in vivo and in vitro by activating receptor-interacting protein kinase 3 (RIPK3) in a RIP homotypic interaction motif (RHIM)-dependent manner, which then phosphorylates mixed-lineage kinase-like (MLKL) to induce necroptosis and can also engage RIPK1 to trigger caspase-8-dependent apoptosis[20,21,23,25,27]. We reasoned that ZBP1 may functionally interact with ADAR1 to regulate cellular responses to Z-RNA and assessed its role in the activation of type I IFN-dependent pathology in mice with *Adar1* mutations.

[1]Institute for Genetics, University of Cologne, Cologne, Germany. [2]Cologne Excellence Cluster on Cellular Stress Responses in Aging-Associated Diseases (CECAD), University of Cologne, Cologne, Germany. [3]Institute for Pathology, Medical Faculty and University Hospital of Cologne, University of Cologne, Cologne, Germany. [4]Center for Molecular Medicine (CMMC), University of Cologne, Cologne, Germany. [5]Institute of Surgical Pathology, Faculty of Medicine, Medical Center–University of Freiburg, Freiburg, Germany. [6]Institute of Neuropathology, Medical Faculty, University of Freiburg, Freiburg, Germany. [7]In Vivo Research Facility, Faculty of Medicine and University Hospital Cologne, University of Cologne, Cologne, Germany. [8]Centre for NeuroModulation (NeuroModBasics), University of Freiburg, Freiburg, Germany. [9]Signalling Research Centres BIOSS and CIBSS, University of Freiburg, Freiburg, Germany. [10]Bioinformatics and Biostatistics STP, London, UK. [11]Retroviral Immunology, The Francis Crick Institute, London, UK. [12]Department of Infectious Disease, Imperial College London, London, UK. [13]These authors contributed equally: Huipeng Jiao, Laurens Wachsmuth. ✉e-mail: pasparakis@uni-koeln.de

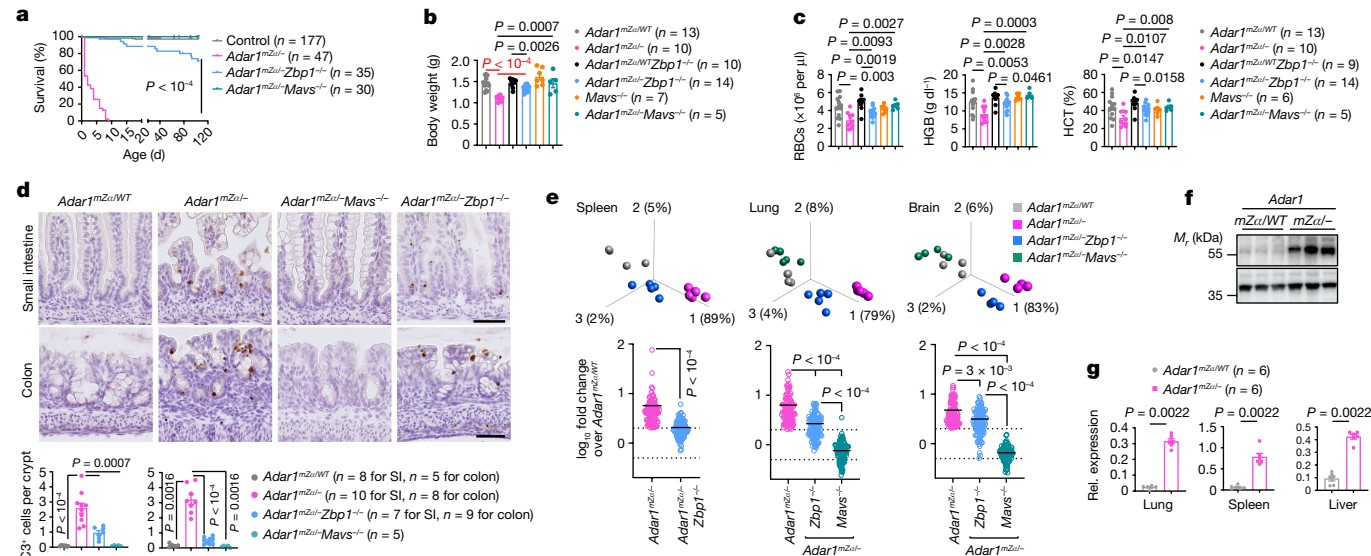

**Fig. 1 | ZBP1 contributes to IFN induction and early postnatal lethality in mice hemizygously expressing ADAR1 with a mutated Zα domain.**
**a**, Kaplan–Meier survival graph of mice with the indicated genotypes. *P* values were calculated by two-sided Gehan–Breslow–Wilcoxon test. Control mice included littermates with the *Adar1*$^{mZα/WT}$, *Adar1*$^{WT/-}$ or *Adar1*$^{WT/WT}$ genotype. **b,c**, Body weight (**b**) and RBC counts and HGB and HCT levels in the blood (**c**) for mice with the indicated genotypes at P1. *Mavs*$^{-/-}$ mice included littermates with the *Adar1*$^{mZα/WT}$*Mavs*$^{-/-}$, *Adar1*$^{WT/-}$*Mavs*$^{-/-}$ or *Adar1*$^{WT/WT}$*Mavs*$^{-/-}$ genotype. **d**, Representative images of small intestine (SI) and colon sections immunostained for CC3 and graphs depicting quantification of CC3$^+$ cells in mice with the indicated genotypes. Scale bars, 50 μm. **e**, Top, PCA on RNA-seq data from spleen, lung and brain tissues isolated from mice with the indicated genotypes at P1. PCA was based on genes differentially expressed between *Adar1*$^{mZα/-}$ and *Adar1*$^{mZα/WT}$ control mice, including 1,594 (*P* ≤ 0.05, *q* ≤ 0.05, ≥2-fold change), 657 (*P* ≤ 0.05, *q* ≤ 0.05, ≥2-fold change) and 379 (*P* ≤ 0.05,

≥2-fold change) genes for the spleen, lung and brain, respectively (Supplementary Table 2). Bottom, fold change (log$_{10}$) in expression of the 93 ISGs commonly upregulated in all three tissues examined (Supplementary Table 2). Symbols represent mean values of individual genes, solid lines show mean expression of the 93 genes and dashed lines denote the twofold change boundaries. *P* values were calculated by two-sided non-parametric Mann–Whitney test (spleen) and Kruskal–Wallis test with Dunn's post hoc test for multiple comparisons (lung and brain). *n* = 5 for lung and spleen; *n* = 4 for brain. **f**, Immunoblot of ZBP1 in lung protein extracts from *Adar1*$^{mZα/WT}$ and *Adar1*$^{mZα/-}$ mice at P1. Lanes represent individual mice. GAPDH was used as a loading control. **g**, qRT–PCR analysis of *Zbp1* mRNA expression in the indicated tissues from mice at P1. In **b**–**d** and **g**, dots represent individual mice, bar graphs show mean ± s.e.m. and *P* values were calculated by two-sided non-parametric Mann–Whitney test. For gel source data, see Supplementary Fig. 1.

## ADAR1 Zα domain inactivation induces IFN responses

To address the role of the ADAR1 Zα domain, we generated knock-in mice expressing ADAR1 with two substitutions disrupting its interaction with Z-RNA (N175D/Y179A)[28,29] (Extended Data Fig. 1a), hereafter referred to as *Adar1*$^{mZα/mZα}$ mice. *Adar1*$^{mZα/mZα}$ mice were born at the expected Mendelian ratio, were viable and fertile and did not develop apparent pathology at least until the age of 1 year (Extended Data Fig. 1b–d). However, RNA sequencing (RNA-seq) showed upregulation of 57 genes in lung tissues from 4- to 5-month-old *Adar1*$^{mZα/mZα}$ mice, all of which were functionally linked to type I IFN responses, compared with *Adar1*$^{mZα/WT}$ and wild-type C57BL/6N animals (Extended Data Fig. 1e and Supplementary Table 1). Quantitative PCR with reverse transcription (qRT–PCR) analysis confirmed upregulation of a selected set of IFN-stimulated genes (ISGs) in lung, spleen and liver tissue from *Adar1*$^{mZα/mZα}$ mice compared with control littermates (Extended Data Fig. 1f). Therefore, disruption of the ADAR1 Zα domain caused elevated expression of ISGs in the absence of overt tissue pathology, in line with recent reports[6,7,30]. Mutations affecting the ADAR1 Zα domain were found to cause AGS and BSN when combined with alleles resulting in loss of ADAR1 p150 expression[3,31]. To model this condition, we generated *Adar1*$^{mZα/-}$ mice and found that they developed a severe phenotype characterized by reduced body weight and early postnatal lethality (Fig. 1a,b). Haematological analysis at postnatal day (P) 1 showed reduced numbers of red blood cells (RBCs) as well as diminished haemoglobin (HGB) and haematocrit (HCT) levels in *Adar1*$^{mZα/-}$ mice compared with *Adar1*$^{mZα/WT}$ mice (Fig. 1c), in line with the important role of ADAR1 in erythropoiesis[32]. Histological examination showed altered architecture with increased numbers of

epithelial cells immunostained for cleaved caspase-3 (CC3) in the small intestine and colon of *Adar1*$^{mZα/-}$ pups, whereas other organs including the liver, lung, heart, kidney and brain did not show prominent pathological features (Fig. 1d and Extended Data Fig. 2a–d). RNA-seq analysis of lung, brain and spleen showed increased expression of several genes in *Adar1*$^{mZα/-}$ mice compared with *Adar1*$^{mZα/WT}$ littermates, the majority of which were linked to type I IFN responses (Extended Data Fig. 3 and Supplementary Table 2). Comparison of RNA-seq data from lung, brain and spleen identified a set of 93 genes, all ISGs, that were consistently upregulated in all three tissues from *Adar1*$^{mZα/-}$ compared with *Adar1*$^{mZα/WT}$ mice (Supplementary Table 2). A smaller number of genes were downregulated in *Adar1*$^{mZα/-}$ mice, particularly in the spleen, most of which were functionally linked to erythrocyte development, in line with the impaired erythropoiesis observed (Extended Data Fig. 3). Crossing to *Mavs*$^{-/-}$ mice (Extended Data Fig. 2g) rescued the lethal phenotype of *Adar1*$^{mZα/-}$ mice, as *Adar1*$^{mZα/-}$*Mavs*$^{-/-}$ animals appeared healthy, did not show upregulation of ISGs and reached adulthood without displaying signs of pathology at least up to the age of 15 weeks (Fig. 1a–e and Extended Data Fig. 2). Therefore, in agreement with recent reports[6–8], hemizygous expression of ADAR1 with a mutated Zα domain induced a strong MDA5–MAVS-dependent type I IFN response, causing severe early postnatally lethal pathology in mice.

## ZBP1 causes pathology in *Adar1*$^{mZα/-}$ mice

The finding that hemizygous expression of ADAR1 with a mutated Zα domain causes severe pathology in both humans and mice indicates that the capacity of ADAR1 to bind Z-RNA is critical to prevent

pathogenic IFN responses, but the underlying mechanisms remain poorly understood. ZBP1 expression was strongly increased in tissues from $Adar1^{mZα/-}$ mice, in line with the pronounced upregulation of ISGs (Fig. 1f,g and Supplementary Table 2). We therefore reasoned that ZBP1 could be functionally involved in the pathology of $Adar1^{mZα/-}$ mice and addressed its role by generating and analysing $Adar1^{mZα/-}Zbp1^{-/-}$ animals. Notably, ZBP1 deficiency largely prevented early postnatal lethality in $Adar1^{mZα/-}$ mice, with about 70% of $Adar1^{mZα/-}Zbp1^{-/-}$ mice surviving at least up to the age of 15 weeks (Fig. 1a). Inspection of mice at P1 showed partially restored body weight as well as RBC, HGB and HCT values in $Adar1^{mZα/-}Zbp1^{-/-}$ compared with $Adar1^{mZα/-}$ mice (Fig. 1b,c). Moreover, ZBP1 deficiency reduced the numbers of CC3[+] cells and ameliorated the intestinal pathology of $Adar1^{mZα/-}$ mice (Fig. 1d and Extended Data Fig. 2a). To gain insight into the ZBP1-dependent mechanisms driving disease pathogenesis in $Adar1^{mZα/-}$ mice, we compared the gene expression profiles from $Adar1^{mZα/-}$ and $Adar1^{mZα/-}Zbp1^{-/-}$ pups at P1 by RNA-seq. Principal-component analysis (PCA) of differentially expressed genes showed that $Adar1^{mZα/-}Zbp1^{-/-}$ samples clustered distinctly between $Adar1^{mZα/-}$ and $Adar1^{mZα/-}Mavs^{-/-}$ or $Adar1^{mZα/WT}$ samples (Fig. 1e, top). Moreover, comparison of the expression of the set of 93 ISGs found to be upregulated in all three tissues showed that ZBP1 deficiency significantly suppressed the IFN response in the spleen, lung and brain of $Adar1^{mZα/-}$ mice (Fig. 1e, bottom). However, ZBP1 deficiency did not prevent ISG expression and all pathological features of $Adar1^{mZα/-}$ mice as efficiently as MAVS knockout, suggesting that ZBP1 probably acts to augment the MDA5–MAVS-dependent pathogenic IFN response. Together, these results identified a critical role for ZBP1 in promoting type I IFN responses and the severe early postnatally lethal pathology caused by hemizygous expression of ADAR1 with a mutated Zα domain.

Adult $Adar1^{mZα/-}Zbp1^{-/-}$ mice showed lower body weight as well as mildly reduced RBC, HGB and HCT values compared with littermate control animals, whereas $Adar1^{mZα/-}Mavs^{-/-}$ and $Adar1^{mZα/-}Zbp1^{-/-}Mavs^{-/-}$ mice exhibited normal body weight and blood values (Fig. 2a,b and Extended Data Fig. 4a). Heterozygous MAVS knockout partially rescued the early postnatal lethality of $Adar1^{mZα/-}$ mice, with about 50% of $Adar1^{mZα/-}Mavs^{WT/-}$ mice surviving at least up to the age of 15 weeks, when they showed strongly diminished body weight as well as RBC, HGB and HCT values compared with control and $Adar1^{mZα/-}Mavs^{-/-}$ mice (Fig. 2a–c and Extended Data Fig. 4a). Notably, additional loss of ZBP1 rescued the lethality and normalized the body weight and RBC, HGB and HCT values in $Adar1^{mZα/-}Zbp1^{-/-}Mavs^{WT/-}$ mice (Fig. 2a–c and Extended Data Fig. 4a). Histological analysis of organs from 15-week-old mice showed severe glomerular mesangial sclerosis (MS) in kidneys from $Adar1^{mZα/-}Mavs^{WT/-}$ mice, characterized by deposition of extracellular matrix, complete obstruction of the glomerular capillary convolute and focal mesangiolysis, compared with the normal tissue architecture in control and $Adar1^{mZα/-}Mavs^{-/-}$ mice (Fig. 2d). ZBP1 deficiency strongly ameliorated this sclerosing phenotype, as reflected by decreased deposition of extracellular matrix and patent capillary lumina in kidneys from $Adar1^{mZα/-}Zbp1^{-/-}Mavs^{WT/-}$ and $Adar1^{mZα/-}Zbp1^{-/-}$ mice (Fig. 2d). Moreover, $Adar1^{mZα/-}Mavs^{WT/-}$ mice showed signs of pericentral sinusoidal dilatation in the liver and mild hyperplasia with small numbers of dying cells in the small intestine, which were ameliorated by ZBP1 deficiency (Extended Data Fig. 4b). RNA-seq analysis identified 399 genes, mostly ISGs, that were upregulated in lungs from 15-week-old $Adar1^{mZα/-}Mavs^{WT/-}$ mice compared with $Adar1^{mZα/WT}$ and wild-type control mice (Extended Data Fig. 5 and Supplementary Table 3). Using this set of 399 ISGs, we compared the effect of ZBP1 deficiency alone or in combination with heterozygous or homozygous MAVS knockout on the IFN response. $Adar1^{mZα/-}Zbp1^{-/-}$ mice clustered distinctly from $Adar1^{mZα/-}Mavs^{WT/-}$ mice and had an overall mildly reduced ISG signature, suggesting that ZBP1 deficiency had a stronger effect than MAVS heterozygosity in limiting the IFN response (Fig. 2e). Additional ZBP1 deficiency considerably suppressed ISG expression in $Adar1^{mZα/-}Mavs^{WT/-}$ mice, with $Adar1^{mZα/-}Zbp1^{-/-}Mavs^{WT/-}$ mice clustering closer

to wild-type animals than to $Adar1^{mZα/-}Mavs^{WT/-}$ animals (Fig. 2e). ISG expression was largely normalized in $Adar1^{mZα/-}Mavs^{-/-}$ mice, although these animals showed small but statistically significant differences in expression compared with $Adar1^{mZα/WT}$ controls (Fig. 2e), indicating that MAVS-independent mechanisms also contribute to the IFN response. Interestingly, double deficiency of ZBP1 and MAVS could fully normalize the expression of ISGs, with expression in $Adar1^{mZα/-}Zbp1^{-/-}Mavs^{-/-}$ mice indistinguishable from that in $Adar1^{mZα/WT}$ or wild-type animals (Fig. 2e), suggesting that ZBP1 contributes to the ISG response also independently of MAVS.

## ZBP1–RIPK3 function in $Adar1^{-/-}$ mice

On the basis of our findings in $Adar1^{mZα/-}$ mice, we reasoned that ZBP1 might also contribute to the severe pathology caused by complete ADAR1 deficiency. However, we did not observe any live $Adar1^{-/-}Zbp1^{-/-}$ mice born from crosses of heterozygous animals, showing that ZBP1 deficiency was not sufficient to rescue the embryonic lethality of $Adar1^{-/-}$ mice (Extended Data Fig. 6a). $Adar1^{-/-}Mavs^{-/-}$ and $Adar1^{-/-}Mda5^{-/-}$ animals develop to term but die during the first postnatal days[19,33], showing that MDA5–MAVS-independent signalling causes postnatal pathology in $Adar1^{-/-}$ mice. We therefore assessed whether ZBP1 deficiency might synergize with MAVS knockout to rescue the phenotype of $Adar1^{-/-}$ mice. Indeed, we found that about 40% of $Adar1^{-/-}Zbp1^{-/-}Mavs^{-/-}$ mice survived to adulthood, in contrast to $Adar1^{-/-}Mavs^{-/-}$ mice that died shortly after birth (Fig. 2f and Extended Data Fig. 6b). The $Adar1^{-/-}Zbp1^{-/-}Mavs^{-/-}$ mice that survived to adulthood showed reduced body weight but appeared healthy at least until the age of 15 weeks (Extended Data Fig. 6b–d). Histological analysis of different organs did not identify apparent pathology, with only mild hyperplasia and small numbers of dying cells observed in the intestine of $Adar1^{-/-}Zbp1^{-/-}Mavs^{-/-}$ mice (Extended Data Fig. 6e). We reasoned that ZBP1 may cause MAVS-independent pathology in $Adar1^{-/-}$ mice by inducing RIPK3-mediated cell death. Indeed, RIPK3 deficiency alone or in combination with heterozygous knockout of the caspase-8 adaptor protein Fas-associated protein with death domain (FADD) mimicked the effect of ZBP1 deficiency, with about 40% of $Adar1^{-/-}Mavs^{-/-}Fadd^{WT/-}Ripk3^{-/-}$ and $Adar1^{-/-}Mavs^{-/-}Fadd^{WT/WT}Ripk3^{-/-}$ mice surviving at least up to the age of 15 weeks (Fig. 2f). Thus, ZBP1–RIPK3-mediated signalling caused MAVS-independent pathology resulting in early postnatal lethality of $Adar1^{-/-}Mavs^{-/-}$ mice. We then assessed whether ZBP1 could induce cell death in $Adar1^{-/-}Mavs^{-/-}$ mouse embryonic fibroblasts (MEFs). Stimulation with IFNγ, which induced robust expression of ZBP1, did not cause cell death in $Adar1^{-/-}Mavs^{-/-}$ or $Mavs^{-/-}$ MEFs (Extended Data Fig. 6f,g). However, treatment of IFNγ-prestimulated cells with a low amount of the protein translation inhibitor cycloheximide (CHX) caused increased cell death in $Adar1^{-/-}Mavs^{-/-}$ compared with $Mavs^{-/-}$ MEFs, which was prevented by the absence of ZBP1 (Extended Data Fig. 6h). Immunoblot analysis showed that stimulation with IFNγ and treatment with CHX induced phosphorylation of MLKL and increased cleavage of caspase-8 in $Adar1^{-/-}Mavs^{-/-}$ compared with $Mavs^{-/-}$ primary MEFs (Extended Data Fig. 6i). ZBP1 deficiency prevented the phosphorylation of MLKL and reduced the cleavage of caspase-8 in these cells (Extended Data Fig. 6i), suggesting that ZBP1 induces necroptosis and, to a lesser extent, apoptosis. In line with this, combined pharmacological inhibition of RIPK3 and caspases strongly reduced the IFNγ- and CHX-induced death of $Adar1^{-/-}Mavs^{-/-}$ primary MEFs (Extended Data Fig. 6j). Although it remains unclear to what extent the ZBP1–RIPK3-dependent cell death induced in cells treated with CHX relates to the in vivo role of ZBP1–RIPK3 signalling in mediating the pathology of $Adar1^{-/-}Mavs^{-/-}$ mice, we reason that CHX treatment might mimic the activation of protein kinase R (PKR) and resulting inhibition of protein translation in ADAR1-deficient cells[7]. Collectively, these results showed that ZBP1–RIPK3-dependent signalling promoted the MAVS-independent pathology of $Adar1^{-/-}Mavs^{-/-}$ mice.

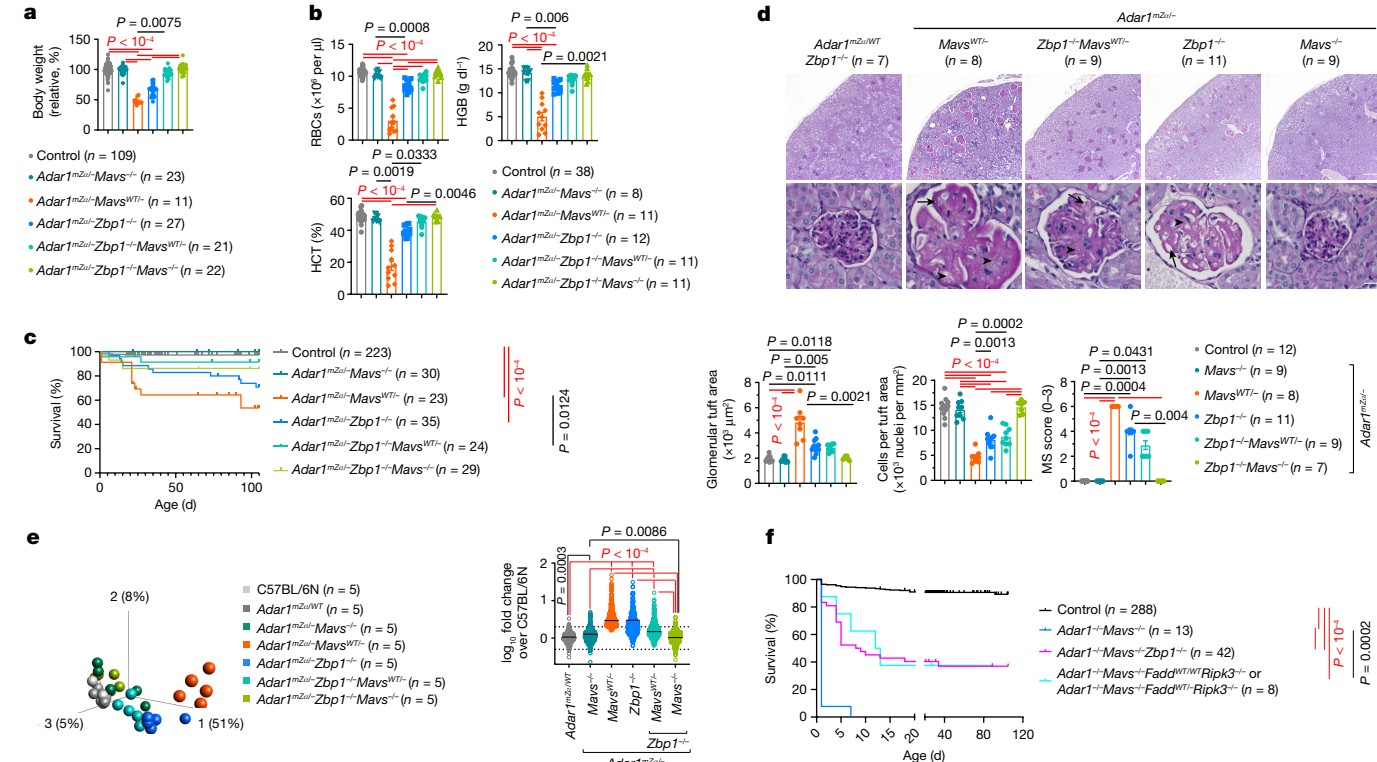

**Fig. 2 | ZBP1 synergizes with MAVS to cause IFN induction and associated pathology in mice with impaired ADAR1 function. a,b**, Relative body weight normalized to that of littermate controls at the age of 10 weeks (**a**) and RBC counts and HGB and HCT levels in the blood of 15-week-old mice (**b**) with the indicated genotypes. **c**, Kaplan–Meier survival graph of mice with the indicated genotypes. Control mice in **a**–**c** included littermates with the *Adar1^mZα/WT*, *Adar1^WT/−* or *Adar1^WT/WT* genotype. Survival data for *Adar1^mZα/−Mavs^−/−* and *Adar1^mZα/−Zbp1^−/−* mice and their littermate controls from Fig. 1a are included for comparison. **d**, Representative images of periodic acid–Schiff (PAS)-stained kidney sections from 15-week-old mice of the indicated genotypes and graphs depicting quantification of histological glomerular MS, glomerular tuft area and cell densities in the glomerular tuft area (arrows indicate non-obliterated capillaries; arrowheads highlight zones of mesangial matrix deposition). Scale bar, 200 μm (top) or 20 μm (bottom). **e**, Left, PCA on lung RNA-seq data from 15-week-old mice with the indicated genotypes. PCA was based on 678 genes differentially expressed between *Adar1^mZα/−Mavs^WT/−* mice and the two control

groups (*Adar1^mZα/WT* and C57BL/6N) combined (*P* ≤ 0.05, *q* ≤ 0.05, ≥2-fold change) (Supplementary Table 3). Right, fold change (log₁₀) in expression of the 399 genes upregulated in *Adar1^mZα/−Mavs^WT/−* mice, calculated for each genotype by comparison to C57BL/6N mice. Symbols represent mean values of individual genes, solid lines represent mean expression of the 399 genes and dashed lines denote the twofold change boundaries. *P* values were calculated by Kruskal–Wallis test with Dunn's post hoc test for multiple comparisons. **f**, Kaplan–Meier survival graph of mice with the indicated genotypes. Control mice included littermates with the *Adar1^WT/−* or *Adar1^WT/WT* genotype. In **a**, **b** and **d**, dots represent individual mice, bar graphs show mean ± s.e.m. and *P* values were calculated by Kruskal–Wallis test with Dunn's post hoc test for multiple comparisons (body weight, HGB, HCT, glomerular tuft area and MS score) or one-way ANOVA with Tukey's correction for multiple comparisons (RBCs and cells per tuft area). In **c** and **f**, *P* values were calculated by two-sided Gehan–Breslow–Wilcoxon test.

## ZBP1 is not involved in *Trex1^−/−* mice

ZBP1 is expressed at very low levels in most tissues under steady-state conditions, but its expression is strongly induced by IFNs. Loss of ADAR1 function could trigger ZBP1-dependent pathology by promoting IFN-inducible upregulation of ZBP1 expression and/or by increasing the abundance of a ZBP1 ligand. To assess whether ZBP1 upregulation functions broadly to drive IFN-dependent pathology, we used another mouse model of type I interferonopathy caused by deficiency in TREX1, a cytosolic 3′–5′ DNA exonuclease found to be mutated in people with AGS, familial chilblain lupus (FCL) and systemic lupus erythematosus (SLE)[34]. TREX1 deficiency in mice triggers cytosolic DNA-induced cyclic GMP-AMP synthase (cGAS)–stimulator of interferon genes (STING)-dependent type I IFN responses, resulting in systemic inflammation primarily manifesting in severe myocarditis[35–38]. Heart tissues from *Trex1^−/−* mice showed profound upregulation of ZBP1; however, *Trex1^−/−Zbp1^−/−* mice were indistinguishable from *Trex1^−/−* and *Trex1^−/−Zbp1^WT/−* littermates in terms of survival, body weight, splenomegaly, heart inflammation and fibrosis (Extended Data Fig. 7a–f). Moreover, ZBP1 deficiency did not inhibit the upregulation of ISGs and

inflammatory cytokines and chemokines in heart tissues of *Trex1^−/−* mice (Extended Data Fig. 7c,g,h). Therefore, although its expression was strongly induced in tissues of *Trex1^−/−* mice, ZBP1 did not contribute to the cytosolic DNA-induced IFN response and pathology caused by TREX1 deficiency, in contrast to its important role in promoting the phenotype caused by impaired ADAR1 function.

## Impaired ERE RNA editing in *Adar1^mZα/−* mice

The specific role of ZBP1 in the pathology caused by ADAR1 deficiency could be explained by the accumulation of Z-RNA ligands in the absence of ADAR1-dependent RNA editing. In support of this hypothesis, previous studies have shown that the Zα domain of ADAR1 is required for editing of RNAs derived from EREs and particularly short interspersed nuclear elements (SINEs) in mouse cells[6,8,30], which we previously identified as possible double-stranded RNA (dsRNA) ligands activating ZBP1 (ref. [21]). Analysis of spleen RNA-seq data showed that expression of ERE groups previously shown to have the highest number of complementary reads with the potential to generate dsRNA[21], including LTR/ERVK, LINE/L1, SINE/Alu and SINE/B2, was strongly induced in *Adar1^mZα/−* compared

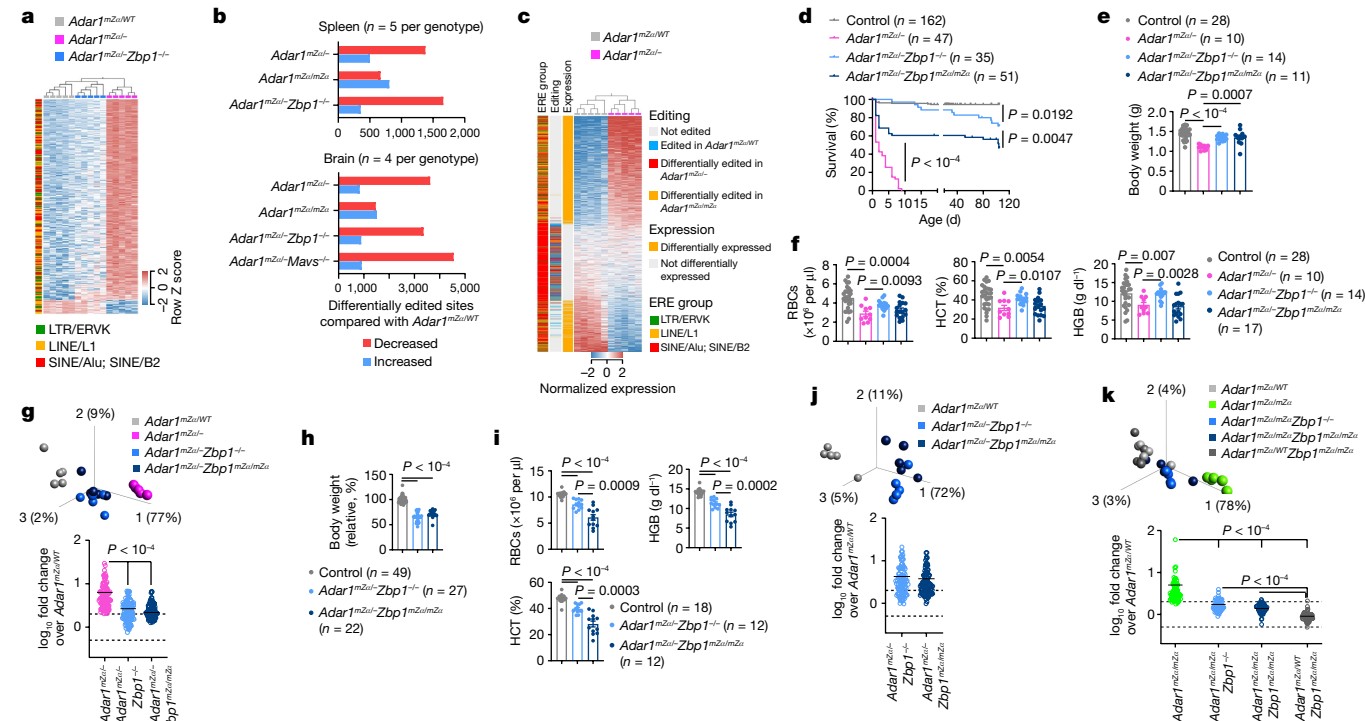

**Fig. 3 | Endogenous Z-RNA likely derived from EREs triggers ZBP1-dependent IFN responses in *Adar1^{mZα/−}* mice. a–c**, ERE expression and editing in spleen and brain tissues from mice with the indicated genotypes at P1. **a**, Heatmap depicting expression of EREs differentially expressed between *Adar1^{mZα/−}* and *Adar1^{mZα/WT}* spleen samples ($q ≤ 0.05$, ≥10-fold change). Only EREs belonging to groups previously linked to formation of dsRNA[21] were included and are ordered by principal component 1 and coloured according to their group. **b**, Number of differentially edited sites (>2-fold) across the indicated group comparisons. **c**, Heatmap depicting expression of EREs differentially expressed or differentially edited between *Adar1^{mZα/−}* and *Adar1^{mZα/WT}* mice. In **a** and **c**, columns represent individual samples, hierarchically clustered according to expression of the selected EREs. **d**, Kaplan–Meier survival graph of mice with the indicated genotypes. *P* values were calculated by two-sided Gehan–Breslow–Wilcoxon test. Survival data for *Adar1^{mZα/−}Zbp1^{−/−}* mice and their littermate controls from Fig. 1a are included for comparison. **e,f**, Body weight (**e**) and RBC counts and HGB and HCT levels in blood (**f**) for mice with the indicated genotypes at P1. Data for *Adar1^{mZα/−}Zbp1^{−/−}* mice and their littermate controls from Fig. 1b,c are included for comparison. **g**, Top, PCA on lung RNA-seq data from mice with the indicated genotypes at P1. PCA was based on the 93 ISGs upregulated in all three tissues of *Adar1^{mZα/−}* mice (Supplementary Table 2). Bottom, fold change (log₁₀) in expression of the same 93 ISGs between

mice of each genotype and control *Adar1^{mZα/WT}* mice. **h,i**, Relative body weight normalized to that of littermate controls at the age of 10 weeks (**h**) and RBC counts and HCT and HGB levels in blood at 15 weeks (**i**) for mice of the indicated genotypes. Data for *Adar1^{mZα/−}Zbp1^{−/−}* mice and their littermate controls from Fig. 2a,b are included for comparison. Control mice in **d–f**, **h** and **i** included littermates with the *Adar1^{mZα/WT}*, *Adar1^{WT/−}* or *Adar1^{WT/WT}* genotype. In **e**, **f**, **h** and **i**, dots represent individual mice, bar graphs show mean ± s.e.m. and *P* values were calculated by two-sided non-parametric Mann–Whitney test. **j**, Top, PCA on lung RNA-seq data from 15-week-old mice with the indicated genotypes. PCA was based on the 93 ISGs upregulated in all three tissues of *Adar1^{mZα/−}* mice (Supplementary Table 2). Bottom, fold change (log₁₀) in expression of the same 93 ISGs between mice of each genotype and control *Adar1^{mZα/WT}* mice. **k**, Top, PCA on lung RNA-seq data from 15-week-old mice with the indicated genotypes. PCA was based on the 57 ISGs upregulated in *Adar1^{mZα/mZα}* mice compared with controls (Supplementary Table 1). Bottom, fold change (log₁₀) in expression of the same 57 ISGs between mice of each genotype and control *Adar1^{mZα/WT}* mice. In **g**, **j** and **k**, symbols represent mean values of individual genes, solid lines show mean expression of the selected genes and dashed lines denote the twofold change boundaries. *P* values were calculated by two-sided non-parametric Mann–Whitney test (**j**) or Kruskal–Wallis test with Dunn's post hoc test for multiple comparisons (**g,k**). In **a**, **c**, **g**, **j** and **k**, $n = 5$.

with *Adar1^{mZα/WT}* pups (Fig. 3a). Interestingly, ZBP1 deficiency inhibited the upregulation of these ERE transcripts in *Adar1^{mZα/−}Zbp1^{−/−}* mice (Fig. 3a), reminiscent of its effect in suppressing ISG expression. These findings suggest that upregulation of EREs depends on IFN-mediated induction of gene expression, probably because these EREs reside near or within ISGs. We then assessed whether adenosine-to-inosine (A-to-I) RNA editing was affected in *Adar1^{mZα/−}* and *Adar1^{mZα/mZα}* compared with *Adar1^{mZα/WT}* mice. Analysis of RNA-seq data from the spleen and brain showed a considerable loss of edited sites in *Adar1^{mZα/−}* compared with *Adar1^{mZα/WT}* mice, the majority of which were found in SINE-derived RNAs (Fig. 3b,c and Extended Data Fig. 8). *Adar1^{mZα/mZα}* mice did not show an overall change in editing compared with *Adar1^{mZα/WT}* mice (Fig. 3b and Extended Data Fig. 8). Notably, ZBP1 deficiency did not rescue the impaired RNA editing in spleen and brain tissues of *Adar1^{mZα/−}* mice (Fig. 3b and Extended Data Fig. 8c), in contrast to its strong effect in normalizing the IFN response. Similarly, MAVS deficiency did not rescue the editing defect in *Adar1^{mZα/−}* mice (Fig. 3b). Together, these results

indicate that increased expression of ERE-derived transcripts together with overall diminished editing of repeat RNAs in *Adar1^{mZα/−}* mice could lead to the accumulation of dsRNAs with the capacity to generate Z-RNA ligands activating ZBP1.

## Zα domain-dependent role of ZBP1 in *Adar1^{mZα/−}* mice

To address the functional role of endogenous Z-RNA sensing by ZBP1, we crossed *Adar1^{mZα/−}* mice with *Zbp1^{mZα/mZα}* mice expressing ZBP1 with both its Zα domains mutated[21]. ZBP1 Zα domain mutation substantially, albeit partially, rescued the early postnatal lethality of *Adar1^{mZα/−}* mice, with about 50% of *Adar1^{mZα/−}Zbp1^{mZα/mZα}* mice surviving to the age of at least 15 weeks (Fig. 3d). Moreover, ZBP1 Zα domain mutation considerably restored body weight but did not substantially improve the impaired erythropoiesis in *Adar1^{mZα/−}* pups (Fig. 3e,f). Comparison of lung RNA-seq data from newborn pups showed that ZBP1 Zα domain mutation suppressed the ISG response of *Adar1^{mZα/−}* mice similarly to

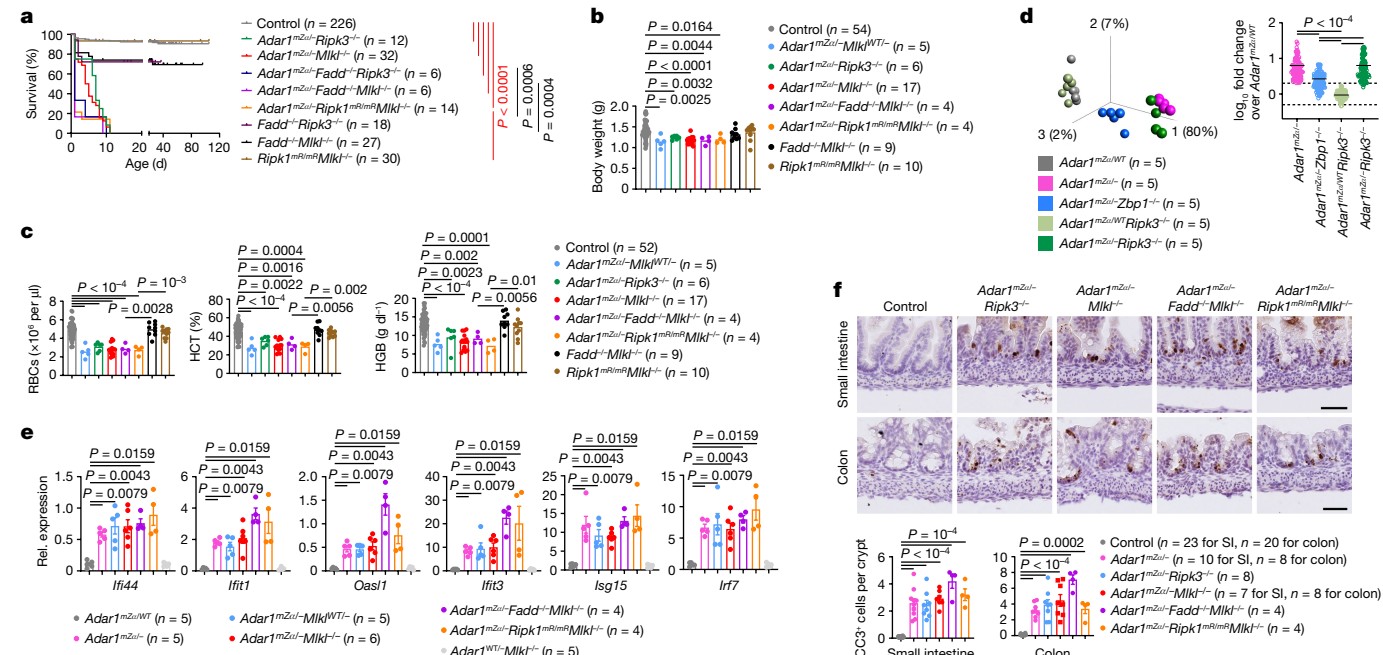

**Fig. 4 | ZBP1 promotes IFN induction and early postnatal lethality in *Adar1^mZα/−* mice independently of RIPK3–MLKL-induced necroptosis, FADD–caspase-8-dependent apoptosis and RHIM-dependent RIPK1 signalling. a**, Kaplan–Meier survival graph of mice with the indicated genotypes. *Fadd^−/−Ripk3^−/−*, *Fadd^−/−Mlkl^−/−* and *Ripk1^mR/mR^Mlkl^−/−* groups included mice with the *Adar1^mZα/WT^*, *Adar1^WT/−^* or *Adar1^WT/WT^* genotype. *P* values were calculated by two-sided Gehan–Breslow–Wilcoxon test. **b,c**, Body weight (**b**) and RBC counts and HGB and HCT levels in blood (**c**) for mice with the indicated genotypes at P1. **d**, Left, PCA on lung RNA-seq data from mice with the indicated genotypes at P1. PCA was based on the 93 ISGs upregulated in all three tissues of *Adar1^mZα/−^* mice (Supplementary Table 2). Right, fold change (log₁₀) in expression of the same 93 ISGs between mice of each genotype and control *Adar1^mZα/WT^* mice. Symbols represent mean values of individual genes,

solid lines show mean expression of the selected genes and dashed lines denote the twofold change boundaries. *P* values were calculated by Kruskal–Wallis test with Dunn's post hoc test for multiple comparisons. **e**, qRT–PCR analysis of mRNA expression of the indicated genes in lung tissues from P1 mice with the indicated genotypes. **f**, Representative images of small intestine and colon sections immunostained for CC3 and quantification of CC3⁺ cells in P1 mice with the indicated genotypes. CC3⁺ cell count data for *Adar1^mZα/−^* pups from Fig. 1d are included for comparison. Scale bars, 50 μm. In **b**, **c**, **e** and **f**, dots represent individual mice, bar graphs show mean ± s.e.m. and *P* values were calculated by two-sided non-parametric Mann–Whitney test. In **a–c** and **f**, control mice included littermates except for mice with the *Adar1^mZα/−^*, *Fadd^−/−Ripk3^−/−*, *Fadd^−/−Mlkl^−/−* and *Ripk1^mR/mR^Mlkl^−/−* genotypes.

ZBP1 deficiency, with *Adar1^mZα/−Zbp1^mZα/mZα* mice clustering together with *Adar1^mZα/−Zbp1^−/−* mice (Fig. 3g). Moreover, qRT–PCR analysis showed reduced expression of a set of ISGs in spleen, lung and liver tissues from *Adar1^mZα/−Zbp1^mZα/mZα* compared with *Adar1^mZα/−* mice at P1 (Extended Data Fig. 9a). Adult *Adar1^mZα/−Zbp1^mZα/mZα* mice appeared healthy but had reduced body weight and RBC, HGB and HCT levels compared with their littermate controls (Fig. 3h,i and Extended Data Fig. 9b). Histological analysis of tissues from 15-week-old mice showed signs of pericentral sinusoidal dilatation in the liver and mild hyperplasia with small numbers of dying cells in the intestine (Extended Data Fig. 9c). Furthermore, comparison of lung RNA-seq data from 15-week-old mice showed that ZBP1 Zα domain mutation suppressed ISG expression in *Adar1^mZα/−* mice to the same extent as ZBP1 deficiency (Fig. 3j). Together, these results showed that Zα domain-dependent sensing of endogenous ligands, presumably Z-RNA, activates ZBP1-dependent signalling, promoting IFN responses and causing the severe postnatally lethal phenotype of *Adar1^mZα/−* mice. However, Zα domain mutation conferred somewhat less protection in terms of mouse survival and could not substantially improve the impaired erythropoiesis of *Adar1^mZα/−* mice compared with ZBP1 deficiency (Fig. 3d,f,i), suggesting that ZBP1 also exerts Zα domain-independent functions, as shown previously in viral infection and inflammation models[21]. The finding that ZBP1 deficiency or disruption of its Zα domains suppressed upregulation of ISG expression argues that ZBP1 promotes the IFN response in *Adar1^mZα/−* mice. However, the reduction in ISG expression could also be secondary to rescue of tissue pathology. Therefore, to investigate whether ZBP1 regulates the IFN

response independently of tissue pathology, we assessed the effect of ZBP1 deficiency or Zα domain disruption in *Adar1^mZα/mZα* mice, which are healthy but have elevated ISG expression (Extended Data Fig. 1). Notably, ZBP1 deficiency or disruption of its Zα domains strongly, albeit incompletely, suppressed the expression of ISGs in *Adar1^mZα/mZα* mice (Fig. 3k and Extended Data Fig. 9d). Therefore, Zα domain-dependent ZBP1 signalling promotes the IFN response induced by disruption of the ADAR1 Zα domain independently of tissue damage.

## RIPK1- and RIPK3-independent role of ZBP1

We reasoned that ZBP1 might engage RIPK3-dependent signalling to cause the severe pathology of *Adar1^mZα/−* mice, as was the case in *Adar1^−/−Mavs^−/−* mice. Unexpectedly however, *Adar1^mZα/−Ripk3^−/−* pups displayed early postnatal lethality as well as reduced body weight and RBC, HGB and HCT values, showing that RIPK3 knockout did not mimic the effect of ZBP1 deficiency (Fig. 4a–c). Furthermore, RNA-seq and qRT–PCR gene expression analysis showed that, in contrast to ZBP1 deficiency, RIPK3 knockout did not suppress ISG expression in lung, spleen and liver tissues from *Adar1^mZα/−* mice (Fig. 4d and Extended Data Fig. 10). To further address the role of necroptosis, we generated *Adar1^mZα/−Mlkl^−/−* mice and found that MLKL deficiency also did not prevent early lethality and upregulation of ISG expression in lung tissues from *Adar1^mZα/−* mice (Fig. 4a–c,e). Thus, ZBP1 caused the pathology in *Adar1^mZα/−* mice independently of RIPK3–MLKL-dependent necroptosis. We then reasoned that FADD–caspase-8-mediated apoptosis could contribute to the ZBP1-dependent pathology. However,

*Adar1^mZα/–Fadd^–/–Mlkl^–/–* mice showed early postnatal lethality and impaired erythropoiesis as well as elevated ISG expression (Fig. 4a–c,e). In addition, combined deficiency in FADD and RIPK3 also did not rescue the early postnatal lethality of *Adar1^mZα/–* mice (Fig. 4a–c). Thus, combined inhibition of FADD–caspase-8-dependent apoptosis and RIPK3–MLKL-dependent necroptosis could not mimic the effect of ZBP1 deficiency, suggesting that ZBP1 induces pathology independently of necroptosis and FADD–caspase-8-dependent apoptosis. ZBP1 has also been implicated in inducing inflammatory gene expression by engaging RIPK1 through a RHIM-dependent interaction[39,40]. To address the role of RIPK1, we used *Ripk1^mR/mR* mice, which express RIPK1 with a mutated RHIM[20]. Because *Ripk1^mR/mR* mice die perinatally owing to MLKL-dependent necroptosis[20], we generated and analysed *Adar1^mZα/–Ripk1^mR/mRMlkl^–/–* mice. Strikingly, *Adar1^mZα/–Ripk1^mR/mRMlkl^–/–* mice also displayed early postnatal lethality and impaired erythropoiesis as well as elevated ISG expression similarly to *Adar1^mZα/–* pups (Fig. 4a–c,e), demonstrating that combined inhibition of necroptosis and RIPK1-dependent signalling could not mimic the effect of ZBP1 deficiency. Furthermore, immunohistological analysis of intestinal tissue sections showed that neither the *Ripk3^–/–*, *Mlkl^–/–* or *Fadd^–/–Mlkl^–/–* genetic background nor the *Ripk1^mR/mRMlkl^–/–* genetic background could prevent intestinal epithelial cell death in *Adar1^mZα/–* mice (Fig. 4f). Because MAVS deficiency completely prevented and ZBP1 knockout strongly inhibited cell death in the gut of newborn *Adar1^mZα/–* mice (Fig. 1d), these results suggest that the death of intestinal epithelial cells is a consequence of the IFN response and occurs independently of RIPK1, RIPK3–MLKL-mediated necroptosis and FADD–caspase-8-mediated apoptosis.

## Discussion

Our results showed that ZBP1 promoted type I IFN responses and the associated pathology in *Adar1^mZα/–* mice independently of RIPK1-mediated signalling, RIPK3–MLKL-dependent necroptosis and FADD–caspase-8-mediated apoptosis. These findings are in contrast to the function of ZBP1 in *Adar1^–/–Mavs^–/–* mice, where it causes early postnatal lethality by inducing RIPK3-dependent signalling (Fig. 2f). Therefore, ZBP1 has a dual role in mice with impaired ADAR1 function. On the one hand, it acts in a RIPK3-dependent manner to cause MAVS-independent pathology in *Adar1^–/–* mice, probably by inducing necroptosis. On the other hand, it acts in a RIPK1- and RIPK3-independent manner to promote a MAVS-dependent pathogenic type I IFN response, causing early postnatal lethality in *Adar1^mZα/–* mice. MAVS deficiency nearly completely normalized whereas ZBP1 deficiency partially rescued ISG expression and the pathology of *Adar1^mZα/–* mice, suggesting that ZBP1 is induced downstream of MAVS and contributes to type I IFN activation and the associated pathologies. The mechanisms by which ZBP1 promotes type I IFN activation in *Adar1^mZα/–* mice remain elusive at present. TIR domain-containing adaptor-inducing interferon-β (TRIF), which induces IFN responses downstream of the Toll-like receptors TLR3 and TLR4 (ref. [41]), also contains a RHIM and could be implicated in driving IFN activation downstream of ZBP1. ZBP1 was recently reported to contribute to TRIF-induced caspase-8 activation and interleukin (IL)-1β release[42]. However, this specific function was mediated via RHIM-dependent interaction with RIPK1 and therefore should be inhibited in a *Ripk1^mR/mRMlkl^–/–* genetic background, which did not rescue type I IFN activation in *Adar1^mZα/–* mice, arguing against the involvement of this particular signalling pathway. It is also possible that RIPK1, RIPK3 and TRIF contribute to ZBP1-dependent IFN activation in *Adar1^mZα/–* mice in a functionally redundant manner; if this is the case, inactivation of all three proteins may be required to mimic the effect of ZBP1 deficiency. Notably, in contrast to its previously suggested role as a cytosolic DNA sensor inducing IFN activation[43], ZBP1 was not required for the cGAS–STING-dependent IFN response in *Trex1^–/–* mice,

suggesting that it specifically functions to augment RNA-induced MDA5–MAVS-dependent IFN responses in *Adar1^mZα/–* mice.

Taken together, our results identified Zα domain-dependent cross-talk between ADAR1 and ZBP1 that critically controls IFN responses to endogenous RNA. Our findings identify suppression of endogenous Z-RNA formation by ADAR1 as a key mechanism preventing aberrant activation of pathogenic IFN responses by ZBP1. Although it remains technically challenging to directly assess Z-RNA formation in living cells and tissues, our results suggest that impaired ADAR1-dependent editing of RNAs primarily derived from SINEs causes the accumulation of Z-RNA ligands activating ZBP1. Strikingly, our genetic studies showed that ZBP1 induced pathogenic IFN responses in *Adar1^mZα/–* mice in a manner independent of RIPK1, RIPK3–MLKL-dependent necroptosis and FADD–caspase-8-dependent apoptosis, suggesting a new mechanism of action. Collectively, while the specific downstream molecular mechanisms remain to be elucidated, our results identified ZBP1 as a key driver of pathogenic type I IFN responses triggered by impaired ADAR1 function and suggest that ZBP1-dependent signalling could contribute to the pathogenesis of type I interferonopathies caused by *ADAR1* mutations in humans.

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

## Methods

### Mice

Zbp1[−/−] (ref. [21]), Zbp1[mZα/mZα] (ref. [21]) and Ripk3[−/−] (ref. [44]) mice have been described previously. Adar1[−/−] (Adar[tm1b(EUCOMM)Wtsi]) mice were generated from the EUCOMM (https://academic.oup.com/bfg/article/6/3/180/237263) line Adar[tm1a(EUCOMM)Wtsi] using CMV:Cre deleter mice[45]. Adar1[mZα] mice, in which amino acids N175 and Y179 of the ADAR1 Zα domain were mutated to aspartic acid and alanine, respectively, were generated using CRISPR–Cas12a technology with all components purchased from Integrated DNA Technologies. Cas12a guide RNA (4 µM; 5′-CAGGGAGTACAAAATACGATTGA-3′; AsCas12a crRNA) targeting exon 2 of the Adar1 gene and 10 µM single-stranded DNA repair oligonucleotide (5′-A*G*G*TTTCCCCCTTCCTCTGTGC AGCTTTCCCTTCTTcTCCAGGGAagcCAAAATACGgTcGATGTCCCTTT TGGGGATTCTGAGCTCTCTGGCTAGCACATGGGCAG*T*G*G-3′; IDT, custom-made ultramer) with three phosphorothioate bonds at both ends (indicated by an asterisk) were co-electroporated essentially as described previously[46] with 4 µM AsCas12a protein and 4 µM DNA-based Cas12a electroporation enhancer into C57BL/6N zygotes. Correct exchange of the nucleotides, represented in the repair oligonucleotide with lower-case letters, was assessed by Sanger sequencing in the resulting $F_0$ mice. Trex1[−/−] and Mavs[−/−] mice were generated using CRISPR–Cas9 technology. Of note, Trex1 is a single-exon gene. For Trex1[−/−] mice, two sgRNAs (5′-TTCCAGGTCTAAGAAGATGA-3′ and 5′-CCTGGGCAGTAAGTCAAGAG-3′), each at 4 µM, in complex with 4 µM Cas9 protein (IDT) were co-electroporated into fertilized oocytes. Deletion of the Trex1 exon between the two sgRNAs was confirmed using primers 5′-ATCCCACTAGAACAACCCTGCC-3′ and 5′-TTCAGACTCCGCACCCTCATTT-3′ as well as by immunoblot analysis. For Mavs[−/−] mice, two sgRNAs (5′-CCGGTTCCCGATCTGCCTGT-3′ and 5′-ATACTGTGACCCCAGACAAG-3′) targeting exons 3 and 6, respectively, were co-injected into fertilized C57BL/6N oocytes at 50 ng µl[−1] together with 100 ng µl[−1] Cas9 mRNA (Trilink). Successful deletion of the critical exons was confirmed using PCR primers 5′-TTGATCC TCACACCGTACTTG-3′ and 5′-GTATTGTGTTGGCAGGTGCTT-3′. Mice used in this study were maintained in the animal facility of the CECAD Research Center, University of Cologne, in individually ventilated cages (Tecniplast, Greenline GM500) at 22 °C (±2 °C) and a relative humidity of 55% (±5%) under a 12-h light/12-h dark cycle on sterilized bedding (Aspen wood, Abedd) with access to a sterilized commercial pelleted diet (Ssniff Spezialdiäten) and acidified water ad libitum. The microbiological status of the mice was examined as recommended by the Federation of European Laboratory Animal Science Associations (FELASA), and the mice were free of all listed pathogens. All animal procedures were conducted in accordance with European, national and institutional guidelines, and protocols were approved by local government authorities (Landesamt für Natur, Umwelt und Verbraucherschutz Nordrhein-Westfalen). Animals requiring medical attention were provided with appropriate care and were killed when they reached predetermined criteria of disease severity. No other exclusion criteria existed. Experimental groups were not randomized as mice were assigned to groups on the basis of genotype. Sample size was estimated on the basis of previous experience. Female and male mice of the indicated genotypes were assigned to groups at random. Mice were analysed at the age stated in the respective figure legends. Mouse studies as well as immunohistochemical assessment of pathology were performed in a blinded fashion. Whole blood samples of the mice were analysed using Abacus Junior Vet (Diatron).

### Immunoblotting

Protein extracts from organs were prepared using a beadmill (Precellys 24) in RIPA buffer supplemented with cOmplete protease inhibitor cocktail (Roche, 04693124001) and phosSTOP phosphatase inhibitor tablets (Roche, 4906837001) and denatured in 2× Laemmli buffer (Bio-Rad, 34095) supplemented with 5% β-mercaptoethanol. Cell lysates were prepared by direct cell lysis in RIPA buffer or IP buffer (20 mM Tris pH 7.4, 150 mM NaCl, 2 mM EDTA, 1% Triton X-100), supplemented with protease and phosphatase inhibitors, followed by denaturation in Laemmli buffer. Lysates were separated by SDS–PAGE, transferred to Immobilon-P PVDF membranes (Millipore, 05317) and analysed by immunoblotting with primary antibodies against ISG15 (Cell Signaling, 2743), p-STAT1 (Cell Signaling, 9167), STAT1 (Cell Signaling, 9172), ZBP1 (Adipogen, AG-20B-0010, or custom made), TREX1 (Santa Cruz, sc-133112), GAPDH (Novus Biologicals, NB300-221), ADAR1 (Santa Cruz, sc-73408), MLKL (Millipore, MABC604), p-MLKL (Cell Signaling, 37333), caspase-8 (Cell Signaling, 4790, or Alexis, ALX-804-447), cleaved caspase-8 (Cell Signaling, 8592) and α-tubulin (Sigma, T6074). Secondary horseradish peroxidase (HRP)-coupled antibodies against rat (Jackson ImmunoResearch, 112-035-003), rabbit (Amersham Pharmacia, NA934V) or mouse (Amersham Pharmacia, NA931V) were used to detect proteins using chemiluminescence with ECL SuperSignal West PicoPlus chemiluminescent substrate (Thermo Scientific, 34578), and signal was measured with a Fusion Solo X system (Vilber).

### Cell death assays

Primary MEFs were maintained at 37 °C and 5% $CO_2$ in DMEM (ThermoFisher, 41965-039) supplemented with 10% FCS (Biosell), 1% penicillin-streptomycin (ThermoFisher Scientific, 15140130), 1% L-glutamine (ThermoFisher Scientific, 25030-123) and 1 mM sodium pyruvate (ThermoFisher Scientific, 11360). For cell death assays, cells were seeded in 96-well plates at a density of $1 \times 10^4$ cells per well. The next day, cells were stimulated with $10^3$ U ml[−1] IFNγ (ImmunoTools, 12343537) for 48 h or 24 h followed by treatment with combinations of 1 µM CHX (Sigma, 239763), 10 µM Q-VD-OPh (R&D, OPH001) and 3 µM GSK'872 (Sigma, 5303890001) in the presence of 0.1 µM DRAQ7 (Biostatus, DR71000). Cells were imaged for the indicated duration of time in 2-h intervals using the IncuCyte S3 live-cell imaging and analysis platform (Essen BioScience) in bright-field and red fluorescence (emission, 635 nm; excitation, 585 nm) mode. DRAQ7-positive cells were automatically counted as dead cells in two to four images per well, and counts were averaged using the Incucyte software package version 2019B rev2.

### Immunohistochemistry and histology

Tissue samples from mice were fixed in 4% paraformaldehyde (PFA) and embedded in paraffin. Sections of 5 µm were subjected to histological analysis by haematoxylin and eosin (H&E) staining or Masson's Trichrome staining. For immunohistochemical analysis, slides were rehydrated and incubated with peroxidase blocking buffer (0.04 M sodium citrate, 0.121 M $Na_2HPO_2$, 0.03 M $NaN_3$, 3% $H_2O_2$) for 15 min. Slides were washed and antigen retrieval was performed by digestion with 10 µg ml[−1] proteinase K (Life Technologies, 25530031) for 5 min in TEX buffer (50 mM Tris, 1 mM EDTA, 0.5% Triton X-100, pH 8.0) for CD45 and F4/80 staining or in citrate Tris buffer (pH 6) in a pressure cooker for CD3 and CC3 staining. Sections were blocked for 60 min and incubated with the primary antibody for CD45 (clone 30 F-11; eBioscience, 14-0451), CD3 (clone CD3-12; Bio-Rad, MCA1477), CC3 (clone D5B2; Cell signaling, 9661) or F4/80 (clone A3-1; AbD Serotec, MCA497) at 4 °C overnight. Sections were incubated with biotinylated anti-rat secondary antibody (Jackson ImmunoResearch, 112-065-003) or anti-rabbit secondary antibody (Invitrogen, B2770) for 60 min. Staining was visualized using the Vectastain Elite ABC-HRP kit (Vector Laboratories, VEC-PK-6100) and DAB substrate (Dako Omnis, GV82511-2, or Vector Laboratories, SK-4100). Sections were then counterstained with haematoxylin for staining of the nuclei, dehydrated and mounted with Entellan. Histological sections were scanned using a Nanozoomer S360 (Hamamatsu) slide scanner. Images were analysed and processed using the Omero software package (https://openmicroscopy.org) and NDP. view2 Viewing software (Hamamatsu).

Histological analysis of brains was conducted following standardized protocols at the Institute of Neuropathology, University Medical Center of Freiburg. Sections (4 μm thick) were processed and stained with H&E by standard laboratory procedures.

Immunohistochemical labelling was performed with an Autostainer Link 48 (Agilent), according to the manufacturer's instructions. Primary antibodies against IBA1 (clone EPR16588; Abcam, ab178846) and MAC3 (clone M3/84; BD Biosciences, 550292) for (activated) macrophages and microglia, B220 for B cells (clone RA3-6B2; BD Biosciences, 553084), CD3 for T cells (clone CD3-12; Bio-Rad, MCA1477) and the corresponding secondary antibodies goat anti-rabbit IgG (H+L) (Southern Biotech, 4050-08; for IBA1) and goat anti-rat IgG (H+L) (Southern Biotech, 3050-08; for MAC3, B220 and CD3) were used.

### Histopathological analysis

Histopathological evaluation of intestinal tissues was performed on 3-μm-thick H&E-stained sections of paraffin-embedded Swiss rolls of intestinal tissues, using a modified version of a previously described scoring system[47]. In brief, histopathology scores were composed of four parameters: epithelial hyperplasia, quantity and localization of tissue inflammation, epithelial cell death and epithelial injury. An 'area factor' for the fraction of affected tissue was assigned and multiplied by the respective parameter score (1, 0–25%; 2, 25–50%; 3, 50–75%; 4, 75–100%). If different severities for the same parameter were observed in the same sample, each area was judged individually and multiplied by the corresponding area factor. Area factors for a given sample always added up to 4. The histology score was calculated as the sum of all parameter scores multiplied by their area factors. The maximum score was 64. Scores and ulcer quantification were based on one Swiss roll section per mouse and were determined in a blinded fashion. Quantification of CC3+ cells was performed on histological sections immunostained with antibodies against CC3. The total number of CC3+ cells was divided by the number of crypts to show the average number in one crypt. Two hundred crypts for small intestine and at least 74 crypts for colon were analysed per mouse. Counting was performed in a blinded fashion.

Histological analysis of kidneys was performed by applying standardized protocols at the Institute of Surgical Pathology, University Medical Center of Freiburg. In brief, 2-μm microtome sections of formalin-fixed, paraffin-embedded tissue were used for PAS reaction staining with standardized diagnostic procedures. Whole kidney slides (WSI) were digitalized using a Ventana DP 200 slide scanner (Roche Diagnostics Deutschland) equipped with a ×40 objective. The MS index was assessed by applying a four-tiered scoring system (0–3; 0, <5%; 1, 6–25%; 2, 26–50%; 3, >50%). Further quantitative analysis of tuft areas and cells (nuclei) per tuft area was performed using QuPath v0.3.2 image analysis software[48,49]. At least 50 glomerular tufts per mouse were manually segmented by random sampling. Nucleus segmentation of individual tufts was performed by applying the built-in nucleus segmentation tool (QuPath). Histopathological evaluation and quantitative analysis were performed in a blinded fashion by an expert renal pathologist. For analysis of histology, an inverted Zeiss Axio Imager microscope equipped with an Axiocam colour camera, ×10, ×40 and ×100 objectives and a Ventana DP 200 slide scanner was used.

### Gene expression analysis by qRT–PCR

Total RNA was extracted with either TRIzol reagent (Life Technologies, 15596018) and chloroform or the NucleoSpin RNA kit (Macherey-Nagel, 740955.50), according to the manufacturer's instructions, followed by cDNA synthesis using the SuperScript III First-Strand Synthesis System (Life Technologies, 18080051) with subsequent treatment with RNase H (Invitrogen, AM2293) or the LunaScript RT SuperMix kit (New England Biolabs, E3010L). qRT–PCR was performed using TaqMan probes and TaqMan Gene Expression Master Mix (ThermoScientific, 4369016) or Luna Universal Probe qPCR Master Mix (New England Biolabs, M3004X)

in a QuantStudio 5 Real-Time PCR System (ABI). Reactions were run in technical duplicates with *Tbp* as a reference gene. Relative expression of gene transcripts is shown using the $2^{-\Delta Ct}$ method and is represented in dot plot graphs as mRNA expression values relative to the reference gene. The TaqMan probes used were as follows: *Tbp* (Mm00446973_m1), *Tnf* (Mm00443258_m1), *Il1b* (Mm00434228_m1), *Il6* (Mm00446190_m1), *Nppb* (Mm01255770_g1), *Col3a1* (Mm01254476_m1), *Irf7* (Mm00516793_g1), *Isg15* (Mm01705338_s1), *Ifi44* (Mm00505670_m1), *Oasl1* (Mm00455081_m1), *Mx1* (Mm00487796_m1), *Ccl2* (Mm00441242_m1), *Cxcl10* (Mm00445235_m1), *Ifit1* (Mm00515153_m1) and *Zbp1* (Mm00457979_m1, Mm00457981_m1).

### RNA-seq and data processing

RNA was prepared with the NucleoSpin RNA Mini kit for RNA purification (Macherey-Nagel, 740955.50). rRNA was depleted with rRNA Removal Mix–Gold. For lung RNA, library preparation was performed using the QuantSeq 3′ mRNA-Seq Library Prep Kit FWD for Illumina (Lexogen). QuantSeq libraries were sequenced on an Illumina NovaSeq 6000 sequencer using Illumina RTA v3.4.4 base-calling software. For spleen and brain RNA, stranded Illumina sequencing libraries were prepared with the TruSeq Stranded Total RNA kit (Illumina, 20020599), according to the manufacturer's instructions, from rRNA-depleted RNA samples and submitted for PE100 sequencing using an Illumina NovaSeq 6000 sequencer, yielding ~50 million reads per sample. The quality of the resulting data was assessed using FastQC v0.11.8 (https://www.bioinformatics.babraham.ac.uk/projects/fastqc/), and reads were subsequently quality and adaptor trimmed using cutadapt (v3.4)[50] with stringent settings to remove error-containing reads ('-q 20 --max-n 0 --max-ee 1'). Remaining reads were passed to HISAT2 (v2.1.0)[51] for strand-aware alignment, and strand-specific counts of uniquely mapping reads were prepared using featureCounts (within Subread v1.6.4; ref. [52]) against Ensembl GRCh38.100 annotations. Additional unstranded counts were obtained with featureCounts against a database of repetitive elements previously prepared for GRCh38 (ref. [53]) using reads unassigned to features during the previous step.

### Differential expression analyses

DESeq2 (v1.22.1)[54] within R was used for read count normalization, and downstream differential expression analysis and visualization were performed within Qlucore Omics Explorer v3.3 (Qlucore). Repeat region annotation, RNA-seq read mapping and counting were carried out as previously described[21].

### Gene functional annotation

Pathway analyses were performed using g:Profiler (https://biit.cs.ut. ee/gprofiler) with genes ordered by the degree of differential expression. *P* values were estimated by hypergeometric distribution tests and adjusted by multiple-testing correction using the g:SCS (set counts and sizes) algorithm, integral to the g:Profiler server[55]. ISGs were defined according to the Interferome v2.01 database[56].

### Detection and analysis of A-to-I editing

Read alignments were processed with samtools markdup[57] to identify likely PCR duplicates within the sequenced libraries, and A-to-I editing was assessed using JACUSA2 (ref. [58]) with settings to flag and exclude from analysis potential editing sites in close proximity to the start and end of reads, indel positions and splice sites, as well as sites within homopolymer runs of more than seven bases, using only primary alignments of properly paired, non-duplicate reads. Detected and differential A>G editing sites were filtered for *Z* score > 1.96, a depth of ≥10 reads, a minimum editing fraction of ≥1%, ≥2 replicates to display editing and exclude potential SNPs, and a maximum editing fraction of <50%. Additionally, for differential sites, a >2-fold difference was required between test and control sample groups. Sites obtained were assigned to genomic features using annotatr v1.16 within R[59]. Assessments of editing enrichment within repetitive elements were

conducted with regioneR v1.22 within R[60] using randomization-based permutation tests with 100 bootstraps. Graphs were generated with GraphPad Prism v9.

## Statistical analysis

Data shown in graphs are the mean or mean ± s.e.m. A non-parametric Mann–Whitney test, one-way ANOVA with Tukey's multiple-comparisons test or Kruskal–Wallis test corrected by Dunn's multiple-comparisons test was performed. Survival curves were compared using the Gehan–Breslow–Wilcoxon test. All statistical analyses were performed with GraphPad Prism.

## Reporting summary

Further information on research design is available in the Nature Research Reporting Summary linked to this paper.

## Data availability

RNA-seq data have been deposited in the ArrayExpress database at EMBL-EBI under accession numbers E-MTAB-10953, E-MTAB-11537 and E-MTAB-11540. Source data are provided with this paper.

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

**Acknowledgements** We thank E. Gareus, M. Hahn, J. Kuth, L. Elles, P. Roggan, E. Stade, C. Uthoff-Hachenberg, J. von Rhein, T. Wagner, K. Gräwe and A. Sammarco for technical assistance. We acknowledge support from the Cologne Center for Genomics for RNA-seq and the CECAD imaging facility for microscopy. We also thank A. Athanasiadis and A. Herbert for valuable discussions and V. Dixit (Genentech) for *Ripk3*[−/−] mice. Research reported in this publication was supported by funding from the European Research Council (ERC) under the European Union's Horizon 2020 research and innovation programme (grant agreement no. 787826 to M. Pasparakis), the Deutsche Forschungsgemeinschaft (DFG, German Research Foundation, projects SFB1403 (project no. 414786233), SFB1399 (project No. 413326622), and CECAD (project no. 390661388) to M. Pasparakis, projects SCHE 2092/3-1, SCHE 2092/4-1 (RP9, CP2, CP3), CRU329 and SFB1453 (project No. 43198400) to C.S. and project no. 413517907 to M.D.), the Swiss National Science Foundation (SNSF, project no. 199310) to M.D. and the Francis Crick Institute (FC001099) to G.K. M.N. was supported by a Japan Society for the Promotion of Science Overseas Research Fellowship (grant no. 201860571) and a postdoctoral fellowship from the Alexander von Humboldt Foundation.

**Author contributions** H.J., L.W., N.O. and M. Pasparakis conceived the study and designed the experiments. H.J. analysed *Adar1*[mZα/−] and *Adar1*[−/−] mice and conducted in vitro biochemical analyses. L.W. analysed *Adar1*[mZα/mZα] mice, performed and analysed qRT–PCR and cell death assays, and prepared RNA samples for RNA-seq. S.W. analysed *Trex1*[−/−] mice and performed CC3 staining in tissues from *Adar1*[mZα/−] mice. J.L. performed qRT–PCR assays, prepared RNA samples for RNA-seq and tissue samples for histology, and performed western blot and cell death experiments. M.N. performed western blot experiments, cell death assays and general histopathology analysis, as well as quantification of CC3[+] cells in colon and small intestinal samples. G.G.K. analysed the histopathology of gut samples and quantified CC3[+] cells in small intestinal samples. N.O. performed FACS analysis, analysed *Trex1*[−/−] mice and prepared tissue samples. V.K. analysed liver histopathology. M.R. and C.S. analysed kidney histopathology. M.D. and M. Prinz analysed brain histopathology. G.R.Y. and G.K. analysed RNA-seq data for gene and ERE expression and A-to-I RNA editing. L.W., N.O., S.E.T. and B.Z. designed and generated *Adar1*[mZα/mZα] and *Trex1*[−/−] mice. C.S., M. Prinz, G.K. and M. Pasparakis supervised the experiments. H.J., L.W., G.K. and M. Pasparakis interpreted data and wrote the paper.

**Funding** Open access funding provided by Universität zu Köln.

**Competing interests** The authors declare no competing interests.

**Additional information**
**Correspondence and requests for materials** should be addressed to Manolis Pasparakis.

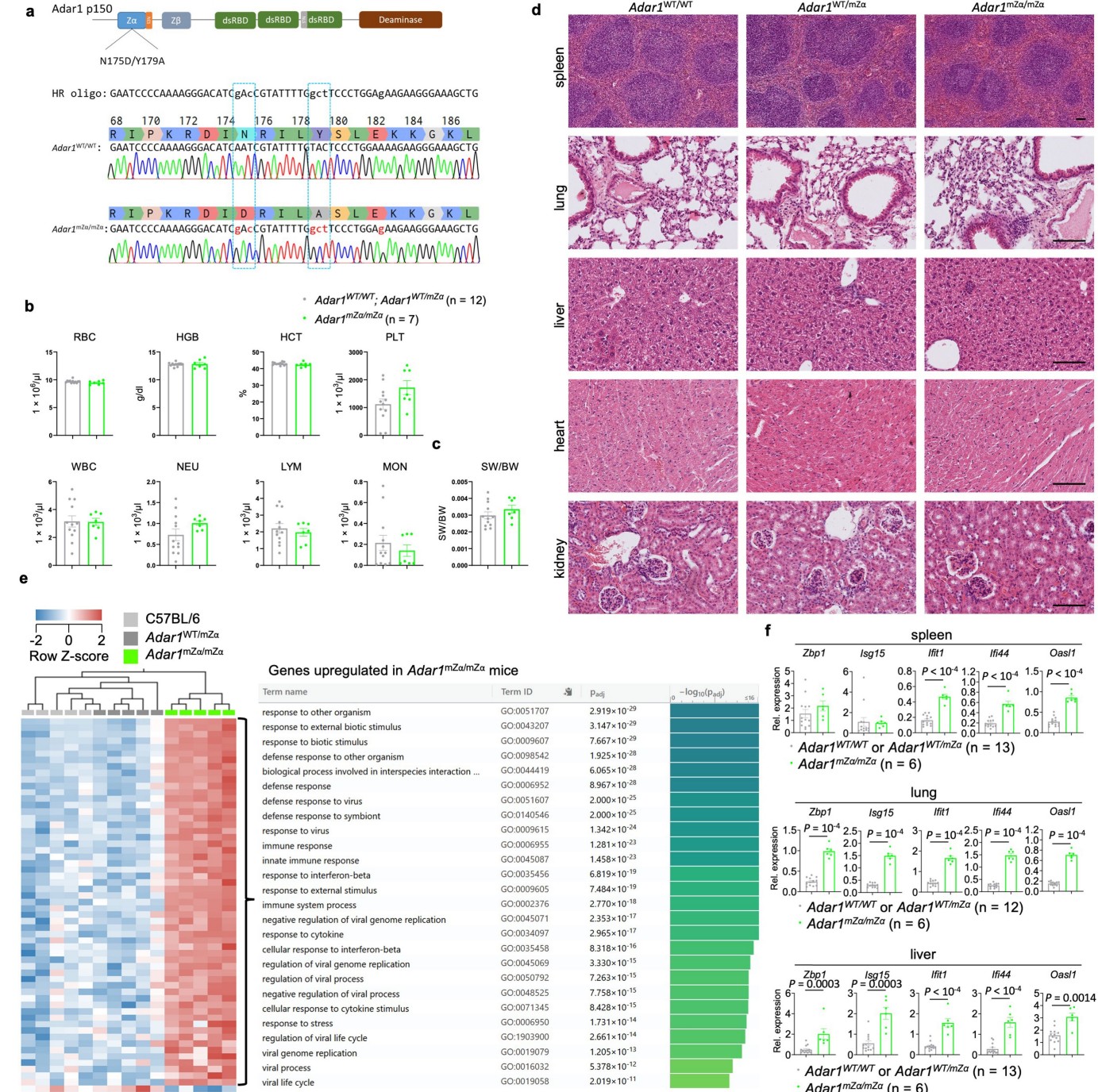

**Extended Data Fig. 1 | Generation and characterization of *Adar1*^mZα/mZα mice. a**, Schematic depicting the generation of the *Adar1*^mZα allele using CRISPR-Cas12a-mediated gene targeting. The indicated nucleotide substitutions were introduced in exon 1 of the *Adar1* gene to substitute amino acids N175 with D and Y179 with A in the Zα domain. Sequencing traces of the desired mutations are shown in homozygous mice. **b**, RBC, platelet (PLT), white blood cell (WBC), neutrophil (NEU), monocyte (MON), lymphocyte (LYM) counts and HGB and HCT levels in the blood of 10-to-13-month-old mice with the indicated genotypes. **c**, Spleen-to-body weight ratio of mice with the indicated genotypes. **d**, Representative H&E-stained sections from spleen, lung, liver, heart and kidney of mice with the indicated genotypes. *Adar1*^mZα/mZα (n = 7), *Adar1*^WT/mZα (n = 10), *Adar1*^WT/WT (n = 3). Scale bar, 100 μm. **e**, Transcriptional comparison of lung tissues from *Adar1*^mZα/mZα, *Adar1*^WT/mZα and wild type C57BL/6N mice (n = 5 for all genotypes) at 4-5 months of age. The heatmap includes 58 genes differentially expressed between *Adar1*^mZα/mZα mice and the two control groups combined (p ≤ 0.05, q ≤ 0.05, ≥ 2-fold-change), with

57 of these upregulated in the former (Supplementary Data Table 1). Columns represent individual mice, hierarchically clustered according to differential gene expression. Table shows gene ontology (GO) functional annotation of the 57 genes upregulated in *Adar1*^mZα/mZα mice, performed with g:Profiler (https://biit.cs.ut.ee/gprofiler). *P* values for differential expression analyses were calculated with Qlucore Omics Explorer using two-sided *t*-tests and with the q value for false discovery rates (FDR) set to 0.05. Calculation of q values was adjusted for multiple hypothesis testing using the Benjamini-Hochberg method. *P* values for pathway analyses were calculated with g:Profiler using hypergeometric distribution tests and adjusted for multiple hypothesis testing using the g:SCS (set counts and sizes) algorithm, integral to the g:Profiler server (https://biit.cs.ut.ee/gprofiler). **f**, qRT-PCR analysis of mRNA expression of the indicated genes in indicates tissues from 10-to-13-month-old mice with the indicated genotypes. In **b, c** and **f**, dots represent individual mice, bar graphs show mean ± s.e.m and *P* values were calculated by two-sided nonparametric Mann-Whitney test.

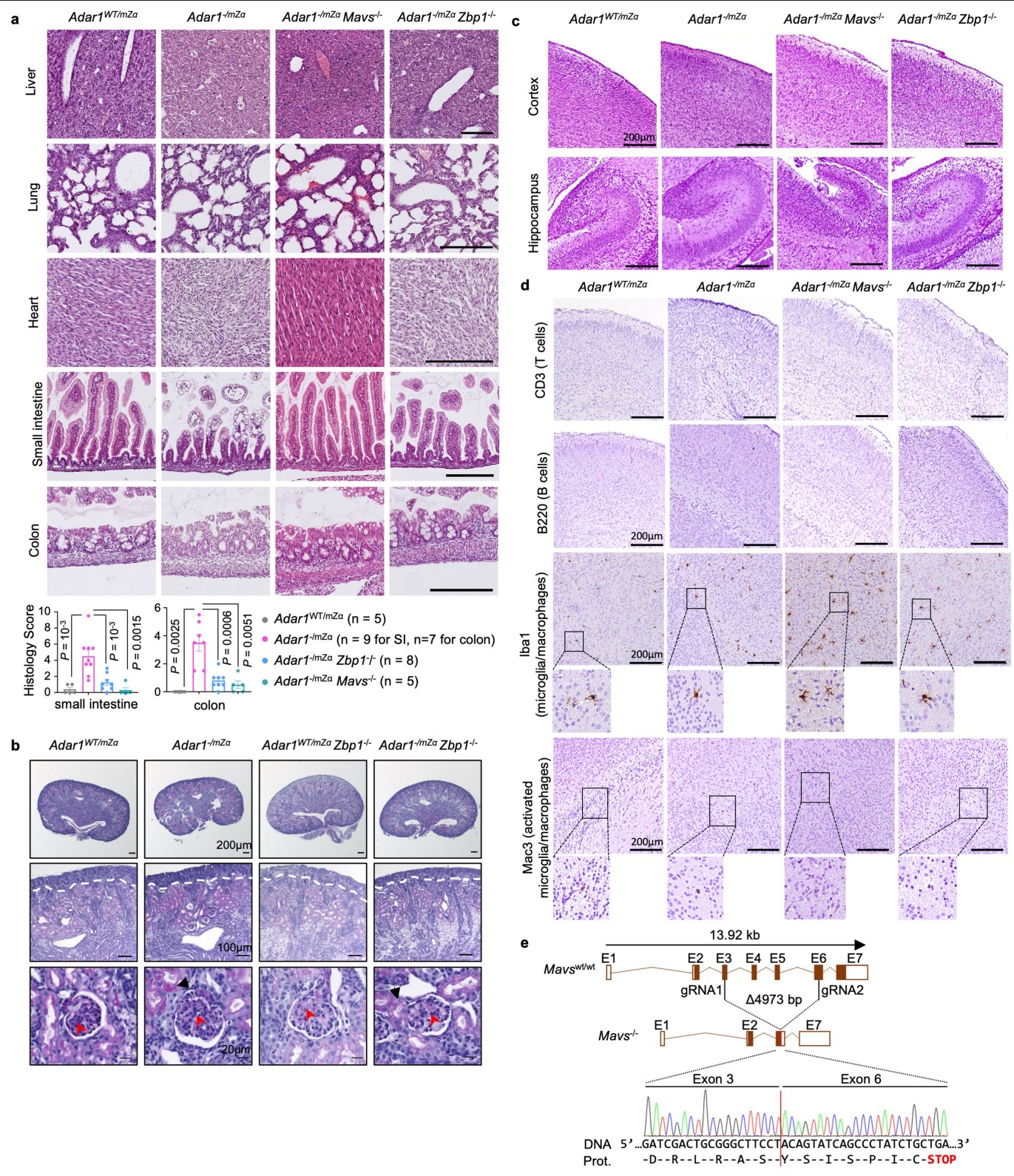

**Extended Data Fig. 2** | See next page for caption.

**Extended Data Fig. 2 | Histological analysis of tissues from *Adar1*<sup>WT/mZα</sup>, *Adar1*<sup>−/mZα</sup>, *Adar1*<sup>−/mZα</sup> *Mavs*<sup>−/−</sup> and *Adar1*<sup>−/mZα</sup> *Zbp1*<sup>−/−</sup> mice. a**, Representative H&E-stained sections from liver, lung, heart, small intestine and colon from *Adar1*<sup>WT/mZα</sup> (n = 5), *Adar1*<sup>−/mZα</sup> (n = 7), *Adar1*<sup>−/mZα</sup> *Mavs*<sup>−/−</sup> (n = 5) and *Adar1*<sup>−/mZα</sup> *Zbp1*<sup>−/−</sup> (n = 8) mice at P1. Scale bar, 200 μm. Graphs depict histological ileitis and colitis scores. Dots represent individual mice, bar graphs show mean ± s.e.m and *P* values were calculated by two-sided nonparametric Mann-Whitney test. **b**, Representative images of PAS-stained kidney sections from *Adar1*<sup>WT/mZα</sup> (n = 6), *Adar1*<sup>−/mZα</sup> (n = 6), *Adar1*<sup>WT/mZα</sup> *Zbp1*<sup>−/−</sup> (n = 6) and *Adar1*<sup>−/mZα</sup> *Zbp1*<sup>−/−</sup> (n = 6) mice at P1. Scale bar, 200 μm (top), 100 μm (middle) or 20 μm (bottom). Mice of all genotypes presented with normal zonal architecture of postnatal kidneys, in particular the cortical nephrogenic zone (white dotted line) appeared unaffected. Black arrows indicate representative protein absorption droplets in proximal tubuli; red arrowheads highlight intact mesangial compartments of respective early mature glomeruli. **c**–**d**, Representative images of brain sections from *Adar1*<sup>WT/mZα</sup> (n = 8), *Adar1*<sup>−/mZα</sup> (n = 11), *Adar1*<sup>−/mZα</sup> *Mavs*<sup>−/−</sup> (n = 5) and *Adar1*<sup>−/mZα</sup> *Zbp1*<sup>−/−</sup> (n = 6) mice at P1 stained with H&E **(c)** or immunolabelled for CD3, B220, IBA1 or MAC3 **(d)**. Scale bar, 200 μm. **e**, Schematic depicting the generation of *Mavs*<sup>−/−</sup> mice using CRISPR-Cas9-mediated gene targeting. Two gRNAs were used to cut in exon 3 and exon 6 resulting in deletion of the respective sequence. DNA sequencing in homozygous mice confirmed the resulting fusion between exons 3 and 6 and the generation of a premature stop codon.

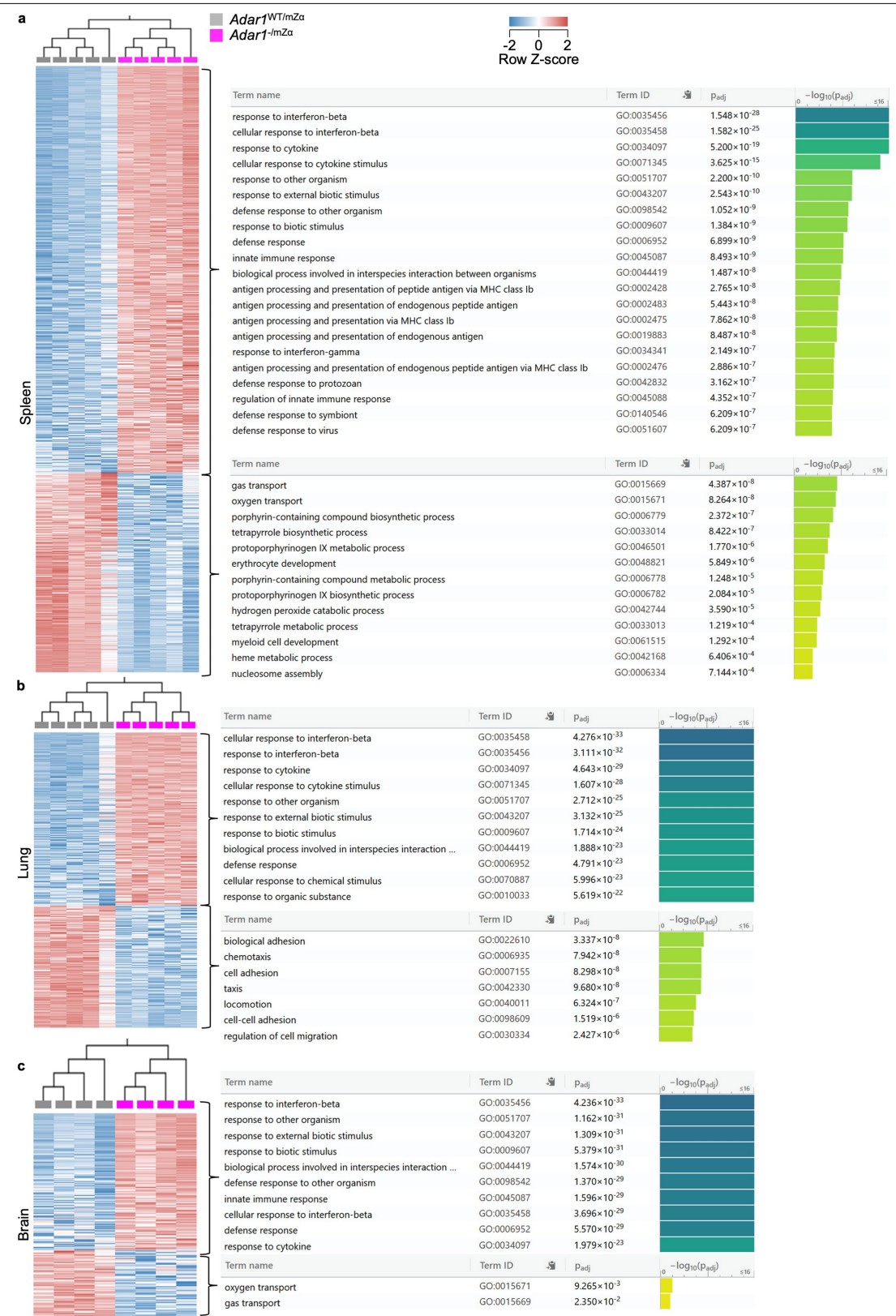

**Extended Data Fig. 3** | See next page for caption.

**Extended Data Fig. 3 | Comparison of RNAseq expression profiles in spleen, lung and brain tissues from *Adar1*<sup>−/mZα</sup> and *Adar1*<sup>WT/mZα</sup> mice.** **a**–**c**, Transcriptional comparison of spleen (**a**), lung (**b**) and brain (**c**) tissues from *Adar1*<sup>−/mZα</sup> and *Adar1*<sup>WT/mZα</sup> littermate mice at P1. Heatmaps show differentially expressed genes between *Adar1*<sup>−/mZα</sup> and *Adar1*<sup>WT/mZα</sup> control mice. These were 1,594 ($p \le 0.05$, $q \le 0.05$, $\ge$ 2-fold-change), 657 ($p \le 0.05$, $q \le 0.05$, $\ge$ 2-fold-change), and 379 ($p \le 0.05$, $\ge$ 2-fold-change) genes for the spleen, lung and brain, respectively (Supplementary Data Table 2). Columns represent individual mice, hierarchically clustered according to differential gene expression. Tables show GO functional annotation of the genes found upregulated and downregulated in the comparison between the two genotypes in each organ, performed with g:Profiler (https://biit.cs.ut.ee/gprofiler). *P* values for differential expression analyses were calculated with Qlucore Omics Explorer using two-sided *t*-tests and with the q value for false discovery rates (FDR) set to 0.05. Calculation of q values was adjusted for multiple hypothesis testing using the Benjamini-Hochberg method. *P* values for pathway analyses were calculated with g:Profiler using hypergeometric distribution tests and adjusted for multiple hypothesis testing using the g:SCS (set counts and sizes) algorithm, integral to the g:Profiler server (https://biit.cs.ut.ee/gprofiler). For all genotypes, n = 5 for spleen and lung, n = 4 for brain.

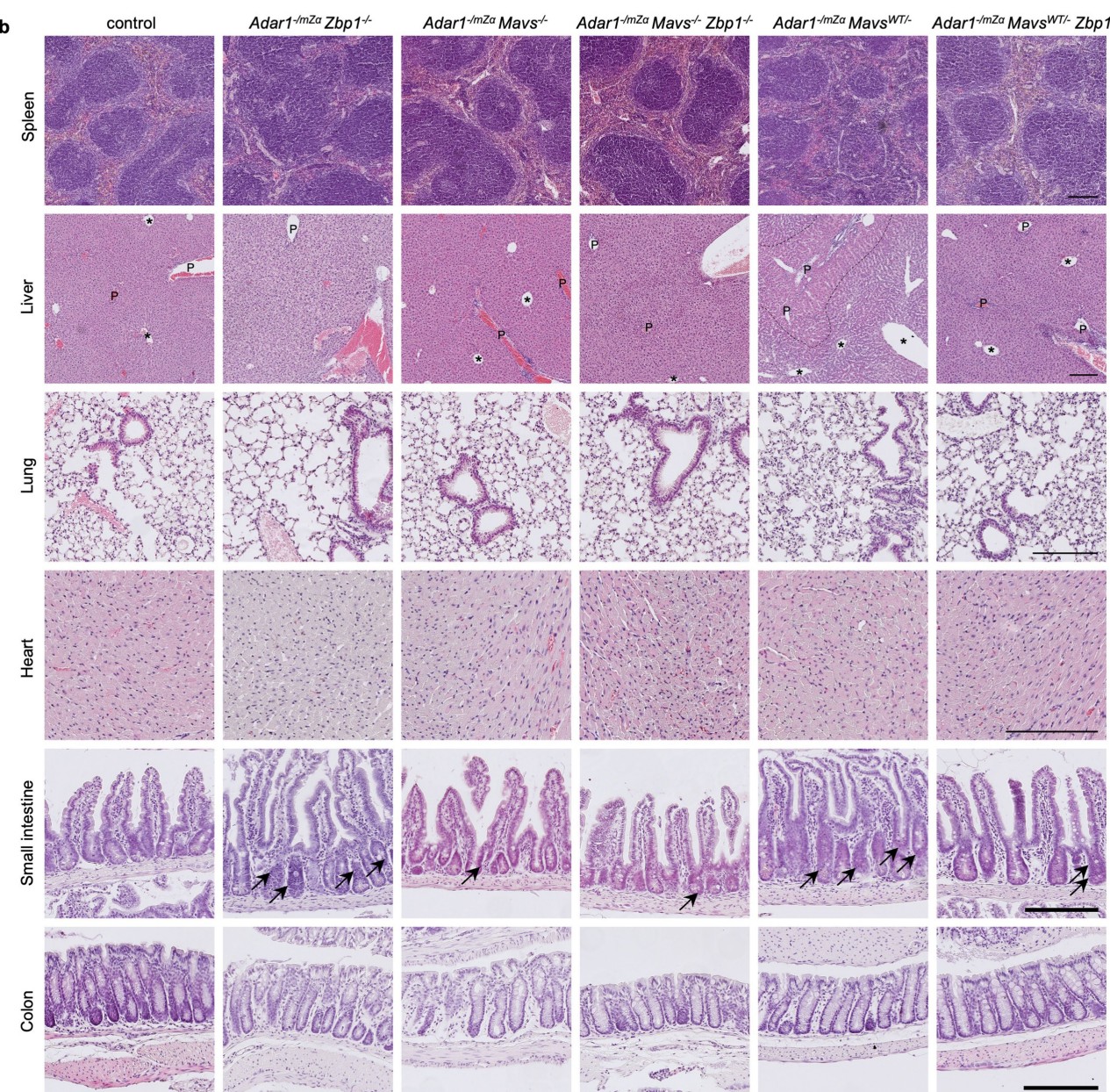

**Extended Data Fig. 4 | Macroscopic and histological analysis of adult** ***Adar1*<sup>−/mZα</sup> mice deficient in ZBP1 or MAVS. a**, Representative images of mice with the indicated genotypes at about 15 weeks of age. **b**, Representative H&E-stained sections from indicated tissues of 15-week-old mice with the indicated genotypes. Scale bars, 200 μm. Periportal and pericentral areas in liver sections are marked with P and *, respectively. Pericentral sinusoidal

dilatation in *Adar1*<sup>−/mZα</sup> *Mavs*<sup>WT/−</sup> liver section is outlined with dashed lines (top left and bottom right areas). Black arrows indicate dying epithelial cells in small intestinal sections. In **a, b**, control mice include littermates with *Adar1*<sup>WT/mZα</sup>, *Adar1*<sup>WT/−</sup> or *Adar1*<sup>WT/WT</sup> genotypes. Control (n = 7), *Adar1*<sup>−/mZα</sup> *Zbp1*<sup>−/−</sup> (n = 11), *Adar1*<sup>−/mZα</sup> *Mavs*<sup>WT/−</sup> (n = 8), *Adar1*<sup>−/mZα</sup> *Zbp1*<sup>−/−</sup> *Mavs*<sup>WT/−</sup> (n = 9), *Adar1*<sup>−/mZα</sup> *Mavs*<sup>−/−</sup> (n = 9), *Adar1*<sup>−/mZα</sup> *Zbp1*<sup>−/−</sup> *Mavs*<sup>−/−</sup> (n = 7). Scale bar, 200 μm.

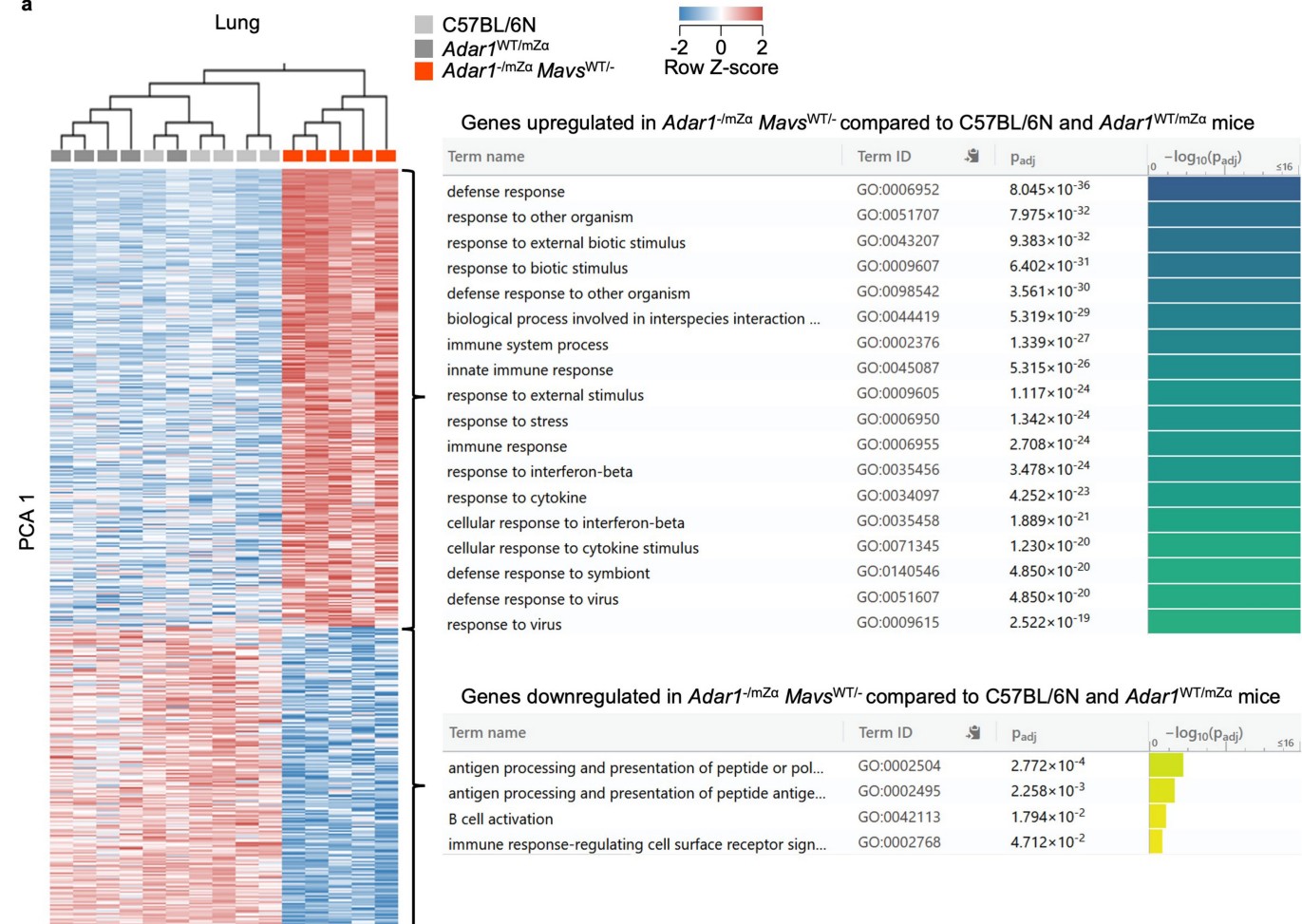

**a**

Lung

Genes upregulated in *Adar1*$^{-/m Z\alpha}$ *Mavs*$^{WT/-}$ compared to C57BL/6N and *Adar1*$^{WT/m Z\alpha}$ mice

| Term name | Term ID | $p_{adj}$ | $-\log_{10}(p_{adj})$ |
|---|---|---|---|
| defense response | GO:0006952 | $8.045 \times 10^{-36}$ | |
| response to other organism | GO:0051707 | $7.975 \times 10^{-32}$ | |
| response to external biotic stimulus | GO:0043207 | $9.383 \times 10^{-32}$ | |
| response to biotic stimulus | GO:0009607 | $6.402 \times 10^{-31}$ | |
| defense response to other organism | GO:0098542 | $3.561 \times 10^{-30}$ | |
| biological process involved in interspecies interaction ... | GO:0044419 | $5.319 \times 10^{-29}$ | |
| immune system process | GO:0002376 | $1.339 \times 10^{-27}$ | |
| innate immune response | GO:0045087 | $5.315 \times 10^{-26}$ | |
| response to external stimulus | GO:0009605 | $1.117 \times 10^{-24}$ | |
| response to stress | GO:0006950 | $1.342 \times 10^{-24}$ | |
| immune response | GO:0006955 | $2.708 \times 10^{-24}$ | |
| response to interferon-beta | GO:0035456 | $3.478 \times 10^{-24}$ | |
| response to cytokine | GO:0034097 | $4.252 \times 10^{-23}$ | |
| cellular response to interferon-beta | GO:0035458 | $1.889 \times 10^{-21}$ | |
| cellular response to cytokine stimulus | GO:0071345 | $1.230 \times 10^{-20}$ | |
| defense response to symbiont | GO:0140546 | $4.850 \times 10^{-20}$ | |
| defense response to virus | GO:0051607 | $4.850 \times 10^{-20}$ | |
| response to virus | GO:0009615 | $2.522 \times 10^{-19}$ | |

Genes downregulated in *Adar1*$^{-/m Z\alpha}$ *Mavs*$^{WT/-}$ compared to C57BL/6N and *Adar1*$^{WT/m Z\alpha}$ mice

| Term name | Term ID | $p_{adj}$ | $-\log_{10}(p_{adj})$ |
|---|---|---|---|
| antigen processing and presentation of peptide or pol... | GO:0002504 | $2.772 \times 10^{-4}$ | |
| antigen processing and presentation of peptide antige... | GO:0002495 | $2.258 \times 10^{-3}$ | |
| B cell activation | GO:0042113 | $1.794 \times 10^{-2}$ | |
| immune response-regulating cell surface receptor sign... | GO:0002768 | $4.712 \times 10^{-2}$ | |

**Extended Data Fig. 5 | Comparison of RNA-seq expression profiles in lung tissues from *Adar1*$^{-/m Z\alpha}$ *Mavs*$^{WT/-}$ and control mice.** Comparison of lung RNAseq data from *Adar1*$^{-/m Z\alpha}$ *Mavs*$^{WT/-}$ (n = 5), *Adar1*$^{WT/m Z\alpha}$ (n = 5) and wild type C57BL/6N mice (n = 5) at 15 weeks of age. Heatmap shows 678 genes differentially expressed between *Adar1*$^{-/m Z\alpha}$ *Mavs*$^{WT/-}$ mice and the two control groups combined (p ≤ 0.05, q ≤ 0.05, ≥ 2-fold-change), with 399 of these upregulated in the former (Supplementary Data Table 3). Columns represent individual mice, hierarchically clustered according to differential gene expression. Tables show GO functional annotation of the genes found upregulated and downregulated in this comparison, performed with g:Profiler (https://biit.cs.ut.ee/gprofiler). *P* values for differential expression analyses were calculated with Qlucore Omics Explorer using two-sided *t*-tests and with the q value for false discovery rates (FDR) set to 0.05. Calculation of q values was adjusted for multiple hypothesis testing using the Benjamini-Hochberg method. *P* values for pathway analyses were calculated with g:Profiler using hypergeometric distribution tests and adjusted for multiple hypothesis testing using the g:SCS (set counts and sizes) algorithm, integral to the g:Profiler server (https://biit.cs.ut.ee/gprofiler).

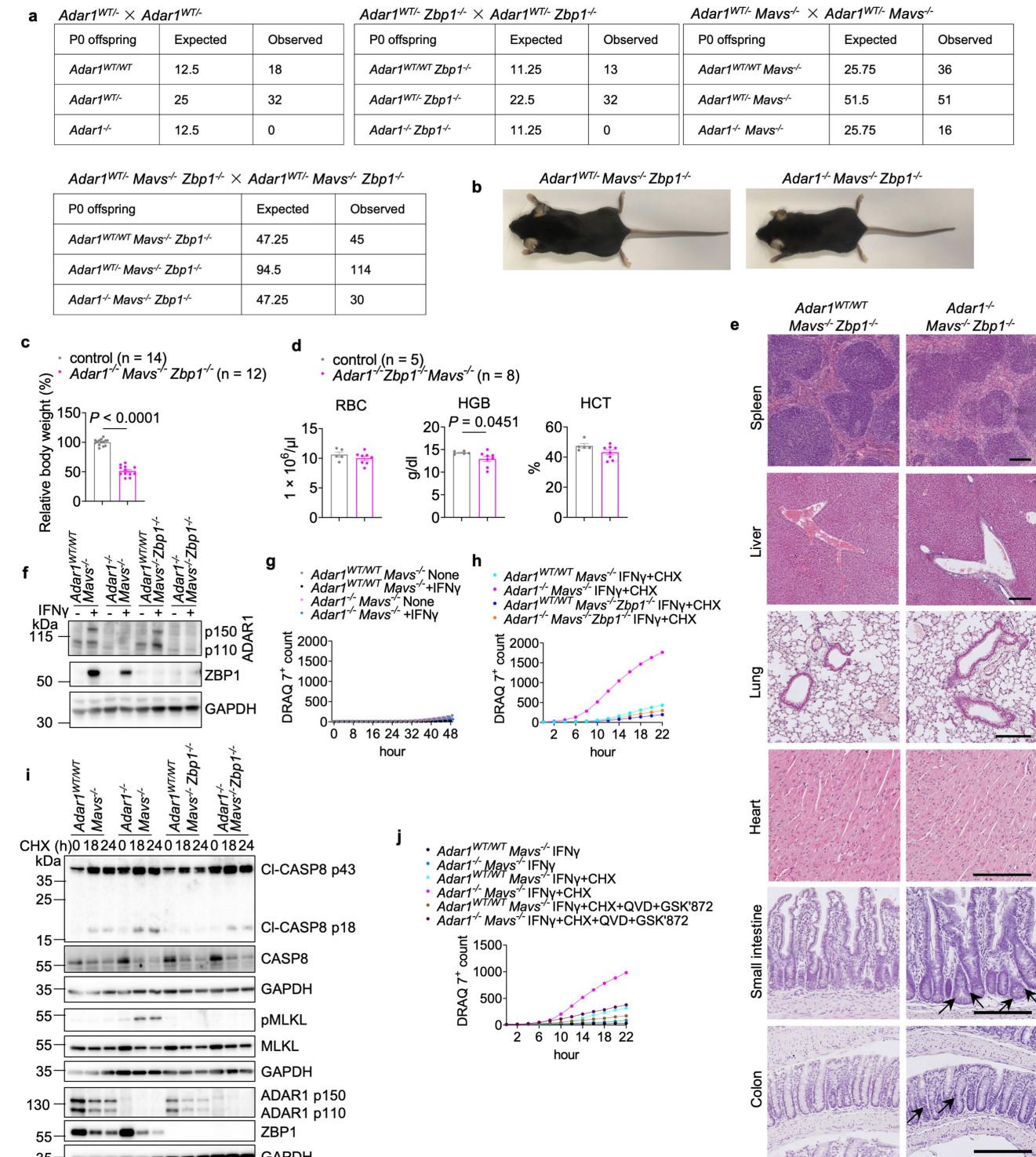

**Extended Data Fig. 6** | See next page for caption.

**Extended Data Fig. 6 | ZBP1 contributes to the early postnatal lethality of _Adar1_−/− _Mavs_−/− mice and induces cell death in MAVS-deficient MEFs.** **a**, Numbers of offspring genotyped at birth from intercrossing _Adar1_WT/−, _Adar1_WT/− _Zbp1_−/−, _Adar1_WT/− _Mavs_−/− or _Adar1_WT/− _Zbp1_−/− _Mavs_−/− mice. **b**, Representative images of _Adar1_WT/− _Mavs_−/− _Zbp1_−/− (n = 6) and _Adar1_−/− _Mavs_−/− _Zbp1_−/− (n = 8) mice at about 15 weeks of age. **c**, Relative body weight compared to littermate controls in 10-week-old mice with the indicated genotypes. **d**, RBC counts, HGB and HCT levels in blood of 15–20 week-old mice with the indicated genotypes. In **c**, **d**, control mice include littermates with _Adar1_WT/− or _Adar1_WT/WT genotypes. Dots represent individual mice, bar graphs show mean ± s.e.m and _P_ values were calculated by two-sided nonparametric Mann-Whitney test. **e**, Representative H&E-stained sections from the spleen, liver, lung, heart, small intestine and colon of 15–20 week-old _Adar1_WT/WT _Mavs_−/− _Zbp1_−/− (n = 5) and _Adar1_−/− _Mavs_−/− _Zbp1_−/− (n = 5) mice. Black arrows indicate dying epithelial cells in small intestine and colon sections. Scale bars, 200 μm. **f**, Immunoblot analysis of total lysates from murine embryonic fibroblasts (MEFs) of the indicated genotypes, stimulated with IFNγ (1,000 U ml−1) for 24 h. One representative out of two independent experiments is shown. GAPDH was used as a loading control. **g, h, j**, Cell death measured by DRAQ7 uptake in MEFs of the indicated genotypes treated with IFNγ (1,000 U ml−1) alone or with combinations of IFNγ (1,000 U ml−1) (24-h pretreatment), cycloheximide (CHX) (1 μg ml−1), QVD-OPh (QVD) (10 μM) and GSK'872 (3 μM). Graph shows mean values from technical triplicates (n = 3), from one representative out of three independent experiments. **i**, Immunoblot analysis of total lysates from MEFs of the indicated genotypes pre-stimulated with IFNγ (1,000 U ml−1) for 24 h followed by treatment with CHX (1 μg ml−1) for the indicated time points. Data are representative of two independent experiments. GAPDH was used as loading control. For gel source data, see Supplementary Figure 1.

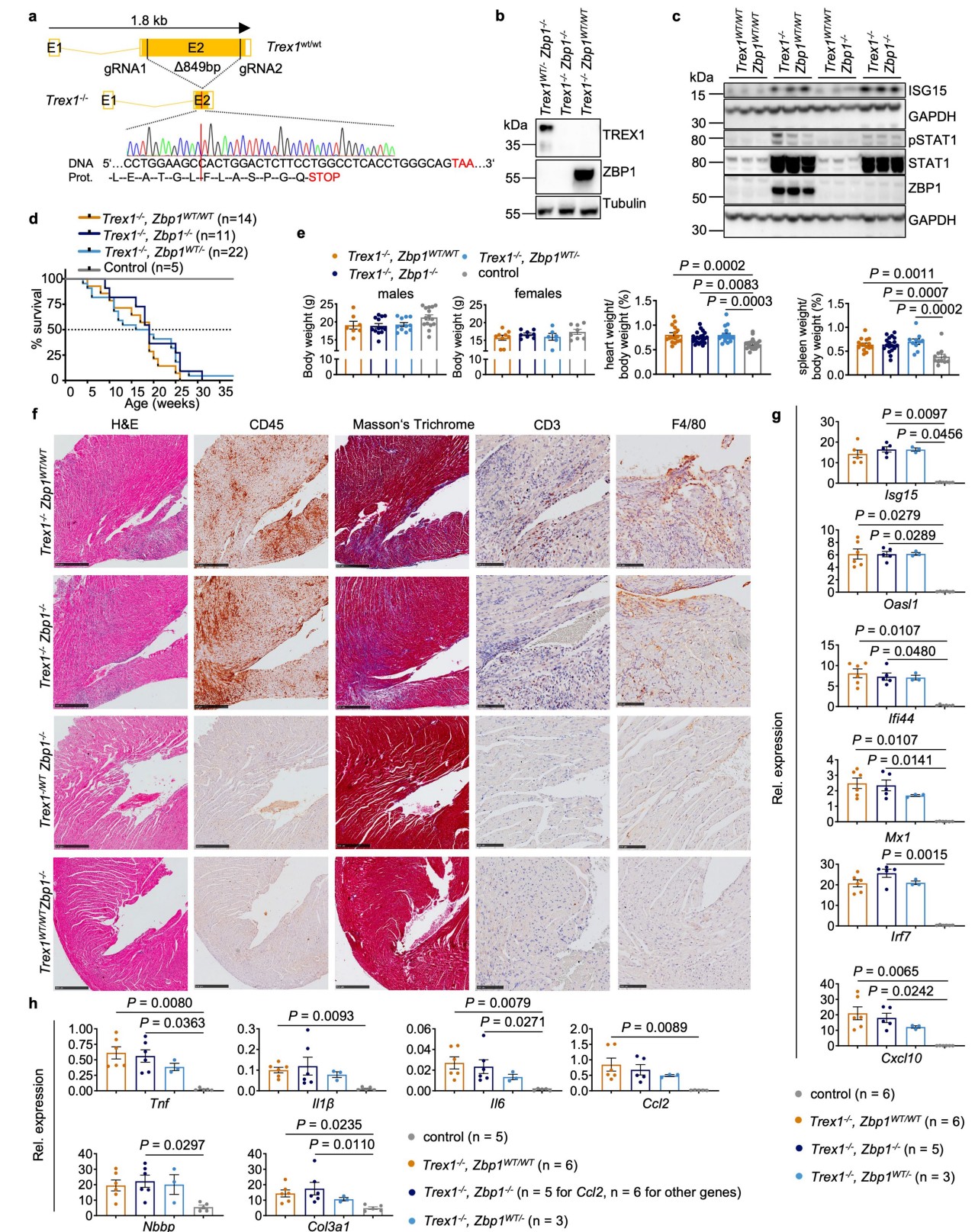

**Extended Data Fig. 7** | See next page for caption.

**Extended Data Fig. 7 | ZBP1 deficiency does not rescue the interferon response and pathology of *Trex1*$^{-/-}$ mice. a**, Schematic depicting the generation of *Trex1*$^{-/-}$ mice using CRISPR-Cas9-mediated gene targeting. Two gRNAs were used to delete the single coding exon of the *Trex1* gene. DNA sequencing in homozygous mice confirmed the deletion. **b, c**, Immunoblot analysis of spleen (**b**) or heart (**c**) protein lysates from 8–10-week-old mice of the indicated genotypes with the indicated antibodies. α-tubulin and GAPDH were used as loading control. Lanes represent individual mice. One representative of two independent experiments is shown. **d**, Kaplan-Meier survival plot of mice of the indicated genotypes. Control mice include *Trex1*$^{WT/-}$ *Zbp1*$^{-/-}$ and *Trex1*$^{WT/WT}$ *Zbp1*$^{-/-}$ mice. **e**, Graph depicting body weight of 8–10-week-old mice of the indicated genotypes. Males: *Trex1*$^{-/-}$ *Zbp1*$^{WT/WT}$ (n = 7), *Trex1*$^{-/-}$ *Zbp1*$^{-/-}$ (n = 13), *Trex1*$^{-/-}$ *Zbp1*$^{WT/-}$ (n = 11), control (n = 15). Females: *Trex1*$^{-/-}$ *Zbp1*$^{WT/WT}$ (n = 9), *Trex1*$^{-/-}$ *Zbp1*$^{-/-}$ (n = 8), *Trex1*$^{-/-}$ *Zbp1*$^{WT/-}$ (n = 6), control (n = 8). Heart weight relative to body weight: *Trex1*$^{-/-}$ *Zbp1*$^{WT/WT}$ (n = 16), *Trex1*$^{-/-}$ *Zbp1*$^{-/-}$ (n = 21), *Trex1*$^{-/-}$ *Zbp1*$^{WT/-}$ (n = 17), control (n = 22). Spleen weight relative to body weight: *Trex1*$^{-/-}$ *Zbp1*$^{WT/WT}$ (n = 15), *Trex1*$^{-/-}$ *Zbp1*$^{-/-}$ (n = 18), *Trex1*$^{-/-}$ *Zbp1*$^{WT/-}$ (n = 10), control (n = 11). Control mice include *Trex1*$^{WT/-}$ *Zbp1*$^{-/-}$ and *Trex1*$^{WT/WT}$ *Zbp1*$^{-/-}$ mice. Dots represent individual mice, bar graphs show mean ± s.e.m and *P* values were calculated by one-way ANOVA test with Tukey's correction for multiple comparison. **f**, Representative images of heart sections of 8–10-week-old *Trex1*$^{-/-}$ *Zbp1*$^{WT/WT}$ (n = 7), *Trex1*$^{-/-}$ *Zbp1*$^{-/-}$ (n = 8), *Trex1*$^{WT/-}$ *Zbp1*$^{-/-}$ (n = 5) and *Trex1*$^{WT/WT}$ *Zbp1*$^{-/-}$ (n = 5) mice stained with H&E or Masson's Trichrome (collagen: blue, cytoplasm: red) or immunostained for CD45, CD3 or F4/80. Scale bars: 500 µm for H&E, CD45 and Masson's Trichrome staining; 100 µm for CD3 and F4/80 staining. **g, h**, qRT-PCR analysis of mRNA expression of the indicated genes in heart tissues of 8–10-week-old mice of the indicated genotypes. Dots represent individual mice, bar graphs show mean ± s.e.m and *P* values were calculated by a Kruskal-Wallis test with Dunn's multiple comparison. For gel source data, see Supplementary Figure 1.

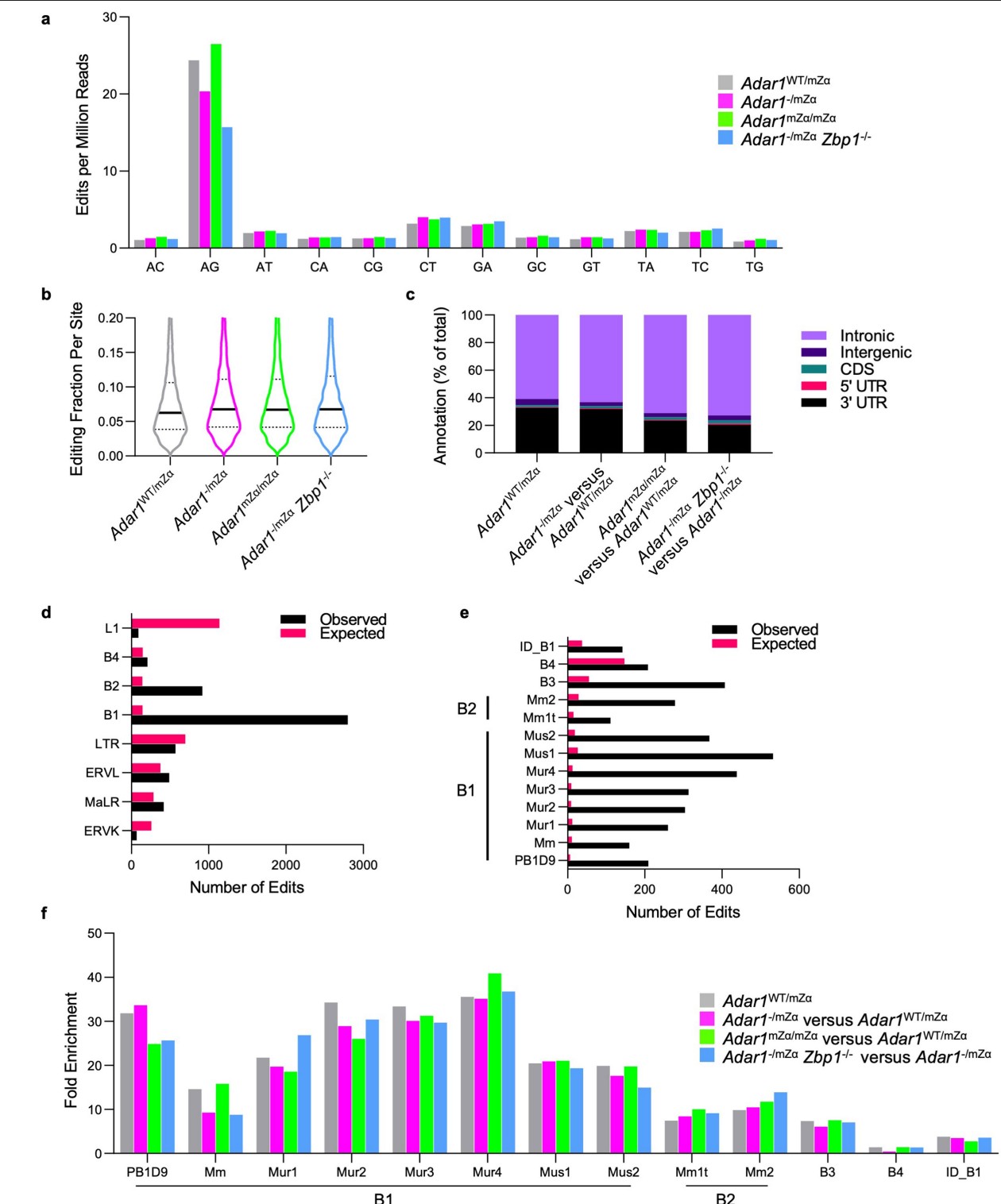

**Extended Data Fig. 8 | RNA editing analysis in spleen RNA from** *Adar1*^WT/mZα^, *Adar1*^mZα/mZα^, *Adar1*^-/mZα^ **and** *Adar1*^-/mZα^ *Zbp1*^-/-^ **mice. a**, Normalised number of edits passing the statistical filters described in the Methods section, split according to the base substitution recorded, showing enrichment of the A > G editing profile associated with deamination of adenine. **b**, Violin plots of the per-site editing percentage for sites with detected editing. Solid horizontal lines show the median and dotted lines indicate quartiles. **c**, Edited sites detected in *Adar1*^WT/mZα^ mice, for reference, and the differential edits determined for the indicated comparisons were matched to annotated genomic features. The percentage of sites is shown for each category. **d**, Number of expected and observed editing sites in *Adar1*^WT/mZα^ mice are shown for families of repetitive elements for which either value exceeded 100. **e**, Number of expected and observed editing sites in *Adar1*^WT/mZα^ mice are shown for individual SINE families for which either value exceeded 100. **f**, Fold enrichment (observed/expected) for the SINE families shown in (**e**), showing detected edits within *Adar1*^WT/mZα^ mice, for reference, and differential edits within the indicated comparisons.

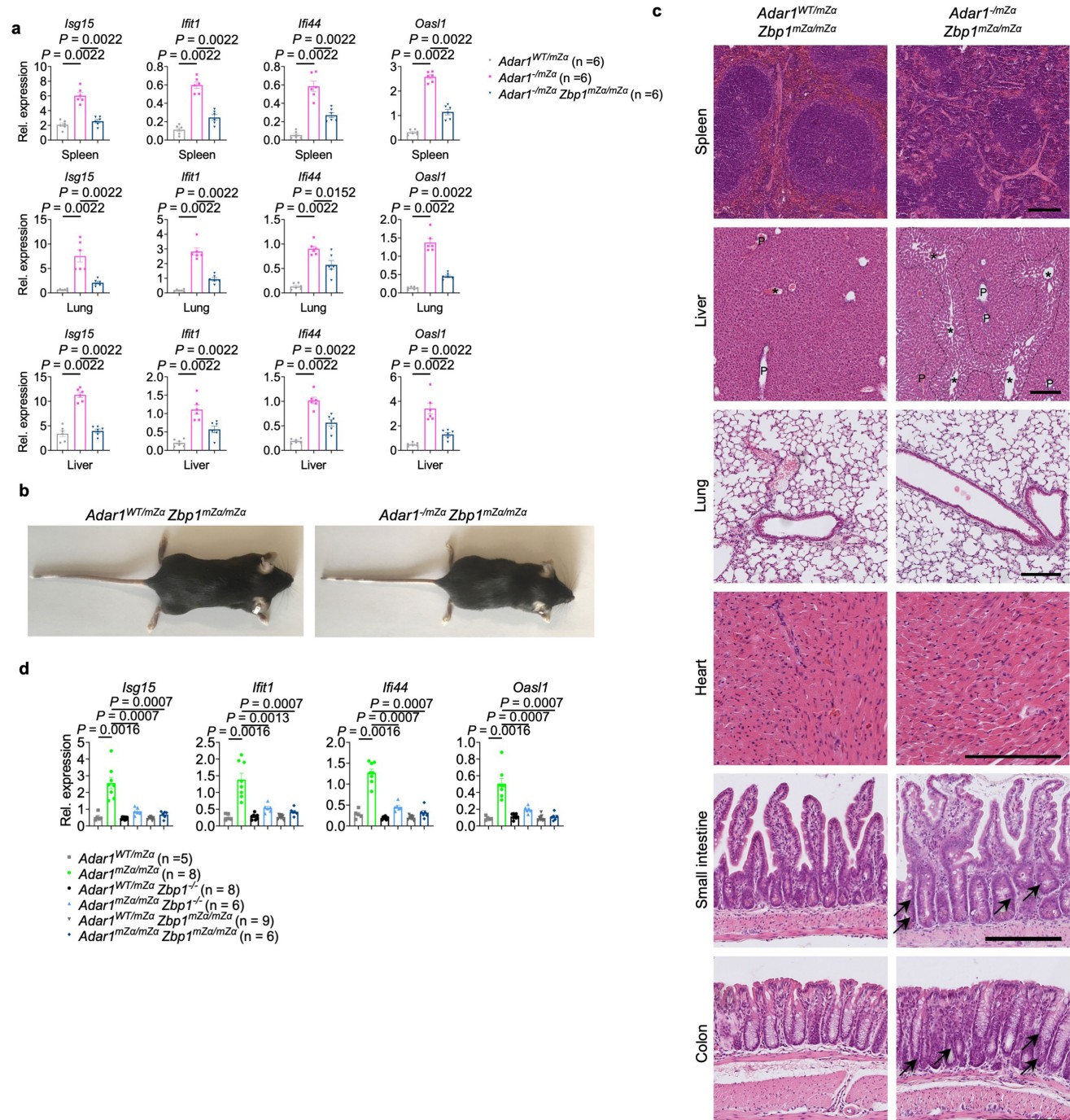

**Extended Data Fig. 9 | ZBP1 Zα domain mutation reduces ISG expression in mice expressing ADAR1 with mutated Zα domain and promotes survival to adulthood in *Adar1*<sup>-/mZα</sup> mice. a**, qRT-PCR analysis of mRNA expression of the indicated genes in spleen, lung and liver tissues from P1 mice with the indicated genotypes. **b**, Representative images of mice of the indicated genotypes at about 15 weeks of age. **c**, Representative H&E-stained sections from spleen, liver, lung, heart, small intestine and colon of mice with the indicated genotypes at 15 weeks old. Scale bar, 200 μm. Periportal and pericentral areas in liver sections are marked with P and *, respectively. Pericentral sinusoidal dilatation in *Adar1*<sup>-/mZα</sup> *Zbp1*<sup>mZα/mZα</sup> liver section is outlined with a dashed line. Black arrows indicate dying epithelial cells in small intestine and colon sections. **d**, qRT-PCR analysis of mRNA expression of the indicated genes in lung tissues from 20-week-old mice with the indicated genotypes. In **b, c**, *Adar1*<sup>WT/mZα</sup> *Zbp1*<sup>mZα/mZα</sup> (n = 5), *Adar1*<sup>-/mZα</sup> *Zbp1*<sup>mZα/mZα</sup> (n = 8). In **a, d**, dots represent individual mice, bar graphs show mean ± s.e.m and *P* values were calculated by two-sided nonparametric Mann-Whitney test.

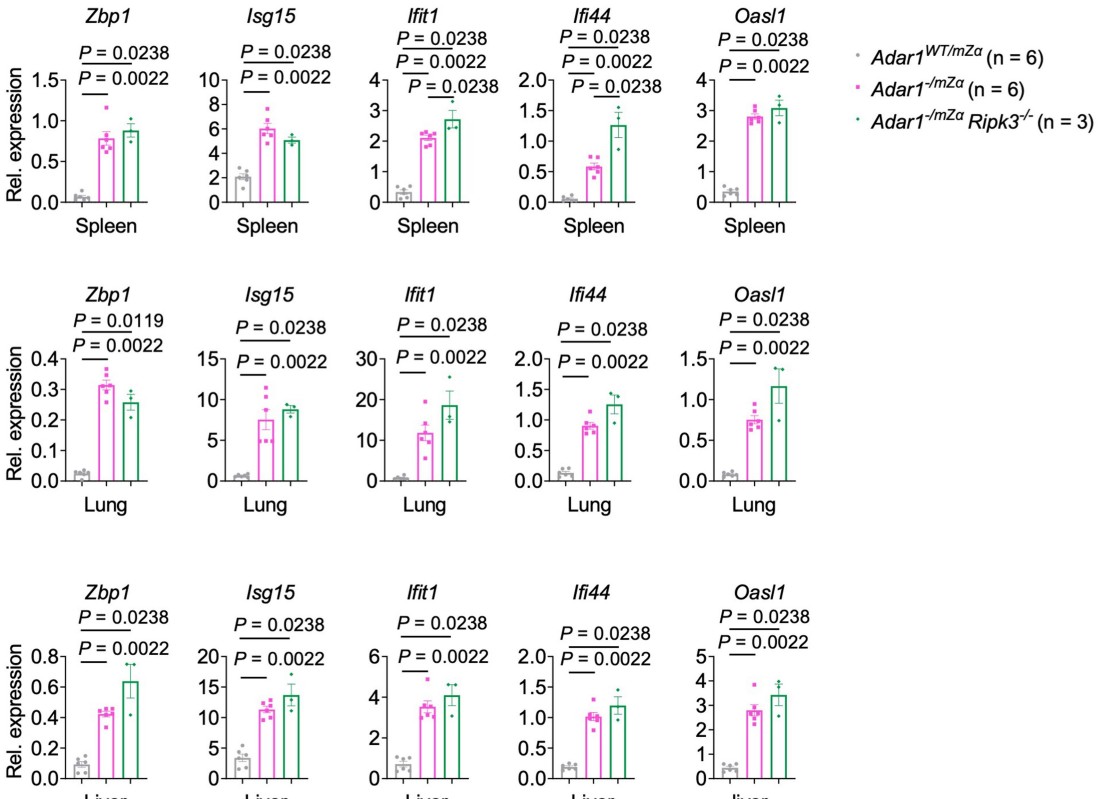

**Extended Data Fig. 10 | RIPK3 deficiency does not suppress ISG expression in *Adar1*$^{-/mZα}$ mice.** qRT-PCR analysis of mRNA expression of the indicated genes in spleen, lung and liver tissues from P1 mice with the indicated genotypes. Dots represent individual mice, bar graphs show mean ± s.e.m and *P* values were calculated by two-sided nonparametric Mann-Whitney test.

| | |
|---|---|

# Reporting Summary

## Statistics

For all statistical analyses, confirm that the following items are present in the figure legend, table legend, main text, or Methods section.

| n/a | Confirmed | |
|---|---|---|
| ☐ | ☒ | The exact sample size (*n*) for each experimental group/condition, given as a discrete number and unit of measurement |
| ☐ | ☒ | A statement on whether measurements were taken from distinct samples or whether the same sample was measured repeatedly |
| ☐ | ☒ | The statistical test(s) used AND whether they are one- or two-sided<br>*Only common tests should be described solely by name; describe more complex techniques in the Methods section.* |
| ☒ | ☐ | A description of all covariates tested |
| ☐ | ☒ | A description of any assumptions or corrections, such as tests of normality and adjustment for multiple comparisons |
| ☒ | ☐ | A full description of the statistical parameters including central tendency (e.g. means) or other basic estimates (e.g. regression coefficient) AND variation (e.g. standard deviation) or associated estimates of uncertainty (e.g. confidence intervals) |
| ☒ | ☐ | For null hypothesis testing, the test statistic (e.g. *F*, *t*, *r*) with confidence intervals, effect sizes, degrees of freedom and *P* value noted<br>*Give P values as exact values whenever suitable.* |
| ☒ | ☐ | For Bayesian analysis, information on the choice of priors and Markov chain Monte Carlo settings |
| ☒ | ☐ | For hierarchical and complex designs, identification of the appropriate level for tests and full reporting of outcomes |
| ☒ | ☐ | Estimates of effect sizes (e.g. Cohen's *d*, Pearson's *r*), indicating how they were calculated |

*Our web collection on statistics for biologists contains articles on many of the points above.*

## Software and code

Policy information about availability of computer code

| Data collection | Cell death assay data was collected by using IncuCyte S3 2018B (sartorius). QPCR data was collected by using QuantStudio 12K Flex Software (applied biosystems). Imaging data was collected by using NanoZoomer S360 Digital slide scanner C13220-01, Zeiss Axio Imager microscope and Ventana DP 200 slide scanner. Western blot data was collected by using Vilber FUSION SOLO X. Whole blood analysis data of the mice was collected by using Abacus Junior vet. |
|---|---|
| Data analysis | Statistical analysis was performed with GraphPad Prism V6. Cell death assay data was analysed by using IncuCyte S3 2018B(sartorius). Pathway analyses were performed using g:Profiler (https://biit.cs.ut.ee/gprofiler). kmers were converted into complementary kmers using shell commands and Python code. Pictures were analyzed and processed using the Omero software package (openmicroscopy.org) and the NDP.view2 Viewing software (Hamamatsu). Nuclei segmentation of individual tufts of kidney sections was performed by applying the built-in nucleus segmentation tool (QuPath). The quality of the resulting RNA-seq data was assessed using FastQC v0.11.8 (bioinformatics.babraham.ac.uk/projects/fastqc/). DESeq2 v1.22.1 within R was used for the read count normalisation and downstream differential expression analysis and visualisation was conducted within Qlucore Omics Explorer v3.3 (Qlucore, Lund, Sweden). A-to-I editing of RNA-seq data was assessed using JACUSA2. |

For manuscripts utilizing custom algorithms or software that are central to the research but not yet described in published literature, software must be made available to editors and reviewers. We strongly encourage code deposition in a community repository (e.g. GitHub). See the Nature Portfolio guidelines for submitting code & software for further information.

## Data

Policy information about availability of data

All manuscripts must include a data availability statement. This statement should provide the following information, where applicable:
- Accession codes, unique identifiers, or web links for publicly available datasets
- A description of any restrictions on data availability
- For clinical datasets or third party data, please ensure that the statement adheres to our policy

RNA-seq data have been deposited in the ArrayExpress database at EMBL-EBI (www.ebi.ac.uk/arrayexpress) under accession number E-MTAB-10953. All data supporting the findings of this study are available from the corresponding author on reasonable request.

# Field-specific reporting

Please select the one below that is the best fit for your research. If you are not sure, read the appropriate sections before making your selection.

☒ Life sciences ☐ Behavioural & social sciences ☐ Ecological, evolutionary & environmental sciences

For a reference copy of the document with all sections, see nature.com/documents/nr-reporting-summary-flat.pdf

# Life sciences study design

All studies must disclose on these points even when the disclosure is negative.

| | |
|---|---|
| Sample size | Sample size was determined empirically and was based on our previous work using RIPK1mR/mR, RIPK1E-KO, FADDIEC-KO mice. We aimed for a number of at least 3 animals per group to allow basic statistical analysis while using a justifiable number of mutant mice. Based on previous experience from similar studies, in vitro experiments with cultured cells for cell death assay were performed at least 3 times (biological replicates including 3 technical replicates for each experiment) to confirm reproducibility. |
| Data exclusions | No data was excluded from the analysis. |
| Replication | Whenever possible, readouts were performed with at least 3 animals of a given genotype. For in vitro studies we independently replicated all experiments at least 3 times (biological replicates) for cell death assay (including 3 technical replicates for all the experiments) and at least two times (biological replicates) for the immunoblotting analysis. All attempts at replication were successful. |
| Randomization | No specific method of randomization had been used to select animals. We compared groups of mice with different genotypes to assess the effect of specific genetic mutations in the phenotype. Group allocation was thus determined by the genotype of the mice. |
| Blinding | No blinding was done during and group generation as the group allocation was determined by the genotype of the mice. Histological evaluation of different tissues was performed blindly. |

# Reporting for specific materials, systems and methods

We require information from authors about some types of materials, experimental systems and methods used in many studies. Here, indicate whether each material, system or method listed is relevant to your study. If you are not sure if a list item applies to your research, read the appropriate section before selecting a response.

## Materials & experimental systems

| n/a | Involved in the study |
|---|---|
| ☐ | ☒ Antibodies |
| ☐ | ☒ Eukaryotic cell lines |
| ☒ | ☐ Palaeontology and archaeology |
| ☐ | ☒ Animals and other organisms |
| ☒ | ☐ Human research participants |
| ☒ | ☐ Clinical data |
| ☒ | ☐ Dual use research of concern |

## Methods

| n/a | Involved in the study |
|---|---|
| ☒ | ☐ ChIP-seq |
| ☒ | ☐ Flow cytometry |
| ☒ | ☐ MRI-based neuroimaging |

## Antibodies

| | |
|---|---|
| Antibodies used | 1 monoclonal rat anti-CD45 (30-F11), Cat. No. 14-0451-85, eBioscience, Dilution 1:1000 for IHC, Lot. No. E03735-1631;<br>2 monoclonal rabbit anti-Phospho-MLKL (Ser345) (D6E3G), Cat. No. 37333, Cell Signaling Technology, Dilution 1:1000 for WB, Lot. No. 2;<br>3 monoclonal rat anti-MLKL (3H1), Cat. No. MABC604, Millipore, Dilution 1:1000 for WB, Lot. No. 3256617;<br>4 monoclonal mouse anti-ZBP1 (Zippy-1), Cat. No. AG-20B-0010, Adipogen, Dilution 1:1000 for WB, Lot. No. A28231605; |

5 monoclonal mouse anti-GAPDH (1D4), Cat. No. NB300-221, NovusBiologicals, Dilution 1:1000 for WB, Lot. No. 082219;
6 monoclonal rabbit anti-Cleaved Caspase-8 (Asp387) (D5B2), Cat. No. 8592, Cell Signaling Technology, Dilution 1:1000 for WB, Lot. No. 3;
7 monoclonal rabbit anti-Caspase-8 (D35G2), Cat. No. 4790, Cell Signaling Technology, Dilution 1:1000 for WB, Lot. No. 2;
8 monoclonal rat anti-F4/80, Cat. No. MCA497, clone A3-1, AbD Serotec, dilution for IHC 1:75;
9 donkey anti-rabbit IgG – HRP, Cat. No. NA934, GE Healthcare, Dilution 1:4000 for WB, Lot. No. 17041907;
10 sheep anti-mouse IgG – HRP, Cat. No. NA931, GE Healthcare, Dilution 1:4000 for WB, Lot. No. 17028693;
11 goat anti-rat IgG – HRP, Cat. No. 112-035-003, Jackson Immuno Research, Dilution 1:4000 for WB, Lot. No. 144357;
12 goat anti-rat IgG – Biotin-SP, Cat. No. 112-065-003, Jackson Immuno Research, Dilution 1:1000 for IHC, Lot. No.112632 ;
13 monoclonal rabbit anti-Phospho-STAT1 (tyr701) (58D6), Cat. No. 9167, Cell Signaling Technology, Dilution 1:1000 for WB, Lot. No. 25;
14 polyclonal rabbit anti-STAT1, Cat. No. 9172, Cell Signaling Technology, Dilution 1:1000 for WB, Lot. No. 25;
15 polyclonal rabbit anti-ISG15, Cat. No. 2743, Cell Signaling Technology, Dilution 1:1000 for WB, Lot. No. 3;
16 monoclonal mouse anti-ADAR1 (15.8.6), Cat. No. sc-73408, Santa Cruz Biotechnology, Dilution 1:500 for WB, Lot. No. H0119;
17 monoclonal rat anti-CD3 (CD3-12), Cat. No. MCA1477, BIO-RAD, Dilution 1:100 for IHC, Lot. No. 149500D or I55872;
18 polyclonal rabbit anti-Cleaved Caspase-3 (Asp175) (D5B2), Cat. No. 9661, Cell Signaling Technology, Dilution 1:500 for IHC, Lot. No. 47;
19 monoclonal mouse anti-TREX1 (C-11), Cat. No. sc-133112, Santa Cruz Biotechnology, Dilution 1:1000 for WB, Lot. No. E1618;
20 monoclonal mouse anti- α-tubulin (B-5-1-2), Cat. No. T6074, Sigma, Dilution 1:5000 for WB, Lot. No. 046M4763V;
21 polyclonal goat anti-rabbit IgG (H+L), Cat. No. B-2770, Invitrogen, Dilution 1:1000 for IHC, Lot. 2300198
22 monoclonal rabbit anti-Iba1(EPR16588), Cat. No. ab178846, Abcam, Dilution 1:300 for IHC, Lot. No. GR3335980 3;
23 monoclonal rat anti-Mac-3 (M3/84), Cat. No. 550292, BD Biosciences, Dilution 1:200 for IHC, Lot. No. 1076190;
24 monoclonal rat anti-B220 (RA3-6B2), Cat. No. 553084, BD Biosciences, Dilution 1:200 for IHC, Lot. No. 0072031;
25 goat anti-rabbit IgG (H+L), Cat. No. 4050-08, Southern Biotech, dilution 1:200, Lot. No. I4114-N395X;
26 polyclonal rabbit anti-ZBP1 serum (custom-made by Eurogentec), Dilution 1:1000 for WB.

Validation

Validation data for all the commercial antibodies are available on vendor websites.
Custom-made ZBP1 antibody is validated for WB using ZBP1 KO cells in this manuscript and ZBP1 KO skin lysates in Lin, J., Kumari, S., Kim, C. et al. RIPK1 counteracts ZBP1-mediated necroptosis to inhibit inflammation. Nature 540, 124–128 (2016).

# Eukaryotic cell lines

Policy information about cell lines

Cell line source(s)

Mouse embryonic fibroblasts derived from the mice.

Authentication

We used primary mouse embryonic fibroblasts derived from the mice.

Mycoplasma contamination

We do not screen primary cell cultures for Mycoplasma.

Commonly misidentified lines
(See ICLAC register)

No ISLAC cell lines were used in this study

# Animals and other organisms

Policy information about studies involving animals; ARRIVE guidelines recommended for reporting animal research

Laboratory animals

C57BL/6 male and female mice, between P0 and 53 weeks of age, were used in these studies.

Wild animals

This study did not involve wild animals.

Field-collected samples

This study did not involve samples collected from the field.

Ethics oversight

All animal procedures were conducted in accordance with national and institutional guidelines and protocols were approved by the responsible local authorities in Cologne (Landesamt für Natur, Umwelt und Verbraucherschutz Nordrhein-Westfalen, Germany).

Note that full information on the approval of the study protocol must also be provided in the manuscript.

