## [Peer Review File · Nature]

Manuscript Title: ADAR1 averts fatal type I interferon induction by ZBP1

Reviewer Comments & Author Rebuttals

Reviewer Reports on the Initial Version:

Referees' comments:

Referee #1 (Remarks to the Author):

In the presented manuscript, Jiao and Wachmuth et al. present their data on ADAR1, an interferon-inducible protein which prevents aberrant activation of MDA5 by deamination of RNA retroelements. Three in a thousand individuals carry a mutation in ADAR1 (position 193). If this mutation is paired with a loss of function, we clinically observed the Aicardi-Goutieres Syndrome (AGS), a devastating inflammatory disease that can also be caused – in a slightly different clinical appearance - by TREX1-mutations. Over the last two decades, we learned a lot about inflammation and cell death by investigating diseases such as AGS. ADAR1 mutations also cause bilateral striatal necrosis, another rare disease.

Here, the authors present a mouse model of this disease - the *Adar1*-*mZa* mice (AGS mice). As expected through previously published data on MDA5 signalling to MAVS, crossing the AGS mice to a MAVS-deficient background entirely reversed the phenotype (Fig. 1A). Based on the secondary structure of the protein, the authors hypothesized that inflammatory cell death (necroinflammation) might be involved in the pathomechanism based on another Zalpha-domain containing protein known as ZBP1 (also known as DAI). Importantly, they find that ZBP1, a RHIM-domain containing protein known to trigger necroptosis, partially reverses this phenotype and rescues most of the mice (Fig. 2C). This alone is a very important finding. This rescue was slightly reduced if only the domains were mutated without a complete loss of the protein (Fig. 3C). The authors speculate that endogenous Z-RNA, a potential ligand for ZBP1, might activate ZBP1 in the absence of a functional ADAR1 Zalpha domain.

The authors noted that a constitutively suppressed interferon expression signature was unleashed by ZBP1 Zalpha in the AGS mice. Importantly, this signature was not reversed on either a RIPK1-kinase dead or a RIPK3 deficient background. This is surprising because the necroptosis pathway would have been predicted to be active following ZBP1 activation by Z-RNA.

That said, three major concerns should be addressed before I can recommend this manuscript for publication. Some minor remarks listed below may be considered by the authors.

Major concerns

- It is unclear from the data presented here if MLKL-deficiency rescues the lethal phenotype of the *Adar1*-*mZa* mice. This is an important aspect of concluding that necroptosis is not involved in this process. Even though RIPK3 currently is the only known kinase that phosphorylates MLKL, other kinases potentially might exert this function.
- The authors state in the abstract that the bilateral striatal necrosis is triggered by mutations in the

Zalpha domain of ADAR1. Despite carefully investigating spleen, lung, liver, heart and kidney of the Adar1mZa/mZa mice and the heterozygous knockout variant, brain section were not looked at. Do the Adar1mZa/mZa mice have a brain phenotype, and if so, how could this contribute to the observed pathology?

- Does the combined mutation used to generate the mouse model (N175D/Y179A) correspond to a described human mutation? Is this potentially a reason for an incomplete reversal of this phenotype on a ZBP1-ko background?

Minor remarks

- In the methods section, the part on “cell death assays” should be described in more detail (e.g. IncuCyte settings, DRAQ7 etc.).

Referee #2 (Remarks to the Author):

The Pasparakis manuscript starts by characterizing an Adar1N175D/Y179A mutant KI mouse. Line 55 indicates that these mutations disrupt Z-RNA binding by ADAR1, but presumably binding is ameliorated, rather than disrupted completely? The viability of the homozygous Adar1 KI, but post-natal lethality of the KI/KO suggest that this mutant is a hypomorph. How does expression of the ADAR1 p150 and p110 proteins in the homozygous KI cells compare to that in WT cells?

Although the Adar1 KI mouse was viable, they had an IFN gene signature in several tissues, and this was exacerbated in the Adar1 KI/KO newborns. Both dysregulated gene expression and lethality in the Adar1 KI/KO mice was prevented by MAVS deficiency. Their notable finding, however, was that lethality was also largely prevented by loss of ZBP1. RNA sequencing showed that aberrant gene expression in the Adar1 KI/KO mouse was not completely normalized by Zbp1 loss, but there was a downward trend in the IFN gene signature. However, Zbp1 loss was sufficient to normalize gene expression in the Adar1 KI mice. Thus, Zbp1 signaling appears to contribute to an IFN gene signature, but the mechanistic details are not deciphered.

Their genetic data that Zbp1 loss improved the phenotype of Adar1 KI/KO Mavs +/- mice, is consistent with ZBP1 amplifying the IFN response. They subsequently implicate the Za domains in ZBP1, and by inference ZBP1 sensing of Z-form nucleic acids, although mutation of these domains did not provide the same level of rescue as ZBP1 loss.

The authors also show that although ZBP1 loss does not, on its own, rescue lethality of the Adar1 KO, it does extend the survival of Adar1 Mavs DKO mice. This is an interesting result because it indicates that ZBP1 signaling can be triggered independent of MAVS. Again, precisely what ZBP1 is doing in this setting is unclear.

The last part of the paper attempts to address the contribution of ZBP1-induced cell death to the Adar1 KI/KO phenotype. In primary MEFs, IFN/CHX treatment killed more Adar1 KI/KO cells than WT/KO cells, and this was dependent on ZBP1, but not MAVS. Increased cell death coincided with

processing of caspase-8 and phosphorylation of MLKL. However, no morphological assessment was performed to determine which form of cell death dominated. Nor do they show if both apoptosis and necroptosis must be inhibited to prevent this death. That said, whether this IFN/CHX treatment is reflective of conditions in the Adar1 KI/KO mouse is unclear. Is there more pMLKL and processed caspase-8 in Adar1 KI/KO tissues? I acknowledge that they show increased cleaved caspase-3 in the Adar1 KI/KO intestine in Fig. 2, but what if they WB tissues for a broader panel of cell death markers?

Importantly, in contrast to Zbp1 deficiency, Ripk3 deficiency did not extend the survival of Adar1 KI/KO mice. Thus, the contribution of cell death to the Adar1KI/KO lethality remains an open question. Presumably ZBP1 could still engage RIPK1 to elicit caspase-8-dependent death and/or gene expression, but there is no data that speaks to this. The authors generated mice expressing RHIM1 mutant ZBP1 in their 2020 study, so I'd be surprised if they didn't test its ability to rescue Adar1 KI/KO mice. Did this not provide any rescue? Have they tried to IP ZBP1 from Adar1 KI/KO tissues and does this reveal any interactions with RIPK1 or TRIF per their speculation?

In sum, the characterization of the mice presented is comprehensive and largely well done. The genetic evidence is clear that ADAR1 is needed to suppress activation of ZBP1. I'm just disappointed at the lack of mechanistic insight as to how ZBP1 is contributing to an IFN gene signature and whether cell death is playing any role here. I defer to the editor, but as a reader I'm left wanting more out of a Nature paper.

Specific point:

In Fig. 3, the authors document expression of endogenous retroviral elements and RNA editing in newborn spleens from mice of the various genotypes. I am no expert in these types of analyses, but I do wonder how much of the differences seen between genotypes merely reflects differences in the cellular subsets making up the spleen. Analysis of a purified cell type of each genotype seems warranted because the bulk RNA seq data would be consistent with skewed hematopoietic subsets in the KI/KO spleens.

Referee #3 (Remarks to the Author):

It is already established that deletion or mutation of the RNA adenosine deaminase ADAR leads to a fatal autoinflammation that is dependent on the dsRNA sensing helicase MDA5 and its downstream effector MAVS, as well as PKR. Here the authors show that ZBP1 is also involved in promoting lethal interferon responses in mice with impaired ADAR1 function.

Interestingly, mutations in the Z-nucleic acid sensing domains of ADAR were very mild compared with loss of ADAR as Adar1mZa/mZa mice were born at the expected Mendelian ratio and were viable and did not develop overt pathology at least until 1 year old. This is in agreement with a previous report published last year. In contrast Adar1-/mZa mice developed a severe phenotype characterised by early postnatal lethality and reduced body weight compared to Adar1wt/mZa littermates, probably related to a gene dosage effect of ADAR1 on erythropoiesis, again in agreement with prior studies.

The ADAR1/MDA5/MAVS link has been made by several prior studies. However, here the authors present data to indicate that ZBP-1/DAI loss (or mutation within the Z-RNA sensing domain) also abrogated the IFN response seen upon loss of ADAR1 Z-nucleic acid sensing function, but not as impressively as loss of MAVS, and also further suppressed the phenotype observed upon loss of ADAR1/MAVS. ZBP1 deficiency was not sufficient to rescue the embryonic lethality of *Adar1*^{-/-} mice. However, ~40% of *Adar1*^{-/-} *Zbp1*^{-/-} *Mavs*^{-/-} mice survived to adulthood.

Given that it is already known that ZBP-1/DAI is involved in sensing of Z-nucleic acid and that loss or mutation within the Za domain of ADAR results in the accumulation of Z-form dsRNA leading to IFN responses, it is not unanticipated that ADAR loss or mutation in the Za domains is likely to result in ZBP-1 activation. Therefore, I am not sure that this, on its own, is a sufficiently novel conceptual advance, beyond recent studies, including from the author's own lab (Jiao, H. et al. Z-nucleic-acid sensing triggers ZBP1-dependent necroptosis and inflammation. *Nature* 580, 391-395), as loss/mutation of ADAR1 is simply another way of permitting z-form RNA to accumulate in cells.

The genetic data that are presented to support a role for ZBP-1 and the Za domain within the latter in the response to ADAR mutation or loss are impressive, but many of the data in the manuscript are confirmatory of prior observations and there is no elucidation of events occurring downstream of ZBP-1, except to rule out a role for RIPK3. The authors have not explored a role for RIPK1, FADD, Caspase-8, or TRIF as all are potential downstream effectors of ZBP-1 in this context.

Specific comments

1. Figure 1 is a recapitulation of what has already been published by others, that MAVS is involved in a lethal IFN response induced by impairment of the Z-nucleic acid-sensing domain of ZBP-1. The authors show that hemizygous expression of ADAR1 with mutated Za domain caused a strong upregulation of ISGs and early postnatal lethality that depended on MDA5/MAVS signaling, in agreement with recent reports.

2. I may have missed this, but have the authors included ADAR WT/- (i.e. simple hemizygous) mice in their analyses with ADAR Za/- animals. I don't see these mice included in any of the analyses to exclude the possibility that some of the effects seen with ADAR Za/- animals are simple gene dosage effects.

3. Figure 2. Implicates ZBP-1 in *Adar1*^{-/mZa} mice and demonstrates that ZBP1 deficiency could partially rescue the early lethality of *Adar1*^{-/-} *Mavs*^{-/-} mice, suggesting that ZBP1 synergises with MAVS to cause the pathology of ADAR1-deficient mice. However, the loss of ZBP-1 in the context of *Adar1*^{-/mZa} animals was much more modest than the loss of MAVS, as evidenced by the data in Fig. 2L. The authors have not included *Adar1*^{-/mZa} *MAVS*^{-/-} animals in many of the analyses in Figure 2, especially Figure 2C and 2H, which makes it hard to judge the relative contributions of MAVS and ZBP-1 to the effects reported.

4. Figure 3 suggests that endogenous Z-RNA likely derived from endogenous retroelements triggers ZBP1-dependent interferon responses in *Adar1*^{-/mZa} mice, which again is fully anticipated from prior studies, including from the authors of the present study. Several previous studies have shown that the Za domain of ADAR1 was required for editing RNAs derived from EREs and SINEs in the

mouse. Mutation of the Za domains in ZBP-1 partially rescued the pathology seen with Adar1-/mZa animals, but the protection achieved by Za domain mutation of ZBP-1 was less complete compared to ZBP1 deficiency, which implicates domains other than the Za domains of ZBP-1 in sensing effects due to the ADAR1 Za/- mutation. Of note, the Adar1-/mZa ZBP-1/- genotype data is missing in Panels c, d, e and f of Figure 3, so this obscures the magnitude of the effects seen with mutation of the Za domains in ZBP-1 relative to the null mutant. This is only included for panel 3g but would be very useful to see for the other panels here.

5. Beyond demonstrating that ZPB-1 loss or mutation of the Z-RNA sensing domain in ZBP-1 suppressed the mild phenotype seen in Adar1mZa/mZa mice, Figure 4 also contains largely negative data which suggests that the ZBP-1 response in ADAR1 mutated mice is RIPK3-independent. RIPK3 deficiency could neither prevent ISG upregulation nor rescue the lethal phenotype of Adar1-/mZa mice. However, it is known that ZBP-1 can also promote RIPK1/FADD/Caspase-8-dependent responses that have not been tested here. Similarly, it is also possible that engagement of TRIF by ZBP-1 could also account for events downstream but this has not been explored.

6. Mutations in the Zalpha domain of ADAR led to surprisingly mild phenotype, as observed by others, leading to fully viable mice that have elevated ISG responses at 1 year but no major pathology. How do the authors interpret this result and the necessity to combine the Za mutation with a null allele to produce lethality? Have they also created mutations in the Zbeta domain of ADAR1 either alone, or in combination with Zalpha mutations to see whether IFN responses are enhanced in these animals?

Author Rebuttals to Initial Comments:

Point by point response to the referees' comments:

Referee #1 (Remarks to the Author):

In the presented manuscript, Jiao and Wachmuth et al. present their data on ADAR1, an interferon-inducible protein which prevents aberrant activation of MDA5 by deamination of RNA retroelements. Three in a thousand individuals carry a mutation in ADAR1 (position 193). If this mutation is paired with a loss of function, we clinically observed the Aicardi-Goutieres Syndrome (AGS), a devastating inflammatory disease that can also be caused – in a slightly different clinical appearance - by TREX1-mutations. Over the last two decades, we learned a lot about inflammation and cell death by investigating diseases such as AGS. ADAR1 mutations also cause bilateral striatal necrosis, another rare disease.

Here, the authors present a mouse model of this disease - the *Adar1*^{-/mZα} mice (AGS mice). As expected through previously published data on MDA5 signalling to MAVS, crossing the AGS mice to a MAVS-deficient background entirely reversed the phenotype (Fig. 1A). Based on the secondary structure of the protein, the authors hypothesized that inflammatory cell death (necroinflammation) might be involved in the pathomechanism based on another Zalpha-domain containing protein known as ZBP1 (also known as DAI). Importantly, they find that ZBP1, a RHIM-domain containing protein known to trigger necroptosis, partially reverses this phenotype and rescues most of the mice (Fig. 2C). This alone is a very important finding. This rescue was slightly reduced if only the domains were mutated without a complete loss of the protein (Fig. 3C). The authors speculate that endogenous Z-RNA, a potential ligand for ZBP1, might activate ZBP1 in the absence of a functional ADAR1 Zalpha domain.

The authors noted that a constitutively suppressed interferon expression signature was unleashed by ZBP1 Zalpha in the AGS mice. Importantly, this signature was not reversed on either a RIPK1-kinase dead or a RIPK3 deficient background. This is surprising because the necroptosis pathway would have been predicted to be active following ZBP1 activation by Z-RNA.

That said, three major concerns should be addressed before I can recommend this manuscript for publication. Some minor remarks listed below may be considered by the authors.

Major concerns

- It is unclear from the data presented here if MLKL-deficiency rescues the lethal phenotype of the *Adar1*^{-/mZα} mice. This is an important aspect of concluding that necroptosis is not involved in this process. Even though RIPK3 currently is the only known kinase that phosphorylates MLKL, other kinases potentially might exert this function.

We agree with the reviewer that functionally addressing the role of MLKL is essential in order to unequivocally assess the role of necroptosis in *Adar1*^{-/mZα} mice. We have indeed performed these experiments and found that MLKL deficiency did not rescue the phenotype of *Adar1*^{-/mZα} mice, similarly to our findings with RIPK3 knockouts. These experiments are now included in Fig. 4 of

the revised manuscript. Moreover, since ZBP1 has been shown to also induce FADD-Caspase-8-mediated apoptosis in addition to RIPK3-MLKL-mediated necroptosis, we reasoned that perhaps ZBP1-mediated apoptosis could also contribute to the phenotype of the *Adar1*^{-mZα} mice. For this reason, we generated and analysed *Adar1*^{-mZα} *Fadd*^{-/-} *Mkl1*^{-/-} and *Adar1*^{-mZα} *Fadd*^{-/-} *Ripk3*^{-/-} mice and found that also these mice were not rescued, demonstrating that even combined inhibition of FADD-caspase-8-mediated apoptosis and RIPK3-MLKL-dependent necroptosis could not prevent the pathology caused by hemizygous expression of ADAR1 with mutated Zα domain. This is a very surprising finding as it shows that ZBP1 induces the pathology of *Adar1*^{-mZα} mice independently of RIPK3-MLKL-dependent necroptosis and FADD-caspase-8-mediated apoptosis. To further interrogate the potential downstream mechanisms, we also considered the possibility that RIPK1 might be implicated downstream of ZBP1 in this model, particularly as earlier studies had provided evidence that ZBP1 induces proinflammatory signalling and cytokine production by engaging RIPK1 in a RIP homotypic interaction motif (RHIM)-dependent manner. We therefore hypothesised that ZBP1 might induce RIPK1-mediated proinflammatory signalling to drive the pathology independently from cell death and for this reason we employed mice expressing RIPK1 with mutated RHIM. Because *Ripk1*^{mR/mR} mice die perinatally due to MLKL-dependent necroptosis [Lin, 2016 #31], we crossed *Adar1*^{-mZα} mice with *Ripk1*^{mR/mR} *Mkl1*^{-/-} mice. However, we found that also *Adar1*^{-mZα} *Ripk1*^{mR/mR} *Mkl1*^{-/-} mice were not rescued, demonstrating that combined inhibition of necroptosis together with RHIM-dependent RIPK1 signalling did not recapitulate the protective effect of ZBP1 deficiency. Therefore, our genetic studies provided experimental evidence demonstrating that ZBP1 promotes type I IFN responses and the associated pathology in *Adar1*^{-mZα} mice independently of RIPK3-MLKL-dependent necroptosis, FADD-caspase-8-dependent apoptosis as well as RIPK1 RHIM-dependent signalling. These findings are surprising and suggest that ZBP1 mediates the pathology of *Adar1*^{-mZα} mice by a novel mechanism of action that is different from its previously reported functions in activating RIPK3 and RIPK1 signalling to induce cell death and inflammation. Whereas our genetic studies showed that ZBP1 functions independently of RIPK1, RIPK3/MLKL and FADD/caspase-8 to promote the MAVS-dependent IFN activation and pathology of *Adar1*^{-mZα} mice, we found that ZBP1 promotes the early postnatally lethal phenotype of *Adar1*^{-/-} *Mavs*^{-/-} mice by activating RIPK3-dependent signalling. These findings revealed a dual mode of action of ZBP1 in mice with deficient ADAR1 function. ZBP1 activates RIPK3-dependent signalling, most likely involving necroptosis, to drive MAVS-independent pathology in *Adar1*^{-/-} mice, while it acts in a RIPK1- and RIPK3-independent manner to promote the MDA5-MAVS-dependent interferon response and associated pathology in *Adar1*^{-mZα} mice. Collectively, our studies revealed a novel and important role of ZBP1 in triggering type I IFN activation and severe pathology in mice with impaired ADAR1 function.

- The authors state in the abstract that the bilateral striatal necrosis is triggered by mutations in the Zalpha domain of ADAR1. Despite carefully investigating spleen, lung, liver, heart and

kidney of the *Adar1^{mZα/mZα}* mice and the heterozygous knockout variant, brain section were not looked at. Do the *Adar1^{mZα/mZα}* mice have a brain phenotype, and if so, how could this contribute to the observed pathology?

Indeed, hemizygous expression of ADAR1 with mutated $Z\alpha$ domain in humans causes a complex pathology that involves primarily the brain. The pathology is caused by hemizygous mutation of the ADAR1 $Z\alpha$ domain, with patients carrying one allele with $Z\alpha$ mutation together with a second allele usually resulting in loss of ADAR1p150 expression (Herbert, 2020).

Adar1^{mZα/mZα} mice do not develop a phenotype and remained healthy at least up to at least one year of age in our facility, as reported previously by others generating similar mutations (N173A/Y177A or P195A) (de Reuver et al., 2021; Maurano et al., 2021; Tang et al., 2021). Interestingly, another group reported that another mutation introduced in the $Z\alpha$ domain of ADAR1 (W197A) did cause a spontaneous pathology involving the brain, however, it remains unclear why this specific mutations has such a severe effect while the other $Z\alpha$ mutations do not cause a pathology when expressed in homozygosity (Nakahama et al., 2021). Since we did not observe a pathology in *Adar1^{mZα/mZα}* mice, we examined whether hemizygous expression of ADAR1 with mutated $Z\alpha$ domain in *Adar1^{-mZα}* mice might cause a brain phenotype. To this end, we performed a detailed immunohistological analysis of brain tissue from *Adar1^{-mZα}* pups in collaboration with Dr. Marco Prinz, a specialist neuropathologist with expertise also in type I interferonopathies. As shown in new ED Fig. 2 in the revised manuscript, we found no evidence of pathological alterations in *Adar1^{-mZα}* brains, which were indistinguishable from the brains of *Adar1^{WT/mZα}*, *Adar1^{-mZα} Mavs^{-/-}* or *Adar1^{-mZα} Zbp1^{-/-}* mice (ED Fig. 2c-e). Although the brain of *Adar1^{-mZα}* pups does not show histopathological features, our new RNAseq analysis revealed activation of type I IFN responses and impaired A-to-I RNA editing in the brain of these mice (see Fig. 1e and Fig. 3b in the revised manuscript). MAVS or ZBP1 deficiency did not rescue the editing defect in *Adar1^{-mZα}* brains, however the ISG response was normalized by MAVS knockout and strongly suppressed by ZBP1 knockout (Fig. 1e and Fig 3b). Therefore, hemizygous expression of ADAR1 with mutated $Z\alpha$ domain induced ZBP1- and MAVS-dependent interferon activation but did not cause profound pathology in the brain of mice.

- Does the combined mutation used to generate the mouse model (N175D/Y179A) correspond to a described human mutation? Is this potentially a reason for an incomplete reversal of this phenotype on a ZBP1-ko background?

We designed the double mutation of the $Z\alpha$ domain aiming to disrupt binding to Z-RNA. Based on structural studies, mutations of N175 and Y179 in mouse ADAR1 (these are equivalent to N173/Y177 in human ADAR1), are predicted to completely prevent binding to Z-RNA (Schade et al., 1999). We have previously used similar mutations to disrupt the $Z\alpha$ domains of ZBP1 (N46D/Y50A in $Z\alpha 1$ and N122D/Y126A in $Z\alpha 2$) and found that these mutations had an identical effect compared to deletion of the two $Z\alpha$ domains in preventing $Z\alpha$ -dependent ZBP1 activation

and function in mouse models of inflammation and viral infection (Jiao et al., 2020; Schwarzer et al., 2020). Therefore, although we have not specifically experimentally assessed Z-RNA binding in our study, based on previous work these mutations are expected to completely disrupt Z-RNA binding by ADAR1. In human patients, mutations of P193 or N173 at the Z α domain of ADAR1 have been reported to cause AGS and BSN when combined with loss of p150 ADAR1 expression from the other allele. Therefore, at least one of the mutations causing disease in humans is found on N173, which is equivalent to N175 in mice that we mutated in our mice together with Y179. The phenotype of *Adar1*^{-mZ α} mice in our study is identical to that reported by Maelfait and colleagues (de Reuver et al., 2021), who mutated the same amino acids (N175A/Y179A) of the ADAR1 Z α domain, with both mouse lines showing lethality within the first days after birth. Stetson and colleagues generated mice with mutation of P195 (equivalent to P193 in human ADAR1) mimicking the mutation found in human patients and found that P195A mutation combined with a null ADAR1 allele caused a somewhat milder phenotype compared to our mice, with most of these animals dying between 3 and 4 weeks of age and some mice even surviving up to 10 weeks (Maurano et al., 2021). These authors also showed that P195A mutation combined with specific loss of p150 ADAR1 from the other allele caused an even milder phenotype, with mice succumbing between 5 – 11 weeks of age. Therefore, double mutation of N175/Y179 causes a more severe phenotype compared to P193A, possibly because mutation of P193 does not completely abolish the interaction with Z-RNA, but this is only an assumption. Although it is possible that ZBP1 deficiency might achieve a more complete reversal of the milder phenotype of mice with hemizygous expression of P193A ADAR1, we have not generated such mice and therefore cannot comment on this.

Minor remarks

- In the methods section, the part on “cell death assays” should be described in more detail (e.g. IncuCyte settings, DRAQ7 etc.).

We have now included a detailed description of the cell death assays in the methods section of the revised manuscript.

Referee #2 (Remarks to the Author):

The Pasparakis manuscript starts by characterizing an *Adar1*^{N175D/Y179A} mutant KI mouse. Line 55 indicates that these mutations disrupt Z-RNA binding by ADAR1, but presumably binding is ameliorated, rather than disrupted completely?

As also discussed above in our response to reviewer 1, we designed the double mutation of the Z α domain aiming to disrupt binding to Z-RNA. Based on structural studies, mutation of these two amino acids in mouse ADAR1 (these are equivalent to N173/Y177 in human ADAR1) is predicted to completely prevent binding to Z-RNA (Schade et al., 1999). We have previously used similar mutations to disrupt the Z α domains of ZBP1 (N46D/Y50A in Z α 1 and N122D/Y126A in Z α 2) and found that these mutations had an identical effect compared to deletion of the two Z α domains in preventing Z α -dependent ZBP1 activation and function in mouse models of inflammation and viral infection (Jiao et al., 2020; Schwarzer et al., 2020). Therefore, although we have not specifically experimentally assessed Z-RNA binding in our study, based on previous structural and functional studies these mutations are expected to completely disrupt Z-RNA binding by ADAR1. Please also see our response to reviewer 1 above for a comparison of these mutations to the mutations found to cause AGS and BSN in human patients.

The viability of the homozygous *Adar1* KI, but post-natal lethality of the KI/KO suggest that this mutant is a hypomorph. How does expression of the ADAR1 p150 and p110 proteins in the homozygous KI cells compare to that in WT cells?

ADAR1 p150 and ADAR1 p110 expression was not reduced in spleen and lung protein extracts from *Adar1*^{mZ α /mZ α} mice compared to wild type controls (see **Figure 1A for reviewers** below). Moreover, lung fibroblasts from *Adar1*^{mZ α /mZ α} mice expressed similar levels of ADAR1 p110 and showed similar upregulation of p150 in response to stimulation with IFN γ compared to lung fibroblasts from wild type mice (**Figure 1B for reviewers**). Therefore, the *Adar1*^{mZ α} allele produces normal levels of the p110 and p150 isoforms of ADAR1 and is not a hypomorph. We agree with the reviewer that it is interesting that the homozygous *Adar1*^{mZ α /mZ α} mice are viable and show no phenotype whereas the *Adar1*^{-/mZ α} mice develop a severe, early postnatally lethal phenotype. The most likely explanation for this is that Z α mutant ADAR1 retains sufficient capacity to mediate nearly normal editing of critical transcripts when expressed at full level. Hemizygous expression could reduce ADAR1 editing capacity below a certain threshold that is critical to prevent the activation of pathogenic IFN responses. This is consistent with our findings that A-to-I editing was overall normal in *Adar1*^{mZ α /mZ α} mice but severely impaired in *Adar1*^{-/mZ α} mice (Figure 3b and Extended Data Fig. 8).

Figure 1 for reviewers. Expression of ADAR1 p110 and p150 in tissues and cells from *Adar1*^{mZ α /mZ α} mice is not reduced compared to wild type mice.

A) Spleen and lung protein extracts from 15-week-old *Adar1*^{mZ α /mZ α} and wild type control mice were analysed by immunoblotting with antibodies recognizing ADAR1. Extracts from IFN γ -stimulated *Adar1*^{WT/WT} *Mavs*^{-/-} and *Adar1*^{-/-} *Mavs*^{-/-} MEFs were used as positive and negative controls for the antibody specificity, respectively. B) Protein extracts from primary lung fibroblasts from wild type and *Adar1*^{mZ α /mZ α} mice that were untreated or stimulated with IFN γ for 24 hours were immunoblotted with antibodies against ADAR1. GAPDH was used for loading control.

Although the *Adar1* KI mouse was viable, they had an IFN gene signature in several tissues, and this was exacerbated in the *Adar1* KI/KO newborns. Both dysregulated gene expression and lethality in the *Adar1* KI/KO mice was prevented by MAVS deficiency. Their notable finding, however, was that lethality was also largely prevented by loss of ZBP1. RNA sequencing showed that aberrant gene expression in the *Adar1* KI/KO mouse was not completely normalized by *Zbp1* loss, but there was a downward trend in the IFN gene signature. However, *Zbp1* loss was sufficient to normalize gene expression in the *Adar1* KI mice. Thus, *Zbp1* signaling appears to contribute to an IFN gene signature, but the mechanistic details are not deciphered.

Their genetic data that *Zbp1* loss improved the phenotype of *Adar1* KI/KO *Mavs* +/- mice, is consistent with ZBP1 amplifying the IFN response. They subsequently implicate the Za domains in ZBP1, and by inference ZBP1 sensing of Z-form nucleic acids, although mutation of these domains did not provide the same level of rescue as ZBP1 loss.

In the revised manuscript we now include additional RNAseq analyses and provide a very much strengthened dataset demonstrating that ZBP1 deficiency strongly suppressed the type I IFN response in both *Adar1*^{mZ α /mZ α} and *Adar1*^{-/mZ α} mice. The reduction in ISG expression achieved by ZBP1 deficiency is not simply a downward trend. Analysis of the RNAseq dataset showed

that *Adar1*^{-/mZα} *Zbp1*^{-/-} mice had a highly statistically significant reduction in ISG expression in three different tissues analysed, namely brain, lung and spleen, compared to *Adar1*^{-/mZα} mice (Fig. 1e of the revised manuscript). However, as the reviewer correctly points out, the effect of ZBP1 deficiency was not as strong as that of MAVS knockout, which nearly completely prevented the upregulation of ISGs (Fig. 1e). Our new RNAseq analysis in adult mice also showed that ZBP1 knockout had a stronger effect in suppressing the type I IFN response compared to heterozygous knockout of MAVS, and that ZBP1 deficiency synergised with MAVS heterozygosity to strongly suppress ISG expression in *Adar1*^{-/mZα} mice (Fig. 2e). Moreover, these analyses showed that MAVS deficiency could not fully normalise the expression of ISGs in *Adar1*^{-/mZα} mice, as *Adar1*^{-/mZα} *Mavs*^{-/-} mice showed a small but statistically significant upregulation of ISGs compared to C57BL/6 and *Adar1*^{WT/mZα} control mice (Fig. 2e). Importantly, combined deficiency in MAVS and ZBP1 could completely normalise the ISG response, with *Adar1*^{-/mZα} *Mavs*^{-/-} *Zbp1*^{-/-} mice being indistinguishable from *Adar1*^{WT/mZα} or C57Bl/6N controls, showing that ZBP1 also contributes to the type I IFN response independently of MAVS (Fig. 2e). Collectively, these results provided experimental evidence unequivocally demonstrating that ZBP1 critically contributes to the activation of the pathogenic type I IFN response that causes the severe pathology of *Adar1*^{-/mZα} mice. Our results suggest that ZBP1 is required for full activation of the MAVS-dependent type I IFN response but also acts in a MAVS-independent manner to promote ISG expression in *Adar1*^{-/mZα} mice. Mutation of the ZBP1 Zα domains had a somewhat less pronounced effect in terms of rescuing the early postnatal lethality of *Adar1*^{-/mZα} mice compared to ZBP1 knockout (about 50 % of *Adar1*^{-/mZα} *Zbp1*^{mZα/mZα} mice survived up to 15 weeks of age, compared to about 70 % of the *Adar1*^{-/mZα} *Zbp1*^{-/-} mice, Fig. 3d). *Adar1*^{-/mZα} *Zbp1*^{mZα/mZα} mice showed a similar body weight compared to *Adar1*^{-/mZα} *Zbp1*^{-/-} mice both in newborn pups and in the adult (Fig. 3e, h) but reduced RBC, HGB and HCT levels in the blood (Fig. 3f, i), showing that Zα domain mutation of ZBP1 could not rescue the erythropoiesis defect to the extent of the ZBP1 knockout. However, RNAseq analysis of lung tissue both from pups and from adult mice revealed an overall similar level of suppression of ISG expression in *Adar1*^{-/mZα} *Zbp1*^{mZα/mZα} and *Adar1*^{-/mZα} *Zbp1*^{-/-} mice (Fig. 3g, j), showing that Zα-dependent sensing of endogenous ligands generated in ADAR1 deficient cells activates the ZBP1-dependent IFN response. Collectively, these results showed that the role of ZBP1 in promoting the IFN response and lethal pathology of *Adar1*^{-/mZα} is largely dependent on Zα-mediated sensing of endogenous ligands, however, ZBP1 also exerts Zα-independent functions as we showed previously in mouse models of MCMV infection and skin inflammation induced by epidermal keratinocyte-specific RIPK1 knockout.

We would also like to emphasise our results showing that ZBP1 deficiency did not ameliorate the cGAS-STING-dependent type I IFN response and associated pathology caused by deficiency in the cytosolic DNase TREX1 (ED Fig. 7). These results are very important in the context of our findings in *Adar1*^{-/mZα} mice, because they show that ZBP1 does not contribute to the activation of type I IFN responses by cytosolic DNA. Therefore, our results revealed a novel

function of ZBP1 that is different from its previously suggested role as a DNA sensor activating interferons in a study published by Taniguchi and colleagues several years ago (Takaoka et al., 2007), which the Akira group could not confirm subsequently using ZBP1 knockout mice (Ishii et al., 2008). We have made a great effort to genetically dissect the downstream signalling by which ZBP1 induces the type I IFN response, as explained in detail below in response to the specific relevant comment of the reviewer. In summary, these results showed that ZBP1 activates the type I IFN response causing the severe pathology in *Adar1*^{-mZα} mice independently from RIPK1, RIPK3, MLKL-mediated necroptosis and FADD-caspase-8-mediated apoptosis. This is very surprising, as all known functions of ZBP1 to date have been assigned to the activation of RIPK3- and RIPK1-dependent signalling primarily involving MLKL-dependent necroptosis and FADD-caspase-8-dependent apoptosis but also RIPK1-dependent inflammatory signalling. Therefore, our results suggest that ZBP1 promotes type I IFN activation and the associated pathology in response to ADAR1 Zα domain mutation by a novel mechanism of action, which remains to be fully elucidated.

The authors also show that although ZBP1 loss does not, on its own, rescue lethality of the Adar1 KO, it does extend the survival of Adar1 Mavs DKO mice. This is an interesting result because it indicates that ZBP1 signaling can be triggered independent of MAVS. Again, precisely what ZBP1 is doing in this setting is unclear.

We agree with the reviewer that the results showing that ZBP1 promotes perinatal lethality in *Adar1*^{-/-} *Mavs*^{-/-} mice are very interesting as they reveal a MAVS-independent role of ZBP1. We were curious whether this role of ZBP1 is also independent of RIPK3 and FADD, therefore we crossed the *Adar1*^{-/-} *Mavs*^{-/-} mice with *Fadd*^{WT/-} *Ripk3*^{-/-} mice. Importantly, we found that RIPK3 deficiency, with or without heterozygous knockout of FADD, could also partly prevent the lethality of *Adar1*^{-/-} *Mavs*^{-/-} mice similarly to ZBP1 deficiency (Fig. 2f of the revised manuscript). These results showed that ZBP1 promotes perinatal lethality in *Adar1*^{-/-} *Mavs*^{-/-} mice by inducing RIPK3-dependent signalling. Collectively, our results revealed that ZBP1 induces RIPK3-dependent signalling to cause MAVS-independent pathology in *Adar1*^{-/-} *Mavs*^{-/-} mice, but acts in a manner independent of RIPK3 and RIPK1 to promote the MAVS-dependent type I IFN response and associated pathology in *Adar1*^{-mZα} mice.

The last part of the paper attempts to address the contribution of ZBP1-induced cell death to the Adar1 KI/KO phenotype. In primary MEFs, IFN/CHX treatment killed more Adar1 KI/KO cells than WT/KO cells, and this was dependent on ZBP1, but not MAVS. Increased cell death coincided with processing of caspase-8 and phosphorylation of MLKL. However, no morphological assessment was performed to determine which form of cell death dominated. Nor do they show if both apoptosis and necroptosis must be inhibited to prevent this death. That said, whether this IFN/CHX treatment is reflective of conditions in the Adar1 KI/KO mouse is unclear. Is there more pMLKL and processed caspase-8 in Adar1 KI/KO tissues? I acknowledge

that they show increased cleaved caspase-3 in the *Adar1* KI/KO intestine in Fig. 2, but what if they WB tissues for a broader panel of cell death markers?

We have made an extensive effort to address the contribution of ZBP1-induced cell death to the phenotype of the *Adar1*^{-mZα} mice. Following the reviewer's advice, we also made an effort to assess activation of RIPK3 and MLKL in the intestines of *Adar1*^{-mZα} mice by western blotting. These blots proved to be particularly challenging and we could not observe MLKL or RIPK3 phosphorylation in these samples, as shown in **Figure 2 for reviewers** below, although we are not fully confident that the assay has the required sensitivity to detect activation of these markers in a small percentage of cells that would be expected to undergo necroptosis within the tissue. We could detect however increased activation of caspase-8 in the intestines of *Adar1*^{-mZα} mice that was partly inhibited by ZBP1 deficiency (**Figure 2 for reviewers**). However, these assays do not provide functional proof and can only be suggestive. Therefore, to unequivocally assess the role of necroptosis and apoptosis we resorted to using genetic studies. To this end, we generated *Adar1*^{-mZα} *Mlkl*^{-/-} mice and found that MLKL deficiency did not rescue the phenotype of *Adar1*^{-mZα} mice similarly to our findings with RIPK3 knockouts (Fig. 4). Since ZBP1 has been shown to also induce FADD-Caspase-8-mediated apoptosis in addition to RIPK3-MLKL-mediated necroptosis, we reasoned that perhaps ZBP1-mediated apoptosis could also contribute to the phenotype of the *Adar1*^{-mZα} mice. For this reason, we generated and analysed *Adar1*^{-mZα} *Fadd*^{-/-} *Mlkl*^{-/-} and *Adar1*^{-mZα} *Fadd*^{-/-} *Ripk3*^{-/-} mice and found that also these mice were not rescued (Fig. 4), demonstrating that even combined inhibition of FADD-caspase-8-mediated apoptosis and RIPK3-MLKL-dependent necroptosis could not prevent the pathology caused by hemizygous expression of ADAR1 with mutated Zα domain. This is a very surprising finding as it implies that ZBP1 induces the pathology of *Adar1*^{-mZα} mice independently of RIPK3-MLKL-mediated necroptosis and FADD-caspase-8-mediated apoptosis.

However, as discussed also above, in contrast to its RIPK3-MLKL- and FADD-caspase-8-independent role in *Adar1*^{-mZα} mice, our new results revealed that ZBP1 promotes perinatal lethality in *Adar1*^{-/-} *Mavs*^{-/-} mice by inducing RIPK3-dependent signalling (Fig. 2f). We therefore focused on analysing cell death responses in *Adar1*^{-/-} *Mavs*^{-/-} MEFs. As we had already shown in the previous version of the manuscript, IFNγ stimulation, resulting in strong upregulation of ZBP1 expression, failed to induce cell death in *Adar1*^{-/-} *Mavs*^{-/-} MEFs (ED Fig. 6f, g). However, treatment with a low concentration of CHX could sensitize these cells to undergo ZBP1-dependent cell death after IFNγ stimulation (ED Fig. 6h). Under these conditions, *Adar1*^{-/-} *Mavs*^{-/-} MEFs showed increased cleavage of caspase-8 and phosphorylation of MLKL, that were inhibited by ZBP1 deficiency (ED Fig. 6i), suggesting that these cells undergo both apoptosis and necroptosis. Indeed, combined inhibition of RIPK3 kinase activity and caspases largely prevented IFNγ+CHX-induced cell death in *Adar1*^{-mZα} *Mavs*^{-/-} MEFs (ED Fig. 6j). These results support that ZBP1 promotes the MAVS-independent lethal phenotype of *Adar1*^{-mZα} *Mavs*^{-/-} mice by inducing RIPK3-dependent cell death. While it remains unclear to what extent the ZBP1-RIPK3-dependent cell death induced in *Adar1*^{-/-} *Mavs*^{-/-} MEFs treated with IFNγ+CHX relates to

the *in vivo* role of ZBP1-RIPK3 signalling in mediating the pathology of *Adar1*^{-/-} *Mavs*^{-/-} mice, we speculate that CHX treatment might mimic the activation of PKR and resulting inhibition of protein translation in ADAR1-deficient cells (Maurano et al., 2021). Collectively, our results showed that ZBP1 induces RIPK3-dependent cell death to cause MAVS-independent pathology in *Adar1*^{-/-} *Mavs*^{-/-} mice, but acts in a manner independent of RIPK3-MLKL-mediated necroptosis and FADD-caspase-8-mediated apoptosis to promote the MAVS-dependent type I IFN response and associated pathology in *Adar1*^{-/mZα} mice.

Figure 2 for reviewers. Immunoblot analysis of caspase-8, RIPK3 and MLKL activation in small intestinal tissues from *Adar1*^{-/mZα} and *Adar1*^{-/mZα} *Zbp1*^{-/-} mice.

Total protein extracts from small intestinal tissues of mice with the indicated genotypes were immunoblotted with the indicated antibodies. Blots on the left show increased activation of caspase-8 in SI from *Adar1*^{-/mZα} mice, which is decreased in *Adar1*^{-/mZα} *Zbp1*^{-/-} mice. Blots on the right show that we were not able to detect phosphorylation of MLKL or RIPK3 in SI from *Adar1*^{-/mZα} mice. Extracts from IFN γ +CHX-stimulated *Adar1*^{WT/mZα} MEFs was used as positive control for cleaved caspase-8. Extracts from TSE (TNF+Birinapant+Emricasan)-stimulated lung fibroblasts was used as positive control for pRIPK3 and pMLKL. GAPDH was used as loading control.

Importantly, in contrast to *Zbp1* deficiency, *Ripk3* deficiency did not extend the survival of *Adar1* KI/KO mice. Thus, the contribution of cell death to the *Adar1* KI/KO lethality remains an open question. Presumably ZBP1 could still engage RIPK1 to elicit caspase-8-dependent death and/or gene expression, but there is no data that speaks to this. The authors generated mice expressing RHIM1 mutant ZBP1 in their 2020 study, so I'd be surprised if they didn't test its ability to rescue *Adar1* KI/KO mice. Did this not provide any rescue? Have they tried to IP ZBP1 from *Adar1* KI/KO tissues and does this reveal any interactions with RIPK1 or TRIF per their speculation?

As discussed above, our new genetic experiments revealed that inhibition of both RIPK3-MLKL-dependent necroptosis and FADD-caspase-8-dependent apoptosis could not rescue the *Adar1*^{-mZ α} mice as ZBP1 deficiency did. These results showed that ZBP1 induces the type I IFN response and associated pathology in *Adar1*^{-mZ α} mice independently of necroptosis and caspase-8-induced apoptosis. As the reviewer suggested, we also considered the possibility that RIPK1 might be implicated downstream of ZBP1 in this model, particularly as early studies had provided evidence that ZBP1 induces cytokine production by engaging RIPK1 in a RHIM-dependent manner (Kaiser et al., 2008; Rebsamen et al., 2009). We therefore reasoned that ZBP1 might induce RIPK1-mediated proinflammatory signalling to drive the pathology independently from cell death and for this reason we also crossed the *Adar1*^{-mZ α} mice with mice expressing RIPK1 with mutated RHIM in an MLKL deficient background (*Ripk1*^{mR/mR} *Mkl1*^{-/-}). However, we found that also *Adar1*^{-mZ α} *Ripk1*^{mR/mR} *Mkl1*^{-/-} mice were not rescued, demonstrating that combined inhibition of necroptosis together with RHIM-dependent RIPK1 signalling did not recapitulate the protective effect of ZBP1 deficiency. Therefore, our genetic studies provided experimental evidence demonstrating that ZBP1 promotes type I IFN responses and the associated pathology in *Adar1*^{-mZ α} mice independently of RIPK3-MLKL-dependent necroptosis, FADD-caspase-8-dependent apoptosis as well as RIPK1 RHIM-dependent signalling.

We have also considered the possibility that TRIF might be involved in ZBP1-mediated activation of type I IFN responses and the associated pathology of *Adar1*^{-mZ α} mice. However, we thought this would be unlikely based on a very interesting paper published last year by Poltorak and colleagues (Muendlein et al., 2021). These authors showed that ZBP1 is implicated in TRIF-induced signalling, however, this function was dependent on RHIM-dependent interactions with RIPK1 (Muendlein et al., 2021), therefore we would expect this to be inhibited in *Adar1*^{-mZ α} *Ripk1*^{mR/mR} *Mkl1*^{-/-} mice. However, in light of our recent findings excluding a role of RIPK3 and RIPK1 we agree that studying the potential involvement of TRIF as a possible downstream mediator of the ZBP1-dependent pathology of *Adar1*^{-mZ α} mice is warranted. The reviewer also correctly points out that crossing the *Adar1*^{-mZ α} mice to mice expressing ZBP1 with mutated RHIM could also provide additional mechanistic insight. Unfortunately, we had not given high priority to crossing the *Adar1*^{-mZ α} mice with the *Zbp1*^{mR1/mR1} mice, as we expected that ZBP1 could mediate this phenotype as usually by inducing RIPK3-dependent necroptosis, and therefore mutating the ZBP1 RHIM domain would simply recapitulate the knockout phenotype. In retrospect, it was a mistake not to give a higher priority to the genetic crosses assessing the role of the ZBP1 RHIM and also of TRIF. We have now started the crosses to generate *Adar1*^{-mZ α} *Zbp1*^{mR1/mR1} mice as well as *Adar1*^{-mZ α} *Trif*^{-/-} mice. However, considering the time it takes to obtain the required permission from the local government authorities regulating animal experiments to start these crosses (this unfortunately usually takes several weeks to months in Germany) and then the time it takes to interbreed these strains to homozygosity, regrettably we could not obtain these results within a time frame allowing to include them in the current manuscript. We have also attempted to probe for possible interaction of ZBP1 with RIPK1 and

TRIF by immunoprecipitating ZBP1 from tissues of *Adar1*^{-mZ α} mice, however, these IPs from tissue lysates did not work in our hands. Taken together, our new genetic studies included in the revised manuscript demonstrate that ZBP1 promotes type I IFN responses and the early postnatally lethal pathology of *Adar1*^{-mZ α} mice independently of RIPK3, RIPK1 RHIM-dependent signalling, MLKL-dependent necroptosis and FADD-caspase-8-dependent apoptosis. We find these results fascinating, as they show that ZBP1 drives this response by a mechanism different to its previously assigned, RIPK3- and RIPK1-dependent functions in mediating anti-viral defence and inflammation.

In sum, the characterization of the mice presented is comprehensive and largely well done. The genetic evidence is clear that ADAR1 is needed to suppress activation of ZBP1. I'm just disappointed at the lack of mechanistic insight as to how ZBP1 is contributing to an IFN gene signature and whether cell death is playing any role here. I defer to the editor, but as a reader I'm left wanting more out of a Nature paper.

As discussed and explained in detail above, we have made a great effort to dissect the downstream mechanisms by which ZBP1 promotes type I IFN responses and the lethal pathology of *Adar1*^{-mZ α} mice. To provide the most physiologically relevant results, we have used extensive genetic crosses to interrogate the role of downstream pathways *in vivo*. In fact, our revised manuscript contains data from 10 different double knockout or mutant mouse strains and 7 different triple knockout or mutant mouse strains. We now provide genetic evidence demonstrating that even combined inhibition of both RIPK3-MLKL-dependent necroptosis and FADD-caspase-8-mediated apoptosis, or loss of RHIM-dependent RIPK1 function combined with MLKL deficiency could not phenocopy the effect of ZBP1 deficiency. These results argue that ZBP1 promotes type I IFN activation and the associated pathology in *Adar1*^{-mZ α} mice independently of RIPK3, necroptosis, FADD-caspase-8-dependent necroptosis and of RIPK1-mediated signalling. Our experiments also revealed a very interesting dual function of ZBP1, which acts via RIPK3 to mediate the MAVS-independent pathology of *Adar1*^{-/-} mice, while it acts in a RIPK1- and RIPK3-independent manner to promote the MAVS-dependent IFN response and pathology in *Adar1*^{-mZ α} mice. Even though at this stage we cannot deliver more specific mechanistic insight into how ZBP1 mediates this function, we hope that the reviewer will appreciate the great effort we have put in dissecting these pathways and will agree that our results reveal a new and exciting function of ZBP1 in regulating cellular responses to endogenous RNA. Our findings provide a paradigm shift in our understanding of the mechanisms regulating cellular responses to retroelement-derived RNA, highlighting the critical role of Z-RNA formation and sensing by the two proteins containing Z α domains, namely ADAR1 and ZBP1, in controlling IFN responses to endogenous RNA.

Specific point:

In Fig. 3, the authors document expression of endogenous retroviral elements and RNA editing

in newborn spleens from mice of the various genotypes. I am no expert in these types of analyses, but I do wonder how much of the differences seen between genotypes merely reflects differences in the cellular subsets making up the spleen. Analysis of a purified cell type of each genotype seems warranted because the bulk RNA seq data would be consistent with skewed hematopoietic subsets in the KI/KO spleens.

We have examined the cell populations of the spleen in *Adar1*^{-/mZ α} mice compared to *Adar1*^{WT/mZ α} littermates and found that although the spleens of *Adar1*^{-/mZ α} pups are smaller and contain less cells, the overall composition of the different immune cell populations is largely similar with the control mice (**Figure 3 for reviewers**). The small differences detected primarily in increased numbers of monocytes could not account for the dramatic upregulation of retroelement expression or the strong reduction in A-to-I edited transcripts in *Adar1*^{-/mZ α} mice (Fig. 3a-c). The reviewer suggested to perform RNAseq on a purified cell type, however this is not feasible as a large quantity of RNA (1 μ g of total RNA) is needed to perform the deep sequencing required for editing analysis (we used TruSeq ribo-zero protocol with 50M reads), which will be impossible to obtain from a sorted cell population from one newborn mouse spleen. Instead, to further strengthen the dataset, we have now performed RNAseq analysis in RNA isolated from total brain tissue of *Adar1*^{WT/mZ α} , *Adar1*^{mZ α /mZ α} , *Adar1*^{-/mZ α} , *Adar1*^{-/mZ α} *Mavs*^{-/-} and *Adar1*^{-/mZ α} *Zbp1*^{-/-} pups. As shown in Fig 3b, analysis of brain RNAseq data also revealed strongly impaired A-to-I editing in *Adar1*^{-/mZ α} compared to *Adar1*^{WT/mZ α} mice, while *Adar1*^{mZ α /mZ α} mice did not show altered editing. Importantly, neither MAVS deficiency nor ZBP1 deficiency could rescue the editing defect of *Adar1*^{-/mZ α} mice, demonstrating that even when the ISG response is prevented and the hematopoietic defects are rescued the editing defect persists.

Figure 3 for reviewers. Analysis of major cell populations in the spleen of newborn *Adar1*^{WT/mZa} and *Adar1*^{-/mZa} mice.

(A) Gating strategy for the identification of the major immune cell populations in splenocytes isolated from mice at post-natal days 0-1. (n=3 for *Adar1*^{WT/mZa} and n=4 for *Adar1*^{-/mZa} mice). Single cell suspension of spleens was stained with the following antibodies: Live/Dead: APC-Cy7; CD45: BV711; CD19: BV605; CD3: BV786; CD4: PE Texas Red; CD8: BV510; NK1.1: APC; CD11b: BV421; Ly6G: PE; Ly6C: PerCP-Cy5.5; CD11c: PE Cy7; MHCII: AlexFluor 700; CD44: FITC. No CD3⁺ T cells could be detected as expected in newborn mice. Also, CD11b⁺CD11c⁺ and CD11c⁺ MHCII⁺ cells were almost undetectable. **(B)** Graph depicting the frequency of the different cell populations as a percentage of total CD45⁺ cells.

Referee #3 (Remarks to the Author):

It is already established that deletion or mutation of the RNA adenosine deaminase ADAR leads to a fatal autoinflammation that is dependent on the dsRNA sensing helicase MDA5 and its downstream effector MAVS, as well as PKR. Here the authors show that ZBP1 is also involved in promoting lethal interferon responses in mice with impaired ADAR1 function.

Interestingly, mutations in the Z-nucleic acid sensing domains of ADAR were very mild compared with loss of ADAR as *Adar1^{mZa/mZa}* mice were born at the expected Mendelian ratio and were viable and did not develop overt pathology at least until 1 year old. This is in agreement with a previous report published last year. In contrast *Adar1^{-/mZa}* mice developed a severe phenotype characterised by early postnatal lethality and reduced body weight compared to *Adar1^{wt/mZa}* littermates, probably related to a gene dosage effect of ADAR1 on erythropoiesis, again in agreement with prior studies.

The ADAR1/MDA5/MAVS link has been made by several prior studies. However, here the authors present data to indicate that ZBP-1/DAI loss (or mutation within the Z-RNA sensing domain) also abrogated the IFN response seen upon loss of ADAR1 Z-nucleic acid sensing function, but not as impressively as loss of MAVS, and also further suppressed the phenotype observed upon loss of ADAR1/MAVS. ZBP1 deficiency was not sufficient to rescue the embryonic lethality of *Adar1^{-/-}* mice. However, ~40% of *Adar1^{-/-} Zbp1^{-/-} Mavs^{-/-}* mice survived to adulthood.

Given that it is already known that ZBP-1/DAI is involved in sensing of Z-nucleic acid and that loss or mutation within the Za domain of ADAR results in the accumulation of Z-form dsRNA leading to IFN responses, it is not unanticipated that ADAR loss or mutation in the Za domains is likely to result in ZBP-1 activation. Therefore, I am not sure that this, on its own, is a sufficiently novel conceptual advance, beyond recent studies, including from the author's own lab (Jiao, H. et al. Z-nucleic-acid sensing triggers ZBP1-dependent necroptosis and inflammation. *Nature* 580, 391-395), as loss/mutation of ADAR1 is simply another way of permitting z-form RNA to accumulate in cells.

While we agree with the reviewer that ZBP1 was previously implicated in sensing Z-RNA to induce cell death in anti-viral immunity and models of inflammation, we respectfully disagree that the functional connection between ADAR1 and ZBP1 was expected and therefore is not novel. Of course, ADAR1 p150 and ZBP1 are the only two proteins in the mammalian genome known to contain $Z\alpha$ domains, therefore one might have thought these should functionally interact. This was indeed the starting point of our work more than 5 years ago, when we hypothesized that ADAR1 and ZBP1 could be functionally connected via their $Z\alpha$ domains. To the best of our knowledge, this connection has not been documented to date, therefore, our studies are the first to experimentally demonstrate that $Z\alpha$ -dependent ZBP1 signalling causes a pathology when $Z\alpha$ -dependent ADAR1 function is impaired. Moreover, whereas the fact that

ADAR1 Z α domain mutations were found to cause AGS and BSN clearly pointed towards an important function of the Z α domain in restraining type I IFN responses, the finding that these IFN responses are to a large extent dependent on Z α -dependent ZBP1 function is novel. ZBP1 was proposed to function as DNA sensor activating IFN responses 15 years ago by Taniguchi and colleagues (Takaoka et al., 2007), however, this function could not be confirmed in subsequent studies by Akira and colleagues using knockout mice (Ishii et al., 2008). Our findings that ZBP1 knockout could not rescue or ameliorate the cGAS-STING-dependent IFN response and associated myocarditis in *Trex1*^{-/-} mice provide clear evidence that ZBP1 does not promote IFN responses to cytosolic DNA, supporting a specific and important function of ZBP1 in cells with ADAR1 deficiency. Therefore, our results provide experimental evidence showing that the capacity of ADAR1 to bind Z-RNA is critical to prevent the accumulation of a ligand, most likely Z-RNA itself, that activates ZBP1 to promote type I IFN responses. We hope the reviewer will agree with us that experimentally demonstrating this link and revealing a new function of ZBP1 in cooperating with MAVS to promote pathogenic type I IFN responses in cells with impaired ADAR1 function, provides a paradigm shift in our understanding of the mechanisms controlling cellular responses to endogenous retroelement-derived RNA.

The genetic data that are presented to support a role for ZBP-1 and the Za domain within the latter in the response to ADAR mutation or loss are impressive, but many of the data in the manuscript are confirmatory of prior observations and there is no elucidation of events occurring downstream of ZBP-1, except to rule out a role for RIPK3. The authors have not explored a role for RIPK1, FADD, Caspase-8, or TRIF as all are potential downstream effectors of ZBP-1 in this context.

In the revised manuscript, we have now genetically dissected the role of possible downstream effectors of ZBP1 and provide experimental evidence that ZBP1 promotes type I IFN responses and the associated pathology in *Adar1*^{-mZ α} mice independently of RIPK1, RIPK3, MLKL-dependent necroptosis and FADD-caspase-8-dependent apoptosis. Please see a detailed response to the relevant specific comment (number 5) below.

Specific comments

1. Figure 1 is a recapitulation of what has already been published by others, that MAVS is involved in a lethal IFN response induced by impairment of the Z-nucleic acid-sensing domain of ZBP-1. The authors show that hemizygous expression of ADAR1 with mutated Za domain caused a strong upregulation of ISGs and early postnatal lethality that depended on MDA5/MAVS signaling, in agreement with recent reports.

Indeed, Fig. 1 in the previous version of the manuscript contained primarily the description of the phenotype caused by homozygous or hemizygous expression of ADAR1 with Z α domain mutation and its rescue by MAVS, which was reported recently by others. We felt that, since our ADAR1 mutant mice had not been published previously, it was important to report the

phenotype and the basic characterisation. In the revised manuscript, which now contains a very much expanded and strengthened dataset, we have re-organised the figures with Fig. 1 now containing all the basic characterisation of the phenotype of the *Adar1*^{-mZα} mice and the role of MAVS, but in addition also presenting the rescue of the phenotype by ZBP1 knockout, which is a novel and important finding.

2. I may have missed this, but have the authors included ADAR WT/- (i.e. simple hemizygous) mice in their analyses with ADAR Zα/- animals. I don't see these mice included in any of the analyses to exclude the possibility that some of the effects seen with ADAR Zα/- animals are simple gene dosage effects.

ADAR1 heterozygous knockout (*Adar1*^{WT/-}) mice are born normally and reach adulthood remaining healthy for as long as we observed them in our facility (up to 7-8 months of age). This has also been reported by others, for example recently by Maurano et al. (Maurano et al., 2021), therefore heterozygous knockout of ADAR1 causes no pathology. We have not performed an extensive characterisation of *Adar1*^{WT/-} mice in our experiments, as we preferred to use *Adar1*^{WT/mZα} as controls because of the nature of the breedings that usually relied on one parent being heterozygous for the ADAR1 knockout allele (*Adar1*^{WT/-}) and the other parent homozygous for the Zα mutant ADAR1 allele (*Adar1*^{mZα/mZα}). Nevertheless, in our experiments addressing the role of MLKL we did obtain some *Adar1*^{WT/-} *Mlkl*^{-/-} pups and used these to assess the expression of ISGs side-by-side with *Adar1*^{-mZα} *Mlkl*^{-/-} mice. As shown in new Fig. 4e, *Adar1*^{WT/-} *Mlkl*^{-/-} mice did not show any upregulation of ISG expression as opposed to *Adar1*^{-mZα} *Mlkl*^{-/-} mice that showed strong induction of ISGs. Therefore, the effects seen in *Adar1*^{-mZα} mice are not due to gene dosage effects of wild type ADAR1. However, comparison of *Adar1*^{mZα/mZα} with *Adar1*^{-mZα} mice showed that hemizygous expression of ADAR1 with mutated Zα domains causes a very strong ISG response and severe early postnatal pathology, whereas homozygous *Adar1*^{mZα/mZα} mice showed only a very mild induction of ISGs and no pathology, demonstrating a strong gene dosage effect of the ADAR1 mutant Zα allele. We speculate this might be caused by a reduced capacity of the Zα-mutated ADAR1 protein to edit critical RNAs, which is sufficient to maintain editing above a critical threshold under normal expression levels. However, in hemizygous mice, the resulting 50 % reduction in protein levels combined with the reduced editing capacity of this mutant most likely result in compromised editing efficiency, which falls below the critical threshold required to prevent strong MDA5 activation, thus triggering strong type I IFN responses and the resulting pathology.

3. Figure 2. Implicates ZBP-1 in *Adar1*^{-mZα} mice and demonstrates that ZBP1 deficiency could partially rescue the early lethality of *Adar1*^{-/-} *Mavs*^{-/-} mice, suggesting that ZBP1 synergises with MAVS to cause the pathology of ADAR1-deficient mice. However, the loss of ZBP-1 in the context of *Adar1*^{-mZα} animals was much more modest than the loss of MAVS, as evidenced by the data in Fig. 2L. The authors have not included *Adar1*^{-mZα} *MAVS*^{-/-} animals in many of

the analyses in Figure 2, especially Figure 2C and 2H, which makes it hard to judge the relative contributions of MAVS and ZBP-1 to the effects reported.

Indeed, in the previous version of the manuscript the data showing the rescue effects of MAVS and ZBP1 knockout were presented in different figures making it difficult to compare. We have now re-arranged the figures of the manuscript to allow for a direct comparison. Fig. 1 of the revised manuscript now includes the data from both the MAVS knockout and ZBP1 knockout rescue experiments in newborn mice, with all figure panels including both genotypes. Fig. 2 of the revised manuscript includes data from adult mice from the MAVS heterozygous and homozygous knockout and ZBP1 knockout crosses, which allows for the direct comparison of the effects of MAVS and ZBP1 deficiencies. Moreover, we have now performed additional RNAseq analysis in lung tissues from both newborn and adult mice from the MAVS and ZBP1 knockout rescue crosses, which are presented in Figures 1 and 2. As clearly shown in these figures, MAVS deficiency fully rescued early postnatal lethality, body weight and impaired erythropoiesis in *Adar1^{-mZα}* mice. ZBP1 knockout had a strong effect in preventing these pathologies, however, this was not as complete as that achieved by MAVS knockout. We also observed an interesting correlation between MAVS heterozygosity and ZBP1 deficiency particularly in the adult rescued mice from these crosses. As shown in Fig. 2, MAVS heterozygosity partially rescued the phenotype of *Adar1^{-mZα}* mice, as about 50 % of the *Adar1^{-mZα} Mavs^{WT/-}* mice survived to adulthood but showed strongly reduced body weight, severely impaired erythropoiesis and strong upregulation of ISGs. In a direct comparison, *Adar1^{-mZα} Zbp1^{-/-}* mice showed a stronger rescue effect compared to *Adar1^{-mZα} Mavs^{WT/-}* mice, with increased survival and body weight, much improved erythropoiesis and significantly reduced expression of ISGs (Fig. 2), demonstrating that ZBP1 deficiency had a stronger effect compared to MAVS heterozygosity. Based on these results, one could conclude that ZBP1 accounts for more than 50 % of the effect of MAVS, however, in our view such a statement might not be fully correct as heterozygous expression often does not necessarily mean 50 % reduction of protein function. Nevertheless, these results clearly demonstrate that a large part of the MAVS-dependent ISG response and associated pathology are dependent on ZBP1. In addition, RNAseq analysis in lungs of adult mice revealed that *Adar1^{-mZα} Mavs^{-/-}* mice showed a small but statistically significant upregulation of ISG expression compared to *Adar1^{WT/mZα}*, demonstrating that MAVS deficiency did not completely prevent the IFN response. Importantly, additional knockout of ZBP1 in *Adar1^{-mZα} Mavs^{-/-} Zbp1^{-/-}* mice fully normalised gene expression as these mice did not show the small upregulation of ISGs observed in *Adar1^{-mZα} Mavs^{-/-}* mice and were indistinguishable from wild type C57Bl/6 or *Adar1^{WT/mZα}* mice. These findings revealed that ZBP1 also has a MAVS-independent function in addition to its role in promoting the MAVS-dependent response.

4. Figure 3 suggests that endogenous Z-RNA likely derived from endogenous retroelements triggers ZBP1-dependent interferon responses in *Adar1^{-mZα}* mice, which again is fully anticipated from prior studies, including from the authors of the present study. Several previous

studies have shown that the $Z\alpha$ domain of ADAR1 was required for editing RNAs derived from EREs and SINEs in the mouse. Mutation of the $Z\alpha$ domains in ZBP-1 partially rescued the pathology seen with *Adar1*^{-/^{mZ α} animals, but the protection achieved by $Z\alpha$ domain mutation of ZBP-1 was less complete compared to ZBP1 deficiency, which implicates domains other than the $Z\alpha$ domains of ZBP-1 in sensing effects due to the ADAR1 $Z\alpha$ ^{-/} mutation. Of note, the *Adar1*^{-/^{mZ α} ZBP-1^{-/} genotype data is missing in Panels c, d, e and f of Figure 3, so this obscures the magnitude of the effects seen with mutation of the $Z\alpha$ domains in ZBP-1 relative to the null mutant. This is only included for panel 3g but would be very useful to see for the other panels here.}}

As discussed also above in our response to an introductory comment on the reviewer, while we agree that ZBP1 was previously implicated in sensing Z-RNA to induce cell death in anti-viral immunity and models of inflammation, and that ADAR1 was shown to edit RNA from endogenous retroelements, we respectfully disagree that the functional connection between ADAR1 and ZBP1 was expected and therefore is not novel. Of course, ADAR1 p150 and ZBP1 are the only two proteins in the mammalian genome known to contain $Z\alpha$ domains, therefore one might have thought these should functionally interact. This was indeed the starting point of our work more than 5 years ago, when we hypothesized that ADAR1 and ZBP1 could be functionally connected via their $Z\alpha$ domains. To the best of our knowledge, this connection has not been documented to date, therefore our studies are the first to experimentally demonstrate that $Z\alpha$ -dependent ZBP1 signalling causes a pathology when $Z\alpha$ -dependent ADAR1 function is impaired. Moreover, whereas the fact that ADAR1 $Z\alpha$ domain mutations were found to cause AGS and BSN clearly pointed towards an important function of this domain in restraining type I IFN responses, our results that these IFN responses are to a large extent dependent on $Z\alpha$ -dependent ZBP1 function have not been reported or predicted previously, and therefore reveal a novel and important cross-talk between ADAR1 and ZBP1 via their $Z\alpha$ domains.

We agree with the reviewer that the presentation of the data in the previous version of the paper was not optimally structured to allow a direct comparison of the effects of ZBP1 deficiency versus mutation of its $Z\alpha$ domains in rescuing the phenotype caused by mutation of the ADAR1 $Z\alpha$ domain. In the revised manuscript, we now include data from ZBP1 knockout mice in all panels of Fig. 3 to allow a direct comparison with the *Zbp1*^{mZ α /mZ α} crosses. As is now clearly shown in this figure, mutation of the ZBP1 $Z\alpha$ domains had a somewhat less pronounced effect in terms of rescuing the early postnatal lethality of *Adar1*^{-/^{mZ α} mice compared to ZBP1 knockout (about 50 % of *Adar1*^{-/^{mZ α} *Zbp1*^{mZ α /mZ α} mice survived up to 15 weeks of age compared to about 70 % of the *Adar1*^{-/^{mZ α} *Zbp1*^{-/} mice, Fig. 3d). Moreover, *Adar1*^{-/^{mZ α} *Zbp1*^{mZ α /mZ α} mice showed a similar body weight compared to *Adar1*^{-/^{mZ α} *Zbp1*^{-/} mice both in newborn pups and in the adult (Fig. 3e, h), but had reduced RBC, HGB and HCT levels in the blood (Fig. 3f, i), showing that $Z\alpha$ domain mutation of ZBP1 could not rescue the erythropoiesis defect to the extent of ZBP1 knockout. However, RNAseq analysis of lung tissue both from pups and from adult mice revealed an overall similar level of suppression of ISG expression in *Adar1*^{-/^{mZ α} *Zbp1*^{mZ α /mZ α} and}}}}}}

Adar1^{-mZα} *Zbp1*^{-/-} mice (Fig. 3g, j), showing that Zα-dependent sensing of endogenous ligands generated in ADAR1-deficient cells activates the ZBP1-dependent type I IFN response. Collectively, these results showed that the role of ZBP1 in promoting the type I IFN response and lethal pathology of *Adar1*^{-mZα} is largely dependent on Zα-mediated sensing of endogenous ligands, however, ZBP1 also exerts Zα-independent functions, as we showed previously in mouse models of MCMV infection and skin inflammation induced by epidermis specific RIPK1 knockout (Jiao et al., 2020).

5. Beyond demonstrating that ZPB-1 loss or mutation of the Z-RNA sensing domain in ZBP-1 suppressed the mild phenotype seen in *Adar1*^{mZα/mZα} mice, Figure 4 also contains largely negative data which suggests that the ZBP-1 response in ADAR1 mutated mice is RIPK3-independent. RIPK3 deficiency could neither prevent ISG upregulation nor rescue the lethal phenotype of *Adar1*^{-mZα} mice. However, it is known that ZBP-1 can also promote RIPK1/FADD/Caspase-8-dependent responses that have not been tested here. Similarly, it is also possible that engagement of TRIF by ZBP-1 could also account for events downstream but this has not been explored.

As discussed also above in our response to reviewer 2, we have now undertaken extensive additional studies to dissect the contribution of RIPK1, MLKL and FADD-caspase-8 to the ZBP1-dependent phenotype of *Adar1*^{-mZα} mice. To provide the most physiologically relevant results, we have used extensive genetic crosses to interrogate the role of downstream pathways *in vivo*. In fact, our revised manuscript contains data from 10 different double knockout or mutant mouse strains and 7 different triple knockout or mutant mouse strains. Our new genetic experiments that are now included in Fig. 4 of the revised manuscript revealed that inhibition of both RIPK3-MLKL-dependent necroptosis and FADD-caspase-8 dependent apoptosis could not rescue the phenotype of *Adar1*^{-mZα} mice. These results showed that ZBP1 induces the type I IFN response and associated pathology in *Adar1*^{-mZα} mice independently of necroptosis and FADD-caspase-8-induced apoptosis. We also considered the possibility that RIPK1 might be implicated downstream of ZBP1 in this model, particularly as early studies had provided evidence that ZBP1 induces cytokine production by engaging RIPK1 in a RHIM-dependent manner (Kaiser et al., 2008; Rebsamen et al., 2009). We therefore reasoned that ZBP1 might induce RIPK1-mediated proinflammatory signalling to drive the pathology independently from cell death, and for this reason we also crossed the *Adar1*^{-mZα} mice with mice expressing RIPK1 with mutated RHIM in an MLKL deficient background (*Ripk1*^{mR/mR} *Mkl1*^{-/-}). However, we found that also *Adar1*^{-mZα} *Ripk1*^{mR/mR} *Mkl1*^{-/-} mice were not rescued in terms of early lethality, impaired growth and erythropoiesis as well as ISG expression (Fig. 4), demonstrating that combined inhibition of necroptosis and RHIM-dependent RIPK1 signalling did not recapitulate the protective effect of ZBP1 deficiency. Therefore, our genetic studies provided experimental evidence demonstrating that ZBP1 promotes type I IFN responses and the associated pathology in *Adar1*^{-mZα} mice independently of RIPK3, MLKL-dependent necroptosis, FADD-caspase-8-dependent apoptosis as well as of RIPK1 RHIM-dependent signalling. We have also considered the possibility that

TRIF might be involved in ZBP1-mediated activation of type I IFN responses and the associated pathology of *Adar1*^{-mZα} mice. We find this is unlikely as a very interesting paper published last year by Poltorak and colleagues (Muendlein et al., 2021) showed that ZBP1 is implicated in TRIF-induced signalling, however, this function was dependent on RIPK1, therefore we would expect this to be inhibited in *Adar1*^{-mZα} *Ripk1*^{mR/mR} *Mlkl*^{-/-} mice. Nevertheless, our recent results arguing against a role of RIPK1 and RIPK3 warrant investigating the role of TRIF as a potential downstream effector of ZBP1 in this model. However, considering the time it takes to obtain the required permission from the local government authorities regulating animal experiments to start these crosses (usually several weeks to months) and then the time it takes to interbreed these strains to homozygosity, unfortunately we could not obtain these results within a time frame allowing to include them in the current manuscript. Taken together, our new genetic studies included in the revised manuscript demonstrate that ZBP1 promotes type I IFN responses and the early postnatally lethal pathology of *Adar1*^{-mZα} mice independently of RIPK3, RIPK1 RHIM-dependent signalling, MLKL-dependent necroptosis and FADD-caspase-8-dependent apoptosis. We find these results fascinating, as they show that ZBP1 drives this response by a mechanism different to its previously assigned RIPK3- and RIPK1-dependent functions in mediating antiviral defence and inflammation. Our results also revealed a very interesting dual function of ZBP1, which acts via RIPK3 to mediate the MAVS-independent pathology of *Adar1*^{-/-} mice, while it acts in a RIPK1- and RIPK3-independent manner to promote the MAVS-dependent IFN response and pathology in *Adar1*^{-mZα} mice. Even though at this stage we cannot deliver more specific mechanistic insight into how ZBP1 mediates this function, we hope that the reviewer will agree with us that our results revealed a new and exciting function of ZBP1 in regulating cellular responses to cellular RNA. Our findings provide a paradigm shift in our understanding of the mechanisms regulating cellular responses to retroelement-derived RNA, highlighting the critical role of Z-RNA formation and sensing by the two proteins containing Zα domains, namely ADAR1 and ZBP1, in controlling IFN responses to endogenous RNA.

6. Mutations in the Zalpha domain of ADAR led to surprisingly mild phenotype, as observed by others, leading to fully viable mice that have elevated ISG responses at 1 year but no major pathology. How do the authors interpret this result and the necessity to combine the Za mutation with a null allele to produce lethality? Have they also created mutations in the Zbeta domain of ADAR1 either alone, or in combination with Zalpha mutations to see whether IFN responses are enhanced in these animals?

We agree with the reviewer that it is surprising and interesting that *Adar1*^{mZα/mZα} mice remain healthy and only show a mild upregulation of type I IFN responses, whereas *Adar1*^{-mZα} animals develop such a severe type I IFN activation resulting in early postnatal lethality. This is however consistent with the findings that hemizygous expression of ADAR1 with Zα domain mutations (one Zα mutant allele and one p150-deficient ADAR1 allele) causes severe type I interferonopathies including AGS and BSN in humans, while no patients have been reported

carrying homozygous Z α mutations, although 0.2 - 0.3 % of humans carry ADAR1 Z α mutant alleles (Herbert, 2020). The most likely interpretation of this is that Z α mutation reduces the capacity of ADAR1 p150 to edit critical RNAs, however, when expressed at full level it is still sufficient to maintain editing above a critical threshold required to suppress strong MDA5 activation. In mice and humans with hemizygous expression of ADAR1 with mutated Z α domain, the resulting 50 % reduction in protein levels combined with the reduced editing capacity of this mutant most likely results in compromised editing efficiency, which falls below the critical threshold required to prevent strong MDA5 activation, thus triggering strong type I IFN responses and the resulting pathology. Our editing analysis indeed did not show altered A-to-I editing in spleen and brain from *Adar1*^{mZ α /mZ α} mice, whereas the same tissues of *Adar1*^{-mZ α} mice showed strongly reduced editing (Fig. 3b and Extended Data Fig. 8). Regarding the question about the possible function of the Zbeta domain of ADAR1, we have not generated mice with Zbeta mutations and are not aware of such mutations generated by others. However, although the ADAR1 Zbeta domain is structurally related to the Z α domains, it does not retain the capacity to interact with Z nucleic acids because it contains mutations on critical amino acids corresponding to Z α N173 and Y177 (Athanasiadis, 2012; Athanasiadis et al., 2005). Therefore, we would not expect Zbeta domain mutations to have a functional effect similar to that of the Z α domain mutations. Nevertheless, it would indeed be interesting to study the role of the Zbeta domain as its function remains enigmatic.

References

- Athanasiadis, A. (2012). Zalpha-domains: at the intersection between RNA editing and innate immunity. *Semin Cell Dev Biol* 23, 275-280.
- Athanasiadis, A., Placido, D., Maas, S., Brown, B.A., 2nd, Lowenhaupt, K., and Rich, A. (2005). The crystal structure of the Zbeta domain of the RNA-editing enzyme ADAR1 reveals distinct conserved surfaces among Z-domains. *J Mol Biol* 351, 496-507.
- de Reuver, R., Dierick, E., Wiernicki, B., Staes, K., Seys, L., De Meester, E., Muyldermans, T., Botzki, A., Lambrecht, B.N., Van Nieuwerburgh, F., Vandenabeele, P., and Maelfait, J. (2021). ADAR1 interaction with Z-RNA promotes editing of endogenous double-stranded RNA and prevents MDA5-dependent immune activation. *Cell Rep* 36, 109500.
- Herbert, A. (2020). Mendelian disease caused by variants affecting recognition of Z-DNA and Z-RNA by the Zalpha domain of the double-stranded RNA editing enzyme ADAR. *Eur J Hum Genet* 28, 114-117.
- Ishii, K.J., Kawagoe, T., Koyama, S., Matsui, K., Kumar, H., Kawai, T., Uematsu, S., Takeuchi, O., Takeshita, F., Coban, C., and Akira, S. (2008). TANK-binding kinase-1 delineates innate and adaptive immune responses to DNA vaccines. *Nature* 451, 725-729.
- Jiao, H., Wachsmuth, L., Kumari, S., Schwarzer, R., Lin, J., Eren, R.O., Fisher, A., Lane, R., Young, G.R., Kassiotis, G., Kaiser, W.J., and Pasparakis, M. (2020). Z-nucleic-acid sensing triggers ZBP1-dependent necroptosis and inflammation. *Nature* 580, 391-395.

- Kaiser, W.J., Upton, J.W., and Mocarski, E.S. (2008). Receptor-interacting protein homotypic interaction motif-dependent control of NF-kappa B activation via the DNA-dependent activator of IFN regulatory factors. *J Immunol* *181*, 6427-6434.
- Maurano, M., Snyder, J.M., Connelly, C., Henao-Mejia, J., Sidrauski, C., and Stetson, D.B. (2021). Protein kinase R and the integrated stress response drive immunopathology caused by mutations in the RNA deaminase ADAR1. *Immunity* *54*, 1948-1960 e1945.
- Muendlein, H.I., Connolly, W.M., Magri, Z., Smirnova, I., Ilyukha, V., Gautam, A., Degterev, A., and Poltorak, A. (2021). ZBP1 promotes LPS-induced cell death and IL-1beta release via RHIM-mediated interactions with RIPK1. *Nat Commun* *12*, 86.
- Nakahama, T., Kato, Y., Shibuya, T., Inoue, M., Kim, J.I., Vongpipatana, T., Todo, H., Xing, Y., and Kawahara, Y. (2021). Mutations in the adenosine deaminase ADAR1 that prevent endogenous Z-RNA binding induce Aicardi-Goutieres-syndrome-like encephalopathy. *Immunity* *54*, 1976-1988 e1977.
- Rebsamen, M., Heinz, L.X., Meylan, E., Michallet, M.C., Schroder, K., Hofmann, K., Vazquez, J., Benedict, C.A., and Tschopp, J. (2009). DAI/ZBP1 recruits RIP1 and RIP3 through RIP homotypic interaction motifs to activate NF-kappaB. *EMBO Rep* *10*, 916-922.
- Schade, M., Turner, C.J., Lowenhaupt, K., Rich, A., and Herbert, A. (1999). Structure-function analysis of the Z-DNA-binding domain Zalpha of dsRNA adenosine deaminase type I reveals similarity to the (alpha + beta) family of helix-turn-helix proteins. *EMBO J* *18*, 470-479.
- Schwarzer, R., Jiao, H., Wachsmuth, L., Tresch, A., and Pasparakis, M. (2020). FADD and Caspase-8 Regulate Gut Homeostasis and Inflammation by Controlling MLKL- and GSDMD-Mediated Death of Intestinal Epithelial Cells. *Immunity* *52*, 978-993 e976.
- Takaoka, A., Wang, Z., Choi, M.K., Yanai, H., Negishi, H., Ban, T., Lu, Y., Miyagishi, M., Kodama, T., Honda, K., Ohba, Y., and Taniguchi, T. (2007). DAI (DLM-1/ZBP1) is a cytosolic DNA sensor and an activator of innate immune response. *Nature* *448*, 501-505.
- Tang, Q., Rigby, R.E., Young, G.R., Hvidt, A.K., Davis, T., Tan, T.K., Bridgeman, A., Townsend, A.R., Kassiotis, G., and Rehwinkel, J. (2021). Adenosine-to-inosine editing of endogenous Z-form RNA by the deaminase ADAR1 prevents spontaneous MAVS-dependent type I interferon responses. *Immunity* *54*, 1961-1975 e1965.

Reviewer Reports on the First Revision:

Referees' comments:

Referee #1 (Remarks to the Author):

In the revised manuscript, Jiao and Wachmuth et al. improved their manuscript in several respects, especially with regard to the two major concerns I raised in my initial comments.

First, the authors provided the MLKL cross (now shown in Fig. 4). This clearly and definitively disproves the hypothesis that necroptosis is of importance in this setting. Consequently, the authors went even further to assess the roles of FADD/MLKL and FADD/RIPK3 and confirmed that also these crosses did not rescue the phenotype. Further chasing the mechanism, they crossed RIPK1 RHIM-mutant mice to the ADAR1-/Zalpha-mice thereby additionally ruled out a role for RIPK1 as a potential regulator downstream of ZBP1. This part is indeed very well done and suggests that ZBP1 mediates the phenotype by a novel, yet unidentified, mechanism which is distinct from apoptosis and necroptosis, but which involves MAVS and ZBP1.

Second, they carefully added data to assess the brains of these mice. I asked for this because the AGS is not the only known human syndrome associated with ADAR1 mutations, and in theory, bilateral striatal necrosis (BSN) might be driven by a similar mechanism. This is another important improvement of the manuscript.

All other concerns raised by me were adequately responded to, so that given the importance of these discoveries, I recommend to consider this manuscript for publication in nature.

Referee #2 (Remarks to the Author):

The additional genetics elevate the story considerably. I support publication.

Referee #3 (Remarks to the Author):

The authors have done an admirable job of addressing all of the issues raised. It is indeed puzzling that the ZBP-1-dependent effects are RIPK3/MLKL, FADD/ caspase-8, and RIPK1-independent. As the authors suggest in the concluding section, possibly there is complex redundancy between the TRIF/RIPK1, FADD/Casp-8 and RIPK3/MLKL pathways here, or a completely novel route to IFN responses in this setting.

I have some minor remaining comments

1. It is very difficult to visualize many of the data as they are presented in landscape format and all graphs are very small indeed. Presenting the data in portrait format would allow individual data panels, especially graphs, to be presented at a more practical size.

2. Figure 1 legend is somewhat misleading: "ZBP1 promotes interferon induction and early postnatal

lethality in mice hemizygotously expressing ADAR1 with mutated Z α domain."

Given that MAVS makes a greater contribution here, this should be 'ZBP1 contributes to interferon induction.....'. Figure 2 legend more correctly summarizes the situation.

3. It's possible that I missed this, but the authors could comment on the likely sequence of events in operation upon loss or mutation of ADAR. The MDA5/MAVS pathway seems to be the major upstream sensor that is activated here, leading to induction of ZBP-1 downstream, which then promotes some of the IFN and other downstream events.

Author Rebuttals to First Revision:

Point-by-point response to the reviewer comments

Referee #1:

In the revised manuscript, Jiao and Wachmuth et al. improved their manuscript in several respects, especially with regard to the two major concerns I raised in my initial comments.

First, the authors provided the MLKL cross (now shown in Fig. 4). This clearly and definitively disproves the hypothesis that necroptosis is of importance in this setting. Consequently, the authors went even further to assess the roles of FADD/MLKL and FADD/RIPK3 and confirmed that also these crosses did not rescue the phenotype. Further chasing the mechanism, they crossed RIPK1 RHIM-mutant mice to the ADAR1-/Zalpha-mice thereby additionally ruled out a role for RIPK1 as a potential regulator downstream of ZBP1. This part is indeed very well done and suggests that ZBP1 mediates the phenotype by a novel, yet unidentified, mechanism which is distinct from apoptosis and necroptosis, but which involves MAVS and ZBP1.

Second, they carefully added data to assess the brains of these mice. I asked for this because the AGS is not the only known human syndrome associated with ADAR1 mutations, and in theory, bilateral striatal necrosis (BSN) might be driven by a similar mechanism. This is another important improvement of the manuscript.

All other concerns raised by me were adequately responded to, so that given the importance of these discoveries, I recommend to consider this manuscript for publication in nature.

We thank the reviewer for their positive assessment of our revised manuscript.

Referee #2:

The additional genetics elevate the story considerably. I support publication.

We thank the reviewer for their positive assessment of our revised manuscript.

Referee #3

(Remarks to the Author)

The authors have done an admirable job of addressing all of the issues raised. It is indeed puzzling that the ZBP-1-dependent effects are RIPK3/MLKL, FADD/ caspase-8, and RIPK1-independent. As the authors suggest in the concluding section, possibly there is complex redundancy between the TRIF/RIPK1, FADD/Casp-8 and RIPK3/MLKL pathways here, or a completely novel route to IFN responses in this setting.

We thank the reviewer for their positive assessment of our revised manuscript.

I have some minor remaining comments

1. It is very difficult to visualize many of the data as they are presented in landscape format and all graphs are very small indeed. Presenting the data in portrait format would allow individual data panels, especially graphs, to be presented at a more practical size.

We have made an effort to improve the presentation of the data in all main figures. We hope that all data and particularly the graphs are now of adequate size and clear and easy to follow and understand.

2. Figure 1 legend is somewhat misleading: "ZBP1 promotes interferon induction and early postnatal lethality in mice hemizygotously expressing ADAR1 with mutated Za domain."

Given that MAVS makes a greater contribution here, this should be 'ZBP1 contributes

to interferon induction.....'. Figure 2 legend more correctly summarizes the situation. We agree with the reviewer and have changed the title of this legend as suggested.

3. It's possible that I missed this, but the authors could comment on the likely sequence of events in operation upon loss or mutation of ADAR. The MDA5/MAVS pathway seems to be the major upstream sensor that is activated here, leading to induction of ZBP-1 downstream, which then promotes some of the IFN and other downstream events.

We thank the reviewer for this remark. While at this stage it is not clear how ZBP1 promotes ISG expression, we agree with the interpretation that ZBP1 likely acts downstream of MAVS to induce IFN activation in mice with ADAR mutation. We have included the following sentence in the discussion in the revised manuscript text: "MAVS-deficiency nearly completely normalized whereas ZBP1 deficiency partially rescued ISG expression and the pathology of *Adar1*^{-*mZα*} mice, suggesting that ZBP1 is induced downstream of MAVS and contributes to type I IFN activation and the associated pathologies."